# Adaptive design of mRNA-loaded extra-cellular vesicles for targeted immunotherapy of cancer

Shiyan Dong [1,2,12], Xuan Liu[2,3,12], Ye Bi[4,12], Yifan Wang [2,12], Abin Antony [2], DaeYong Lee[5], Kristin Huntoon [5], Seongdong Jeong[5], Yifan Ma[6], Xuefeng Li [2], Weiye Deng[2], Benjamin R. Schrank [2], Adam J. Grippin [2], JongHoon Ha[2], Minjeong Kang[2], Mengyu Chang[2], Yarong Zhao[1], Rongze Sun[1], Xiangshi Sun[1], Jie Yang[1], Jiayi Chen[1], Sarah K. Tang[5], L. James Lee [7,8], Andrew S. Lee [9,10], Lirong Teng[1], Shengnian Wang [3] ✉, Lesheng Teng [1] ✉, Betty Y. S. Kim [5,11], Zhaogang Yang[1,2] ✉ & Wen Jiang [2,11] ✉

The recent success of mRNA therapeutics against pathogenic infections has increased interest in their use for other human diseases including cancer. However, the precise delivery of the genetic cargo to cells and tissues of interest remains challenging. Here, we show an adaptive strategy that enables the docking of different targeting ligands onto the surface of mRNA-loaded small extracellular vesicles (sEVs). This is achieved by using a microfluidic electroporation approach in which a combination of nano- and milli-second pulses produces large amounts of IFN-γ mRNA-loaded sEVs with CD64 over-expressed on their surface. The CD64 molecule serves as an adaptor to dock targeting ligands, such as anti-CD71 and anti-programmed cell death-ligand 1 (PD-L1) antibodies. The resulting immunogenic sEVs (imsEV) preferentially target glioblastoma cells and generate potent antitumour activities in vivo, including against tumours intrinsically resistant to immunotherapy. Together, these results provide an adaptive approach to engineering mRNA-loaded sEVs with targeting functionality and pave the way for their adoption in cancer immunotherapy applications.

Messenger RNA (mRNA) has demonstrated promising therapeutic potential in various clinical applications in recent years[1]. However, overcoming physiological barriers to achieve both tissue- and cell-targeted delivery remains challenging for current conventional mRNA delivery systems. Extracellular vesicles (EVs) have emerged as promising delivery vehicles for RNA-based therapeutics because of their advantages over other mRNA delivery systems, including their excellent biosafety and biocompatibility, stability against degradation, and

[1]School of Life Science, Jilin University, Changchun 130012, China. [2]Department of Radiation Oncology, The University of Texas MD Anderson Cancer Center, Houston, TX 77030, USA. [3]Chemical Engineering, Institute for Micromanufacturing, Louisiana Tech University, Ruston, LA 71272, USA. [4]Practice Training Center, Changchun University of Chinese Medicine, Changchun 130117, China. [5]Department of Neurosurgery, The University of Texas MD Anderson Cancer Center, Houston, TX 77030, USA. [6]Department of Biomedical Engineering, The Ohio State University, Columbus, OH 43210, USA. [7]Department of Chemical and Biomolecular Engineering, The Ohio State University, Columbus, OH 43210, USA. [8]Spot Biosystems Ltd., Palo Alto, CA 94305, USA. [9]Institute for Cancer Research, Shenzhen Bay Laboratory, Shenzhen 518055, China. [10]School of Chemical Biology and Biotechnology, Peking University Shenzhen Graduate School, Shenzhen 518055, China. [11]Brain Tumor Center, The University of Texas MD Anderson Cancer Center, Houston, TX 77030, USA. [12]These authors contributed equally: Shiyan Dong, Xuan Liu, Ye Bi, Yifan Wang. ✉e-mail: swang@latech.edu; tenglesheng@jlu.edu.cn; bykim@mdanderson.org; zhaogangyang@jlu.edu.cn; wjiang4@mdanderson.org

ability to cross physiological barriers such as the blood–brain barrier (BBB)[2–5]. Although small EVs (sEVs) have been successfully used to deliver full-length transcripts of mRNAs for cancer therapy[6,7], their use to carry mRNAs specifically chosen to restore tumour immunogenicity and to improve immunotherapy effectiveness in tumours has not been evaluated. Here, we introduce a microfluidic electroporation system combined with a two-step pulse stimulation approach to generate large quantities of sEVs that overexpress CD64 on their surfaces and actively recruit mRNA for IFN-γ within themselves. In these therapeutic sEVs, CD64 functions as a docking site for both anti-CD71 and anti-PD-L1 antibodies to facilitate sEV targeting to glioblastoma (GBM) tumours. We show here that these immune sEVs (imsEVs) can successfully target GBM and induce immunotherapy effects in vivo. Moreover, this imsEV, upon reaching the tumour microenvironment, leads to the upregulation of major histocompatibility complex class I (MHC-I) expression, which is often downregulated in solid tumours like GBM to trigger immune escape[8,9], thereby enhancing the antitumour effects of immunotherapy (Fig. 1). This imsEV may represent a suitable strategy for achieving mRNA loading, tumour targeting, and microenvironmental regulation to enhance the effectiveness of cancer immunotherapies.

## Results

### High-throughput generation of sEVs by nanosecond pulse electroporation (nsEP)

To design an electroporation system that produces highly efficient mRNA-loaded sEVs, source cells are subjected to a two-step electroporation process. First, nanosecond electropulses are used to transiently permeabilize the membrane structure of organelles inside source cells, which are then exposed to millisecond pulses that permeabilize the cell plasma membrane (Fig. 2a and Supplementary Fig. 1). This nanosecond pulse electroporation (nsEP) approach allows high cell transfection performance (Supplementary Fig. 2) and large-scale generation of sEVs, leading to a 45-fold increase in sEV production (relative to control) by mouse embryonic fibroblasts (MEFs) (Fig. 2b) and a 32-fold increase by the human embryonic kidney 293T (HEK293T) cell lines (Supplementary Fig. 3). Non-significant differences in sEV release were observed with or without the presence of plasmids in host cells during nsEP treatment (Fig. 2b and Supplementary Fig. 3). Adjusting the pulse parameters of nsEP allows further optimization of the quantity of sEV secretion as follows: (1) when the pulse voltage is raised from 50 to 200 V, more sEVs are secreted from host cells, reaching a plateau at 180 V (Fig. 2c). A slight decrease in cell viability is observed when the voltage exceeds 180 V (Fig. 2d and Supplementary Table 1). (2) The frequency and duration of nanopulses also have an influence on this nsEP-triggered sEV secretion, with cells treated at 100 kHz and 600 ns releasing the most sEVs (Supplementary Fig. 4a, b and Supplementary Tables 2, 3) whilst maintaining high cell viability (Supplementary Fig. 4c, d). The size distribution of sEVs secreted by microsecond electroporation pulses (msEP) or nsEP-stimulated cells and natural secretion groups (untreated control) were similar and shared the same morphological features, all showing a dominant size of about 120 nm (Fig. 2e and Supplementary Fig. 5). We further characterized the sEVs generated by nsEP after purification by using western blotting, which demonstrated the sole presence of sEVs without apoptotic bodies (Supplementary Fig. 6).

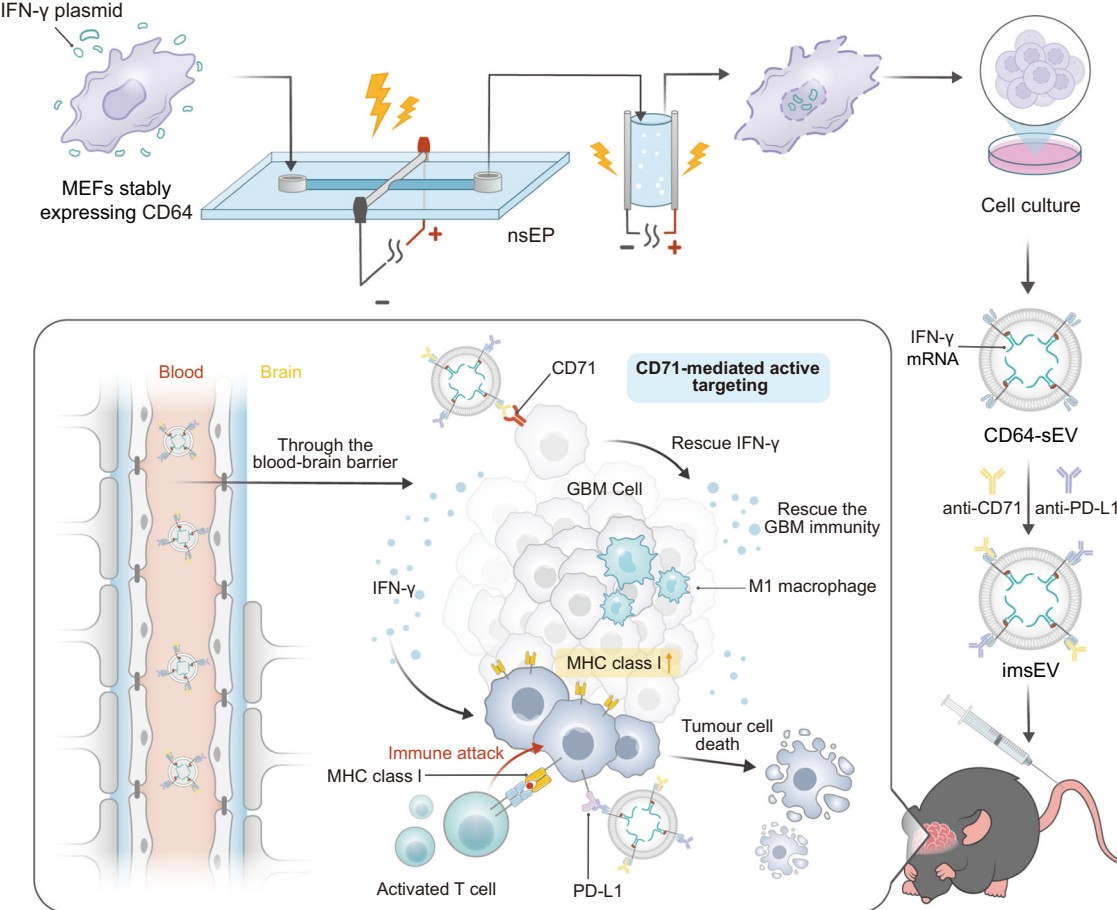

**Fig. 1 | Schematic representation of large-scale production and therapeutic mechanisms of imsEV.** As a mRNA delivery platform, imsEV was generated using a nanosecond-electroporation (nsEP) system and loaded with anti-CD71, anti-PD-L1, and IFN-γ mRNA simultaneously for GBM treatment.

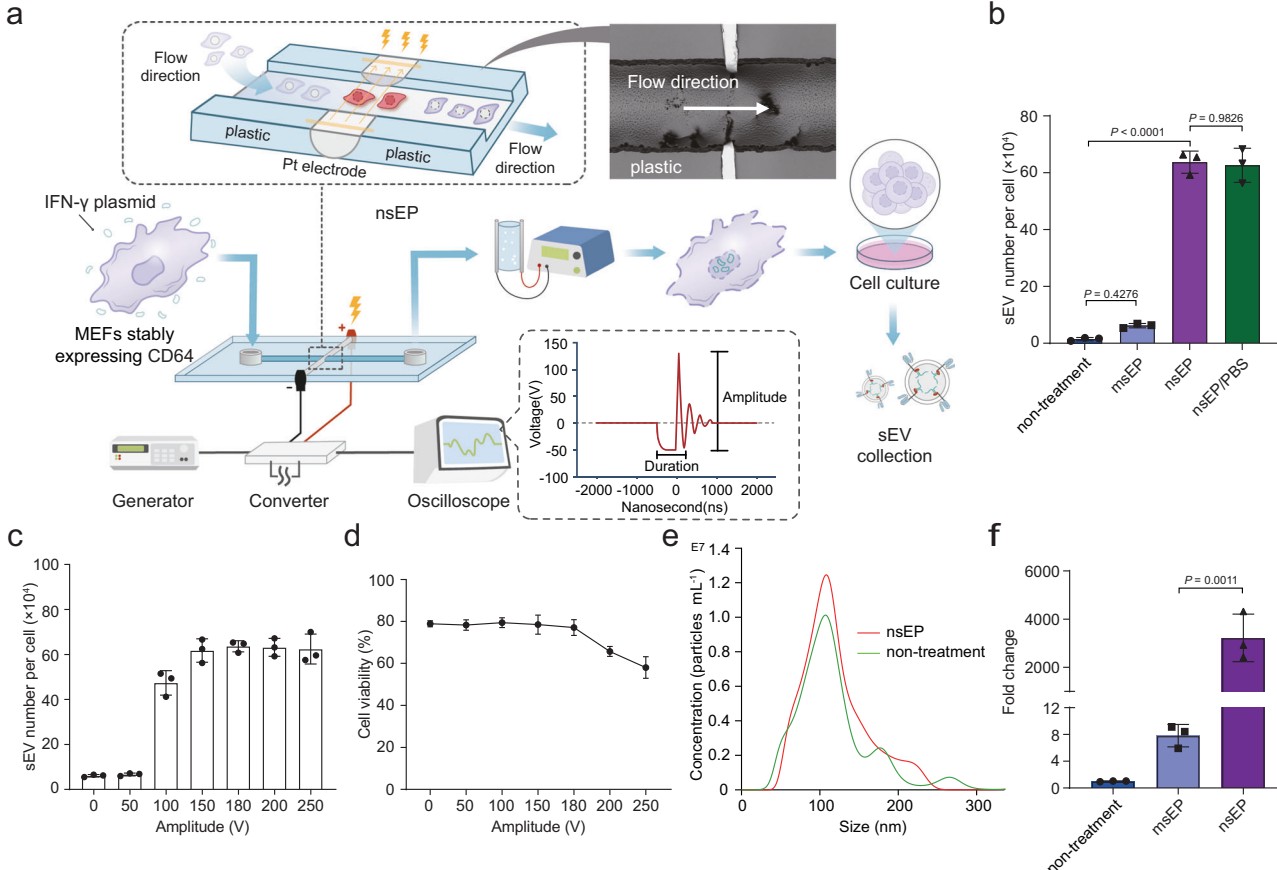

**Fig. 2 | Large-scale generation of sEVs by a nsEP system. a** Schematic representation of the nsEP system for sEV generation. **b** The sEV number per cell produced by mouse embryonic fibroblasts (MEFs) in untreated control, microsecond electroporation pulses (msEP), nsEP, and nsEP/PBS treatment groups. The total amount of sEVs = concentration of sEVs × volume. The number of viable cells was calculated by trypan blue staining and cell counting. The number of sEVs produced per cell = total number of sEVs/number of living cells. **c** Number of sEVs produced per MEF after nsEP system at voltage amplitudes from 0 to 250 V. **d** Viability of MEFs after nsEP system at voltage amplitudes from 0 to 250 V. **e** Size measurements and size distribution of sEVs produced by both the nsEP system (red) and the untreated control (green) cells. **f** RT-qPCR of IFN-γ mRNA revealed that sEVs produced by nsEP treatment contained much larger quantities of transcribed mRNAs than sEVs produced by other methods. Data in **b**–**f** are presented as means ± standard deviation (SD), *n* = 3 biologically independent samples; statistical significance was calculated by one-way analysis of variance with Tukey's multiple comparisons test. Source data are provided as a Source Date file.

With the microfluidic setup and multiple units operated in parallel, a large production capacity of sEVs can be achieved with high throughput ($5.0 \times 10^7$ cells in 5 min), which would be sufficient for many clinical applications with regard to both sEV quantity and processing time. Quantitative reverse transcription polymerase chain reaction (RT-qPCR) confirmed that levels of mRNAs complementary to the IFN-γ plasmid DNAs that had been encapsulated within the sEVs were $10^3$-fold higher in the nsEP system-treated groups than in the control samples (Fig. 2f). Similar levels of sEV secretion induced by nsEP were found in HEK293T cells (Supplementary Fig. 7 and Supplementary Tables 4–6).

**Mechanism of nsEP-induced sEV secretion**
To reveal the cellular mechanisms underlying the nsEP-triggered release of sEVs, we used proteomic profiling to identify the relevant proteins involved in this process. A total of 4423 quantifiable proteins were evaluated, among which 1344 were expressed at statistically different levels before and after the nsEP stimulation when the fold change threshold was set at 1.5. These 1344 proteins were further classified according to their functions by using Gene Ontology annotations. Our results reveal that the nsEP treatment induces multiple cellular and metabolic processes, biological regulation, and responses to stimuli (Fig. 3a). Proteins that differed in the cellular-process component were Gene Ontology-enriched to

obtain three sEV-associated clusters, extracellular space, and sEVs involving 104 proteins (Fig. 3b and Supplementary Table 7). Upon identifying proteins involved in the regulation of sEV secretion, we used protein-protein interaction network analysis to recapitulate proteins associated with classic sEV-generation pathways, including intraluminal vesicle formation (MLKL, Sdcbp, and Cdc2), protein ubiquitination (Ndfip1), endosomal sorting complex required for transport (ESCRT)-dependent cargo sorting (Vps36, Vta1, Chmp4b, Mvb12a, Chmp3, and Chmp5), small GTP-binding proteins leading to exosomal budding from the plasma membranes (Rab8a, Rab27b, and Rala), lysosomal degradation of multivesicular bodies by ISGylation (ISG15 and Usp18) or autophagy (Prnp), and SNARE interactions in vesicular transport (Stx17, Vamp7, Ykt6, and Vamp8) (Fig. 3c, d and Supplementary Table 8). A heatmap was generated for the top 95 proteins showing differences of 6-fold or more (Fig. 3e and Supplementary Table 9). Among them, MLKL was two orders of magnitude higher than other regulatory proteins involved in sEV generation. This suggests that the increased sEV secretion during nsEP stimulation depends strongly on MLKL (Fig. 3d and Supplementary Table 9). To confirm the involvement of MLKL in sEV trafficking, we silenced MLKL expression in MEFs and generated an MLKL-knockdown cell line and analysed the effects on sEV generation after nsEP treatment (Fig. 3f). We found that downregulating MLKL led to significant inhibition of sEV production from cells

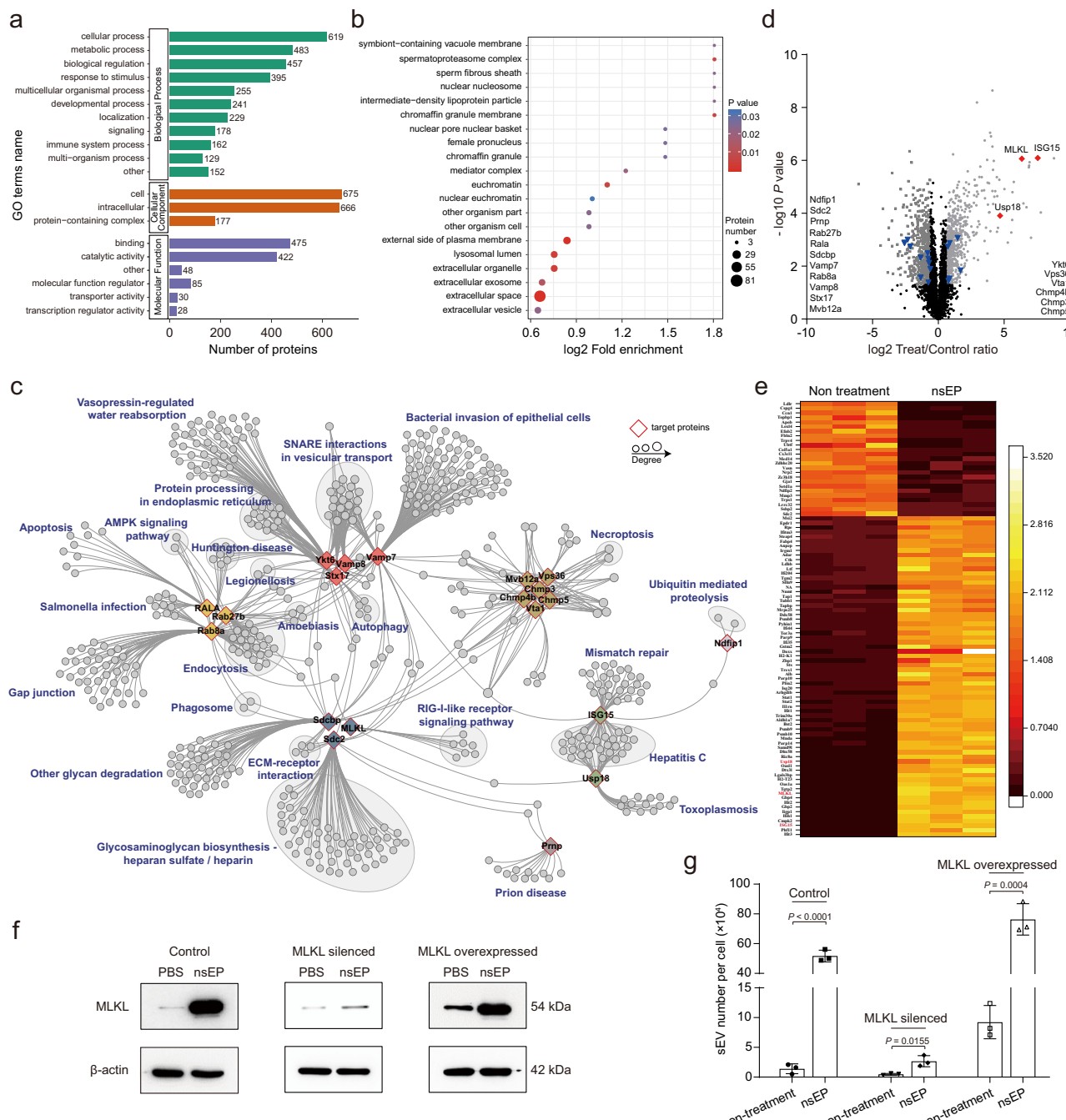

**Fig. 3 | Proteomic profiling of nsEP-treated MEF cells. a** Gene Ontology annotation of proteins expressed at different levels before and after nsEP treatment. **b** Gene Ontology enrichment of cellular components. **c** STRING-based protein–protein interaction (PPI) network analysis of identified proteins with higher expression confidence of 0.7; proteins of interest are displayed as a single diamond-shaped node, and differential proteins that interacted with nodes were analysed by Kyoto Encyclopaedia of Genes and Genomes pathway. **d** Proteomics proteins volcano plot analysis for sEVs derived from nsEP-treated MEFs. Red triangles and blue diamonds represent proteins associated with the induction of sEVs, among which red indicates proteins that were more highly expressed than others.

**e** Heatmap of the top 95 proteins differentially expressed after nsEP treatment. NA indicates UPF0600 protein C5orf51 homologue (a protein whose gene name is unknown). Proteins associated with sEV secretion are highlighted. **f** Representative western blots of MLKL identified in the indicated treatment conditions. Repeated three times independently with similar results obtained. **g** Numbers of sEVs per cell produced by MLKL knockdown or MLKL-overexpressing MEFs. Equal amounts of protein were used for the proteomic analysis, and data were from three independent biological replicates. Data in **g** represent means ± SD, $n = 3$ biologically independent samples; statistical significance was analysed with unpaired two-tailed Student's $t$-tests. Source data are provided as a Source Date file.

exposed to nsEP. A slight increase in MLKL protein expression was observed after nsEP treatment, leading to limited recovery of sEV generation (Fig. 3f and g). Conversely, overexpressing MLKL in MEFs promotes the nsEP-stimulated sEV production, further implicating MLKL in the regulation of sEV secretion during this process (Fig. 3f and g).

**Preparation and characterization of imsEVs**

We further developed a strategy to actively incorporate target mRNAs into these secreted sEVs, with a goal of restoring immunogenicity in solid tumours[10–12]. We chose GBM as a preclinical model system, as it is an aggressive tumour with no effective treatment currently available and does not respond to immunotherapy[13–16]. Another major obstacle

to effective immunotherapy in GBM is the downregulation of MHC-I proteins on the tumour surface, which leads to poor antigen presentation[17–19]. The low mutational load, with correspondingly few infiltrating T cells and M1 macrophages, plus the downregulation of MHC-I, results in a highly immunosuppressive tumour microenvironment[8,9,20]. To prepare imsEVs for immunotherapy of GBM, we first set out to attach both GBM-targeting (anti-CD71) and immunotherapy (anti-PD-L1) antibodies to the surfaces of imsEVs to enable their specific targeting of GBM for immunotherapeutic effects. We took advantage of CD64, an Fcγ receptor that can bind to the constant region of IgG heavy chain[21,22]. We first constructed a MEF line that stably expressed CD64-DsRed (CD64-DsRed+) (Supplementary Fig. 8). In CD64-DsRed+ cells, co-localization of CD64-DsRed with CD63-GFP, a classic surface marker protein of sEVs, was associated with a significant increase in CD64 content in sEVs generated from CD64+ cells, suggested that the presence of CD64 expression on the surface of sEVs generated by our method (Fig. 4a and b). To further evaluate the topology of CD64 on sEVs, we inserted a 3XFLAG epitope into the N-terminus of CD64 and inserted a Myc epitope into the c-terminus of CD64, and used a pulldown assay with anti-FLAG and anti-Myc beads to confirm that the N-terminus of CD64 was localized to the external surface of these sEVs (Fig. 4c and Supplementary Fig. 9). To achieve active loading of target mRNA into sEVs, we took advantage of the N peptide, which specifically binds to the box B sequence in the RNA, as follows[23,24]. We cloned the N peptide onto the C-terminus of the CD64 protein (which is located inside sEVs when they are formed by the inward budding of endosomal membranes, as shown in Fig. 4d), and a box B sequence to the 3' end of IFN-γ mRNA by engineering the IFN-γ plasmid. Briefly, the N peptide fused to the C-terminus of CD64 binds specifically to box B to recruit IFN-γ mRNA into the sEVs. The inward budding of endosomal membranes leaves the N-terminus of the CD64 protein outside of an sEV when it forms, and the C-terminus conjugated with the N peptide is within the sEV. The former (N-terminus) helps gain the specific surface targeting function while the latter (C-terminus) promotes the enrichment of IFN-γ mRNA for the imsEVs. We reasoned that the highly specific binding affinity between N peptide and box B sequence would enrich box B-fused IFN-γ mRNA within sEVs during their formation (Fig. 4d). We then transfected box B-IFN-γ or control IFN-γ into MEFs that stably express CD64-N peptide, and harvested the secreted sEVs for mRNA analysis by RT-qPCR. As shown in Fig. 4e, IFN-γ mRNAs fused with box B-sequence were greatly enriched in sEVs produced by the CD64-N peptide-overexpressing cells. This was further verified by using a tethered lipoplex nanoparticle (TLN) biochip that contains molecular beacons against IFN-γ mRNA. The mean fluorescence intensity in a single sEV from the box B-IFN-γ group was 3.5 times higher than that from the control IFN-γ plasmid group (Fig. 4f, g, and Supplementary Fig. 10a). Similarly, the percentage of sEVs containing IFN-γ mRNA was 20% higher in the box B-IFN-γ group than in the control IFN-γ plasmid group (Fig. 4f, h, and Supplementary Fig. 10b). Flow cytometry-based detection of single sEV yielded similar conclusions (Supplementary Fig. 10c and d). These findings suggest that our active loading strategy indeed enhances the loading of specific mRNA (IFN-γ mRNA in this study) into imsEV, thus potentially improving their potency.

Since CD71 is overexpressed on GBM cell lines[25] but not on MEF or HEK293T cells (Fig. 4i and Supplementary Fig. 11), we chose CD71 as the active GBM-targeting marker. To check the binding affinity of these CD64+ sEVs with antibodies to CD71 or PD-L1, we incubated the sEVs with either anti-CD71 (Mouse IgG2a, κ) or anti-PD-L1 (Rat IgG2b) at different sEV/antibody (w/w) ratios. As shown in Fig. 4j, binding was noted between total sEV protein and anti-CD71, starting at a ratio of 1:0.5 as evidenced by the detection of IgG heavy and light chains on western blotting; additional binding was observed as the IgG concentration was increased before reaching a plateau, in which the ratio of total sEV protein to IgG was 1:4 (w/w). Similar results were also

observed in the anti-PD-L1 group (Fig. 4j). To achieve both targeting and immunotherapy effects in these mRNA-containing imsEV, we incubated the sEVs with anti-CD71 and anti-PD-L1 antibodies and verified their co-existence by western blotting (Supplementary Fig. 12). Aiming to improve the targeting and immunotherapy efficacy, anti-CD71 and anti-PD-L1 at different antibody ratios were investigated for the optimal co-localization rate. We found that most (>70%) of the imsEV were conjugated with both anti-CD71 and anti-PD-L1 antibodies when the ratio of anti-CD71 to anti-PD-L1 was 1:3 (Fig. 4k and Supplementary Table 10). Therefore, in the following experiments, the ratio of CD64-sEV/anti-CD71/anti-PD-L1 was set at 1:1:3 (w/w/w). We further verified the binding capability by flow cytometry (Supplementary Fig. 13). The binding of antibodies on sEVs makes the particle size of the imsEVs derived from MEFs slightly larger than the regular ones (~about 10 nm), as shown in Fig. 4l and m. Cryo-electron microscopy analysis demonstrated that CD64-sEV and imsEV derived from MEF cells treated with the nsEP system exhibited electron-dense cargo in the lumen, whereas sEV from untreated MEFs were devoid of such content. The surface characteristics of imsEV, relative to CD64-sEV, showed increased depth, thereby confirming the presence of IgG attached to the surface of the imsEVs. Similar results were also obtained in HEK293T cells (Supplementary Fig. 14a–e).

## In vitro study of imsEV for GBM therapy

To evaluate the potential therapeutic utility of imsEV, we first studied their cytotoxicity in vitro and found no significant cytotoxicity in the two GBM cell lines tested (SB28 and GL261) at 24 or 48 h (Supplementary Fig. 15). Linking the CD71 antibody to imsEV significantly increased the uptake of the imsEVs by both SB28 and GL261 cells in vitro (Fig. 5a–d and Supplementary Fig. 16a–d). Our studies of endocytosis showed strong co-localization of imsEV with transferrin and partial co-localization of imsEV with other endocytosis markers, indicating that entry of imsEV into target cells was regulated mainly by clathrin-mediated endocytosis (Fig. 5e and Supplementary Fig. 17). Indeed, inhibition of clathrin-mediated endocytosis significantly reduced the cellular uptake of imsEVs, confirming the importance of this pathway in the regulation of imsEV uptake (Fig. 5f). After imsEVs had been incubated with GBM cells for 48 h, we noted much higher concentrations of IFN-γ protein in both the culture medium and cytosol as measured by ELISA (Fig. 5g and Supplementary Fig. 18). Because IFN-γ can upregulate the expression of MHC-I on GBM cells, thereby affecting their immunogenicity, we further investigated MHC-I expression after imsEV treatment by flow cytometry. We noted that the proportion of MHC-I-positive cells increased significantly at 48 h after imsEV treatment (Fig. 5h and i). Western blotting results further verified the increased MHC-I expression in the imsEV-treated condition (Fig. 5j).

To further investigate the therapeutic potential of imsEV for in vivo applications, we evaluated the biosafety of imsEV through co-incubation with blood samples. No haemolytic toxicity was observed at the studied concentrations of imsEV (Supplementary Fig. 19). The results of mouse biosafety and biocompatibility experiments showed that at 24 h after administration of imsEV, serum markers including ALT, AST, BUN, and creatinine in the blood of healthy mice were all within the normal range, with values that are similar to the control groups (Supplementary Fig. 20). Most sEVs were found to accumulate in the livers of healthy mice, and their fluorescence in the brain was weak for mice injected with sEV and CD64-sEV, but slightly stronger for those injected with imsEV, possibly because of TfR1 expression by the brain capillary endothelial cells forming the BBB (Supplementary Fig. 21). In addition, a comprehensive analysis of total blood cell counts revealed no statistically significant changes in red blood cells, white blood cells, or lymphocytes across various preparations and the PBS-negative control in naïve mice (Supplementary Fig. 22). Moreover, no differences were observed in the quantities of distinct T cell subsets

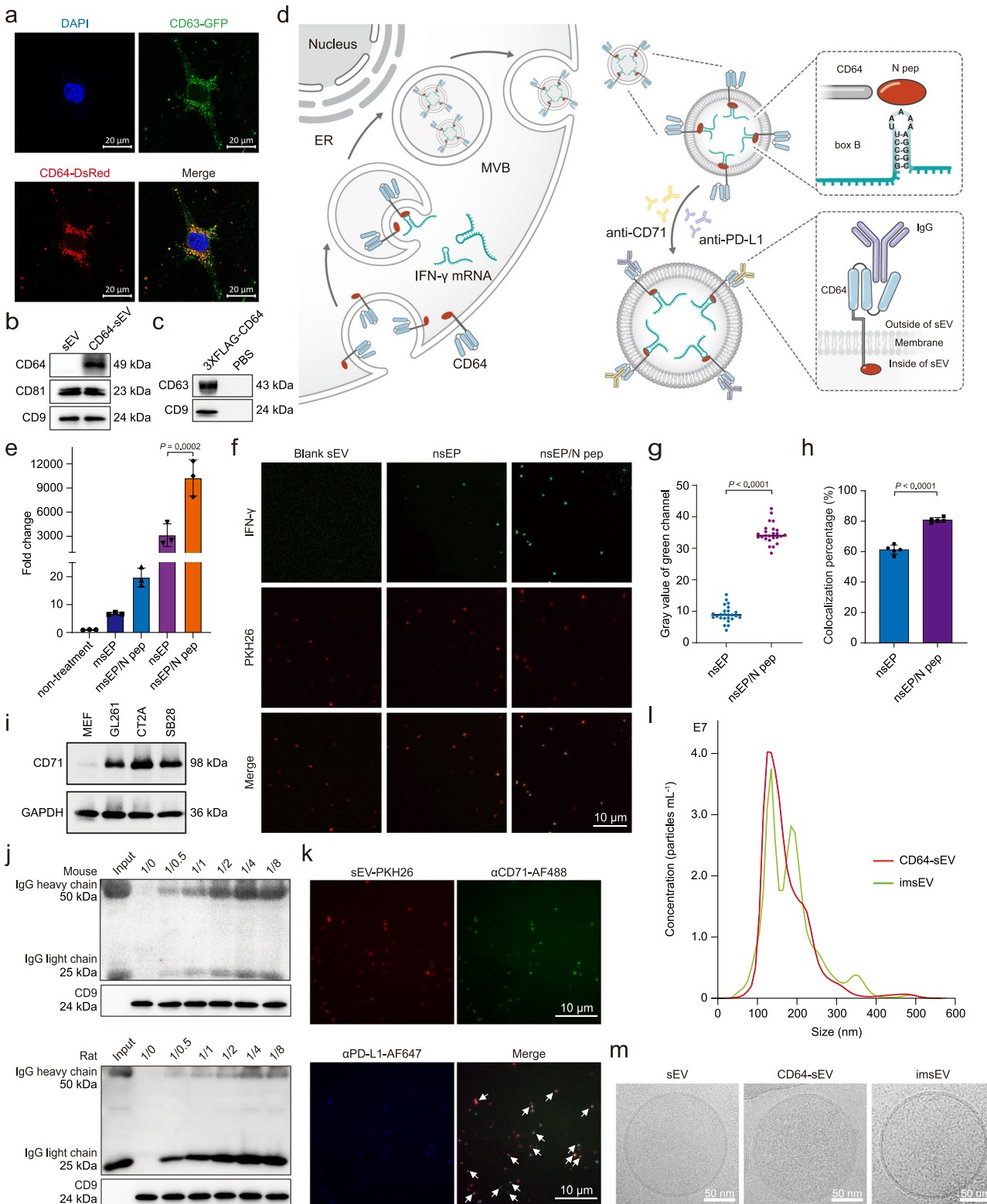

(i.e., CD4$^+$ and CD8$^+$) in the blood and spleen samples (Supplementary Figs. 23 and 24). Collectively, these findings indicate that these imsEVs have a favourable safety profile for in vivo administration.

### Therapeutic efficacy of imsEV in preclinical models

To investigate the immunotherapeutic potential of imsEV in vivo, we first injected imsEV intravenously at a dose of $5 \times 10^{11}$ sEVs into immune-competent mice implanted with GL261 tumours, which are moderately immunogenic[17]. Results from in vivo imaging system (IVIS) showed that the imsEV had significantly improved tumour targeting capability than non-targeted sEVs at 2 and 4 h after injection (Fig. 6a). Ex vivo evaluation of systemic biodistribution indicated a significantly higher accumulation of imsEV within tumours as compared with non-targeted sEVs, with a corresponding reduction in hepatic accumulation (Fig. 6b and c). Administering imsEV to the GL261 tumour-bearing mice every 3 days led to significant inhibition of tumour growth at 7 days after tumour implantation (Fig. 6d and e) and extension of survival, as evidenced

**Fig. 4 | Characterization of immunogenic sEVs (imsEVs). a** Confocal images of MEFs simultaneously transfected with CD64-DsRed and CD63-GFP indicate extensive colocalization of these two surface markers. **b** Western blot assessment of CD64 expression in natural sEV and sEV produced by CD64+ cells. **c** Western blots of an sEV pull-down assay show that FLAG beads could pull down the N-terminal FLAG of 3XFLAG-CD64. **d** Schematic representation of the attachment of IgG to the surface of sEVs through CD64 and the active RNA packaging strategy via the N peptide-box B affinity. **e** CD64-N peptides were co-transfected with box B-IFN-γ or control IFN-γ plasmids in MEFs, and the resulting imsEV were pelleted via ultracentrifugation. RT-qPCR was used to detect IFN-γ in imsEV prepared by the various methods, and U6 was used as the internal standard (n = 3 biologically independent samples). **f** Representative total internal reflection fluorescence (TIRF) images from a tethered lipoplex nanoparticle (TLN) assay. Scale bar: 10 μm. **g** IFN-γ mRNA fluorescence intensity within sEVs, as measured by the TLN assay in the different treatment groups (n = 25 biologically independent samples). **h** Colocalization percentage of IFN-γ mRNA with sEVs after transfecting with IFN-γ or IFN-γ-box B plasmid (n = 5 biologically independent samples). **i** Western blot assessment of CD71 expression in GBM cell lines and MEFs. **j** CD64-sEV were incubated with anti-PD-L1 antibody or anti-CD71 antibody for 4 h and then subjected to immunoprecipitation and western blot assay. **k** Representative TIRF images of the TLN assay showed that CD64-sEV could simultaneously adsorb two different IgGs in a single sEV. Arrow: sEVs with both antibodies adsorbed. Scale bar: 10 μm. **l** The sEV size distribution was measured by NS300 after incubating CD64-sEV with IgG. Red, CD64-sEV; Green, imsEV. **m** Cryo-EM images of sEV, CD64-sEV, and imsEV. Experiments **a**–**c**, **f**, **i**, **j**, **k**, and **m** were repeated three times independently with similar results obtained. Data represent means ± SD, analysed by one-way analysis of variance with Tukey's multiple comparisons test (**e**) or by unpaired two-tailed Student's t-tests (**g** and **h**). Source data are provided as a Source Date file.

by a median survival time of 53 days for the imsEV-treated mice versus 35 days for the antibody combo-treated mice and 29 days for the control groups (Fig. 6f). Immunoassays of residual tumour tissue revealed that IFN-γ protein expression was increased after imsEV treatment (Fig. 6g). MHC-I levels were also upregulated after imsEV treatment relative to the other treatment conditions (Fig. 6h). Upon restoration of MHC-I expression in GBM, the proportion of CD8+ cells in tumour tissues from imsEV-treated mice also increased (Fig. 6i, j, and Supplementary Fig. 25a), and the upregulation of IFN-γ was associated with increased proportions of M1-type macrophages at the tumour site (Fig. 6k and Supplementary Fig. 25b). Histological staining further showed that imsEV treatment greatly reduced tumour cell proliferation in the GBM tissue (Supplementary Fig. 26). We did not observe toxicity effects in major organs including heart, liver, spleen, lung, and kidney (Supplementary Fig. 27).

Finally, after validating the therapeutic potential of imsEV in the moderately immunogenic GL261 mouse model, we went on to investigate its antitumour effects in the orthotopic SB28 murine GBM model, which is poorly immunogenic owing to its low intrinsic MHC-I expression and is phenotypically similar to human GBM. Similar to our observation with GL261 model, imsEV accumulated in SB28 tumours to a greater extent than the non-targeted sEVs did, as evidenced by strong DiR fluorescence at 2 and 4 h in the imsEV-treated group (Fig. 7a). Ex vivo data further confirmed that more imsEV accumulated at the tumour and less at the liver relative to the non-targeting sEVs (Fig. 7b and c). Tumour growth was also drastically reduced after imsEV treatment (Fig. 7d and e), and survival time was extended (median survival time of 50 days in the imsEV-treated group versus 27 days in the PBS-treated group) (Fig. 7f). Magnetic resonance imaging further confirmed that the tumours were the smallest in animals from the imsEV-treated group as compared with the other treatment groups (Fig. 7g and h). Again, we noted an increase in IFN-γ and MHC-I expression in tumours after imsEV treatment (Fig. 7i, j, Supplementary Figs. 28 and 29). The proportions of CD8+ T cells and CD86+ macrophages that penetrated the GBM tumour sites after imsEV treatment were also greatly increased (Fig. 7k, l, Supplementary Figs. 30 and 31). Furthermore, we detected increased expression of Iba1 in tumours of imsEV-treated mice (Supplementary Fig. 32), which is considered evidence of CD8+ T cell-mediated adaptive immunity. Additionally, by blood cell counts and T cell immunity in both the blood and spleen of mice across different treatment groups, it was observed that imsEV partially mitigated the systemic immune suppression associated with GBM. This finding underscores the potential of imsEV as a therapeutic intervention for alleviating the immune suppression observed in GBM (Supplementary Figs. 33–35). Finally, histological staining results confirmed that imsEV treatment reduced the proliferation of SB28 tumour cells, but did not affect proliferation in the heart, liver, spleen, lung, and kidney (Fig. 7m, Supplementary Figs. 36 and 37).

## Discussion

In this study, we report a nsEP system with a microfluidic configuration that is capable of generating large quantities of sEVs that encapsulate mRNA probes. Applying millisecond and nanosecond pulses separately shifted the main impact of electroporation from the cell membrane to the membrane structure of cellular organelles. These effects have been confirmed in work involving irreversible electroporation for cancer treatment in vivo[26,27] and in previous studies of the effective delivery of exogenous probes into cells[28,29]. In the current study, we integrated this adequate stimulation strategy with parallel processing-capable microfluidics to leverage high-throughput sEV secretion and achieved an impressive enhancement of sEV yield (more than 30 times the yield from natural secretion). Additionally, the flow in the microfluidics device quickly sweeps any gas bubbles away from the electrode surface before they grow, ensuring that passing cells receive effective stimulation. It also improves the viability of source cells after electroporation, ensuring high sEV yield.

In addition to the large-scale production of sEVs, our method also facilitates enrichment of the doses of target RNA probes in the sEVs in two ways. First, combining electric pulses of different duration (i.e., nsEP and msEP) enhances the loading efficiency of plasmids as well as their expression kinetics[29]. In detail, the nanosecond pulses help to increase the permeability of the nuclear membrane of the treated cells and accelerate the transportation of plasmids to the nucleus and the overall transcription process. The second means of enriching target-mRNA doses in sEVs is by promoting the recruitment of the target mRNA (IFN-γ mRNA) by engineering a small box B sequence in the 3' end of the target mRNA and the N peptide on CD64, which is over-expressed on the membrane of host cells. We confirmed that the specific binding affinity between the box B and the N peptide on its amino-terminal arginine-rich domain could selectively enrich the target RNA probes in sEVs during their formation and leverage the average mRNA number in individual sEVs. Considering that the sEV population is similar in both cases (nsEP with and without N-peptide introduction), the increase in mRNA probes in sEVs produced by the nsEP-plus-N peptide approach is mainly attributable to having more than one mRNA per individual sEVs.

For a deeper understanding of the biological mechanisms underlying this nsEP-triggered sEV release, we used proteomics analysis, which implicated three proteins in the sEV secretion process: MLKL, ISG15, and Usp18. MLKL is known to be required for the effective generation of intraluminal and EVs[30]. We also verified that MLKL was pivotal in controlling sEV production after nsEP treatment, as MLKL deficiency led to reduced levels of sEV secretion, below the basal level of untreated cells. Others have found that an ISGylation modification of the multivesicular body protein TSG101 by ISG15 can facilitate its co-localization with lysosomes and promote their aggregation, thereby impairing sEV secretion and that this effect could be reversed by the Ub-specific protease Usp18[31]. The ISGylation targets of functional proteins in the secretion of sEVs are TSG101 and heat-shock proteins

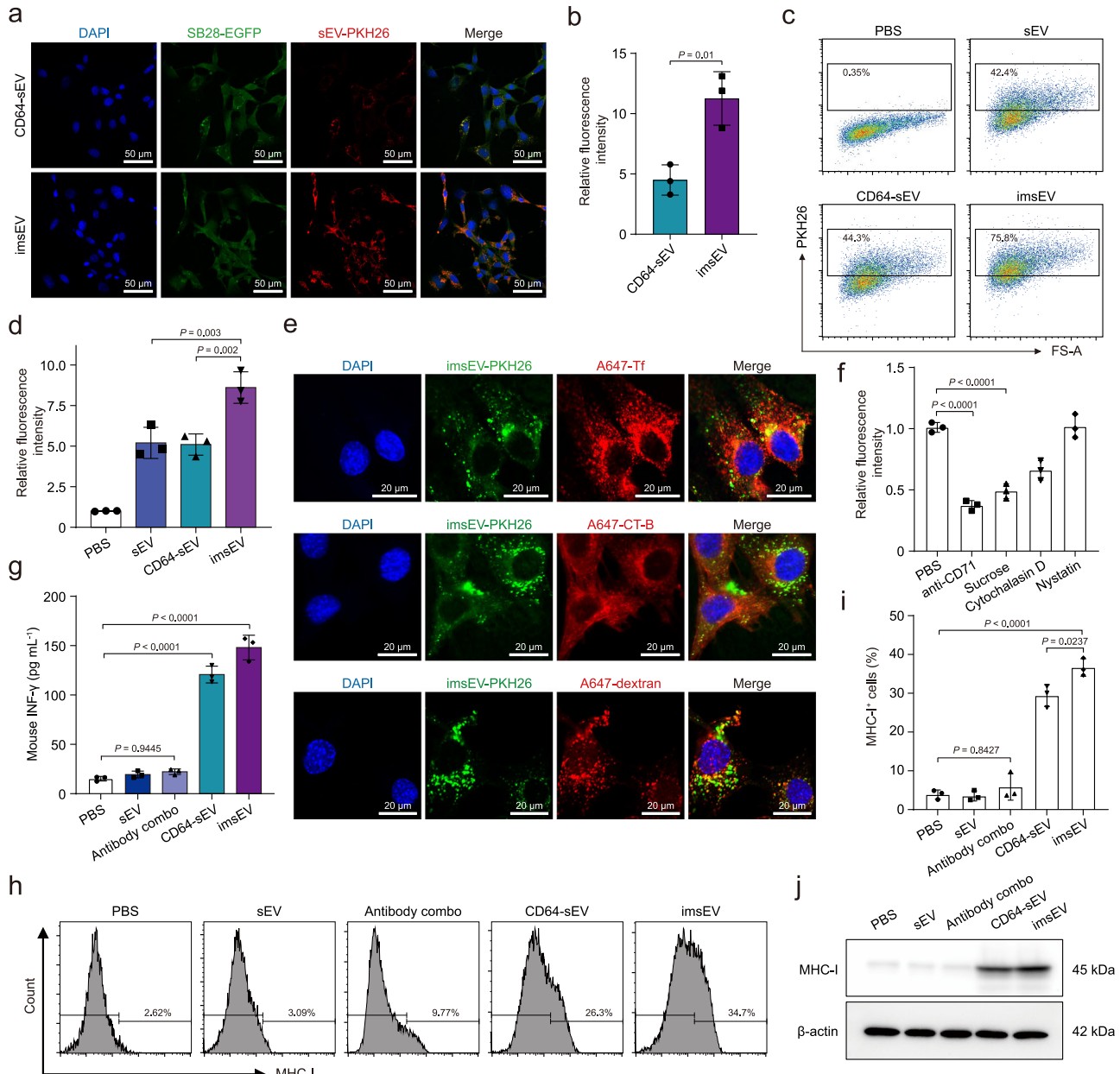

**Fig. 5 | In vitro study of imsEV for cancer therapy. a, b** Increased uptake of imsEV conjugated with anti-CD71 antibody by SB28 glioma cells. Scale bar: 50 μm. **c, d** Fluorescence intensity of PKH26-labelled sEV taken up by SB28 cells, measured by flow cytometry, confirming the effective uptake of imsEV by SB28 cells. **e** Representative immunostains show colocalization of imsEV labelled with PKH26 (i.e., those that fluoresce with $\lambda_{ex}$ 551 nm; $\lambda_{em}$ 567 nm; shown in green) and imsEV labelled with other endocytosis markers (red). Most imsEV were colocalized with transferrin-Alexa Fluor 647 (A647-Tf), suggesting that imsEV are mainly taken up through clathrin-dependent endocytosis. A647-Tf is a marker of clathrin-dependent endocytosis; cholera toxin subunit B-Alexa Fluor 647 (A647-CT-B) is a marker of caveolae-dependent endocytosis; and A647-dextran is a marker of macropinocytosis. Scale bar: 20 μm. **f** Fluorescence intensity of PKH26-labelled imsEV taken up by SB28 cells treated with different endocytosis inhibitors, assessed by flow cytometry, further confirmed that imsEVs are primarily taken up through

clathrin-dependent endocytosis. Sucrose, a clathrin-dependent endocytosis inhibitor; Nystatin, a caveolae-dependent endocytosis inhibitor; and Cytochalasin D, a macropinocytosis inhibitor. **g** Amounts of IFN-γ in the supernatant of SB28 cell culture medium after treatment with PBS, antibody combo (anti-PD-L1 & anti-CD71), CD64-sEV, or imsEV for 48 h and then measured by ELISA. **h, i** Expression of IFN-γ and MHC-I in SB28 cells by flow cytometry after the indicated treatments. **j** Western blot assessment of MHC-I expression in SB28 cells after treatment with PBS, antibody combo, CD64-sEV, or imsEV. The experiments **a, c, e, h,** and **j** were repeated three times independently with similar results obtained. Data in **b, d, f, g,** and **i** represent means ± SD *n* = 3 biologically independent samples; analysed by one-way analysis of variance with Tukey's multiple comparisons tests (**d, f, g,** and **i**) or by unpaired two-tailed Student's *t*-tests (**b**). Source data are provided as a Source Date file.

(HSPs)[32,33]. Interestingly, although our proteomics profiling revealed ISG15 and Usp18 as top candidates in the sEV secretion process (nsEP led to a 189-fold increase in ISG15 and an 81-fold increase in Usp18), most downstream functional proteins of ISG15/Usp18 signalling, including TSG101 and HSP90, were not significantly changed.

Therefore, we excluded ISG15/Usp18 as being the main factors for promoting sEV trafficking during nsEP. This differs from our previous findings on sEV secretion after cellular nanoporation (CNP), in which HSP90 and HSP70 were found to be critical for electroporation-stimulated sEV production: inhibiting both greatly reduced the

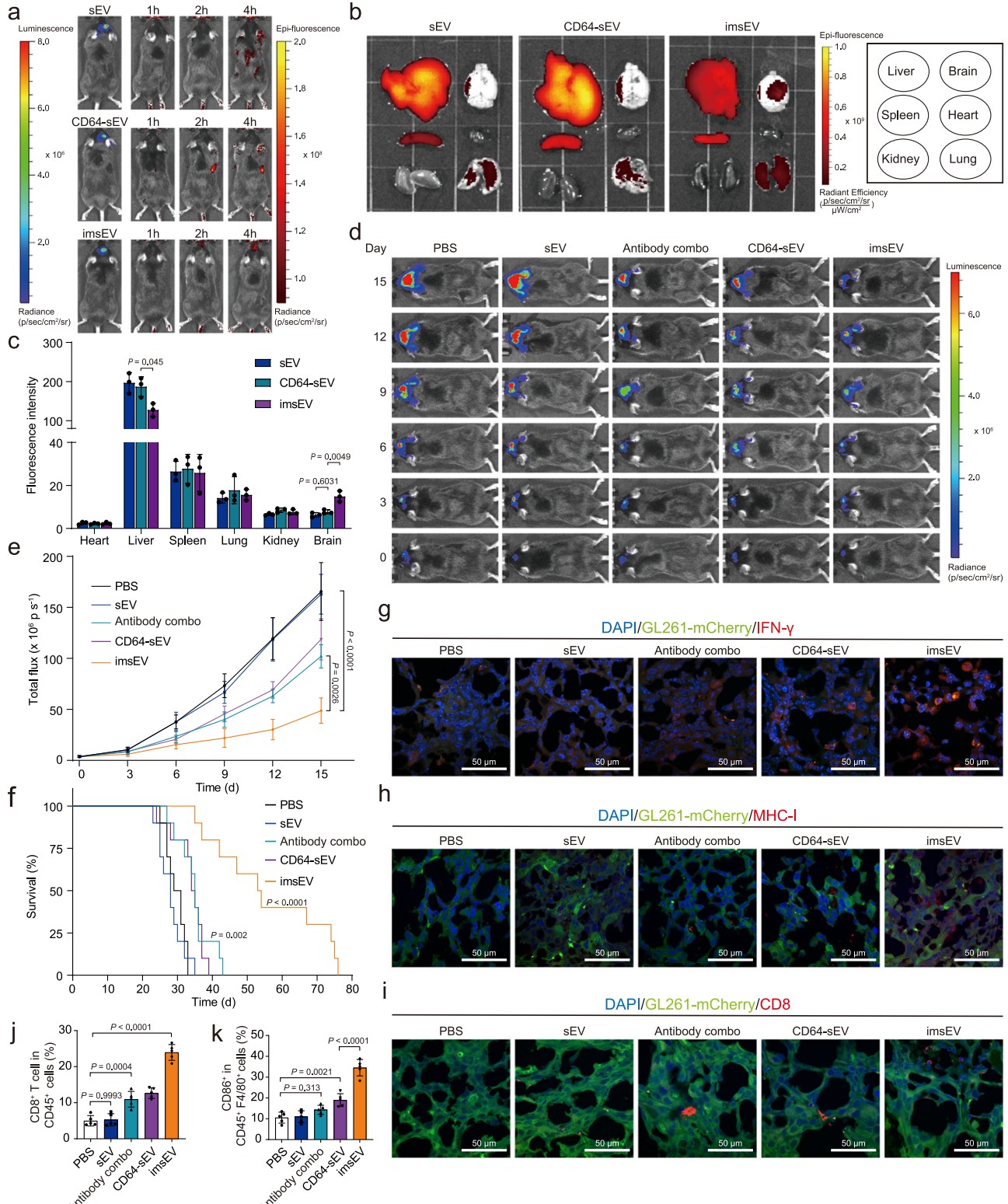

numbers of sEVs produced after CNP[2]. One possible explanation for this difference is the formation of a transient, localized heat shock to the cell membrane close to the nanopore during CNP, but not during nsEP. Hence even though an electroporation step is involved in both techniques, the major mechanisms underlying the enhancement of sEV secretion are different, although they may share some similarities. More detailed investigations may shed light on the molecular mechanisms underlying the biogenesis of sEVs and cargo sorting resulting from electroporation stimulation.

Because a natural receptor for the Fc domain on IgG is anchored on the external surface (on the N-terminal of CD64), the sEVs produced by our approach could be used to selectively target other cell types simply by changing the antibodies. In this work, we investigated the potential of these imsEVs for immunotherapy in GBM. Although the success of checkpoint blockade has generated considerable enthusiasm for immunotherapy in general, immunotherapy for GBM has not been successful clinically[34]. GBM effectively evades immune surveillance, in part through downregulating MHC-I in GBM cells[35]. Exposing

**Fig. 6 | In vivo therapeutic efficacy of imsEV in an orthotopic GL261 glioma model. a** In vivo imaging by IVIS showed preferential accumulation of DiR-labelled imsEV within orthotopically implanted GL261 tumours in mice. **b, c** Tissue distribution analyses indicated that imsEV showed increased brain targeting with low hepatic accumulation (n = 3, biologically independent samples). **d, e** Tumour growth inhibition by tail-vein injection of PBS, empty sEVs (sEVs), antibody combo (anti-PD-L1 and anti-CD71), CD64-sEV, and IFN-γ-mRNA containing sEVs plus antibodies (imsEV) (n = 5, biologically independent samples). **f** imsEV treatment extended the survival of mice with GL261 glioma. (n = 10, biologically independent samples). **g-i** IFN-γ, MHC-I, and CD8 staining of residual GBM tumour tissue in the indicated treatment groups showed that imsEV increased the expression of IFN-γ and MHC-I and increased the proportion of CD8+ cells in tumour tissues. Scale bar: 50 μm. The experiments (**g-i**) were repeated three times independently with similar results obtained. **j** Flow cytometry assessment of the proportions of CD8+ cells in tumour tissues of mice in the indicated treatment groups (n = 5, biologically independent samples). **k** Quantitative analysis of macrophages (gated on F4/80 cells) in the indicated treatment groups (n = 5, biologically independent samples). Data represent means ± SD (**c, e, j,** and **k**); analysed by one-way analysis of variance with Tukey's multiple comparisons tests (**c, e, j,** and **k**), or by log-rank (Mantel–Cox) tests (**f**). Source data are provided as a Source Date file.

GBM cells to IFN-γ is thought to restore MHC-I expression on their surfaces[36]. IFN-γ has antitumour effects by modulating the functions of tumour cells, immune cells, and other cells in the tumour micro-environment (TME)[37], and effective immunotherapy seems to require abundant and constant secretion of IFN-γ into the TME[38–40]. However, delivery of soluble IFN-γ has a wide range of side effects that depend on dose, route of administration, and frequency[41–43]. The US FDA has approved the use of the recombinant protein IFN-γ1b, given as sub-cutaneous injections, to reduce the risk of sEV side effects[44]. Moreover, IFN-γ is known to have a short half-life, which necessitates frequent dosing or continuous infusion to sustain therapeutic efficacy. Thus far, IFN-γ1b has shown disappointing results in the clinic because of the short half-life of the IFN-γ protein and the toxicity associated with frequent dosing[45,46]. Limited tumour targeting is another significant clinical challenge for the clinical use of cytokine immune checkpoint blockade, which in the case of IFN-γ is limited because of the wide-spread expression of IFN-γ receptor[46]. The nonspecific distribution of IFN-γ can also result in off-target effects and potentially limit its ther-apeutic efficacy. For these reasons, we explored ways of introducing the IFN-γ gene into the targeted tumour or immune cells by encap-sulating the mRNA for IFN-γ in carriers to result in localized and con-stant production of IFN-γ.

Various carriers such as adenovirus[47], oncolytic viruses[48], and liposomes[41] have been used to load the gene that encodes IFN-γ and allow cytokine release in the TME; some of these carriers have had beneficial antitumour effects in vitro[48,49]. However, those studies were not designed specifically for GBM therapy. To date, adequate and constant IFN-γ expression in the TME within the brain has not been confirmed in trials of oncolytic virotherapy[50]. One potential challenge for such studies is that, unlike sEVs, only specific groups of viruses can cross the BBB[50]. Second, encapsulating large molecules (e.g., mRNA) into viruses that can cross the BBB is difficult because of their limited capacity (e.g., 4.5 kb for AAV rh10, parvovirus)[50,51]. Moreover, although one study found that an inserted peptide could increase the infectivity of glioma cells, most virus carriers result in untargeted viral replica-tion, whereas sEVs demonstrate flexible surface functionalization capability to target specific cells[52,53]. Hence, sEVs present satisfying gene encapsulation capacity, with easy surface modification for tar-geting, and excellent biocompatibility as an IFN-γ carrier for GBM immunotherapy. Unlike DNA-based drugs, mRNA does not carry a risk of accidental infection or opportunistic insertional mutagenesis, as it does not need to enter the nucleus to be functional[54,55]. An intrinsic advantage of mRNA-based immunotherapy lies in the fact that small amounts of loading are adequate to provide vigorous efficacy signals[56]. Also, the abundance of positive safety and efficacy data obtained from the SARS-CoV-2 mRNA vaccines, together with the approval and reg-ulation of such vaccines by the US FDA, underscores the broad ther-apeutic potential of mRNA therapy, including cancer immunotherapy[57–59]. For all of these reasons, we chose to encapsulate mRNA rather than other IFN-γ encoding drugs for effective immunotherapy.

In our current study, we verified that our imsEVs successfully bound both anti-CD71 and anti-PD-L1. We also found that our GBM cell-targeted imsEV, delivering IFN-γ mRNA and PD-L1 antibody, could reprogramme the immune microenvironment of the tumour from an immunosuppressive to an immune-stimulating phenotype. Evidence of this reprogramming included the increased infiltration of effector immune cells, upregulation of MHC-I on cancer cells, and polarization of suppressive myeloid cells to an activating phenotype. These chan-ges inhibited tumour growth and extended survival in preclinical GBM models, including models that are intrinsically immune-resistant. Correspondingly, our surface-functionalized, non-toxic, low-immunogenic sEVs allowed specific interactions with targeted cells[60], protected IFN-γ from endonucleases, and prevented its detection by the immune system, leading to targeted delivery to cells of interest, efficient entry into those cells, and potency with few severe side effects[61]. Collectively, our findings demonstrate that an adaptive design strategy that efficiently produces mRNA-loaded sEVs with tar-geting functionalities could pave the way for their adoption in cancer immunotherapy applications, opening up avenues for improving the responsiveness of immune-resistant tumours. Nevertheless, to meet manufacturing practice requirements and secure regulatory approval for clinical dosages, further improvements, including production, stability, quality control and safety assessments are still needed. For modified sEVs to be deemed suitable for human use and to mitigate potential risks such as potency loss, stringent control over immuno-genicity is paramount, particularly for interventions involving repe-ated administrations. Encouragingly, modified sEVs derived from the HEK293T cell line have been shown to possess minimal immunogeni-city in mice after repeated doses[62], and modified sEVs sourced from stem cells, such as mesenchymal stem cells (MSCs)[63], are expected to lack immunogenicity, given that MSCs are known for their low immunogenic potential. However, there is no solid evidence to prove whether modified sEV cargos have low immunogenic activities in human recipients at the moment. Rigorous preclinical studies and adherence to regulatory guidelines are imperative before applying modified sEVs in human clinical settings.

## Methods

### Ethics statement

This research complies with all relevant ethical regulations. All experimental procedures were performed in compliance with the institutional policies and approved protocols of Jilin University (no. SY202110005) or MD Anderson Cancer Center (no. 00002163).

### Cell culture

The SB28 and GL261 cell lines were purchased from the German Col-lection of Microorganisms and Cell Cultures GmbH (DSMZ) with cat-alogue numbers ACC 880 and ACC 802. The HEK293T, MEF, U138, T98G, and LN-229 cell lines were purchased from the American Type Culture Collection (ATCC, CRL-3216, SCRC-1040, HTB-16, CRL-1690, and CRL-2611). The CT-2A, U87, and U251 cell lines were purchased from Millipore Sigma Aldrich with catalogue numbers SCC194, 89081402-1VL, and 09063001-1VL. HEK293T, MEF, GL261, LN229, and CT-2A cells were cultured in Dulbecco's modified essential medium (DMEM) containing 10% heat-inactivated foetal bovine serum (FBS, Thermo Fisher Scientific, 26140095; Exosome-depleted FBS, Thermo Fisher Scientific, A2720801) supplemented with 1% penicillin/

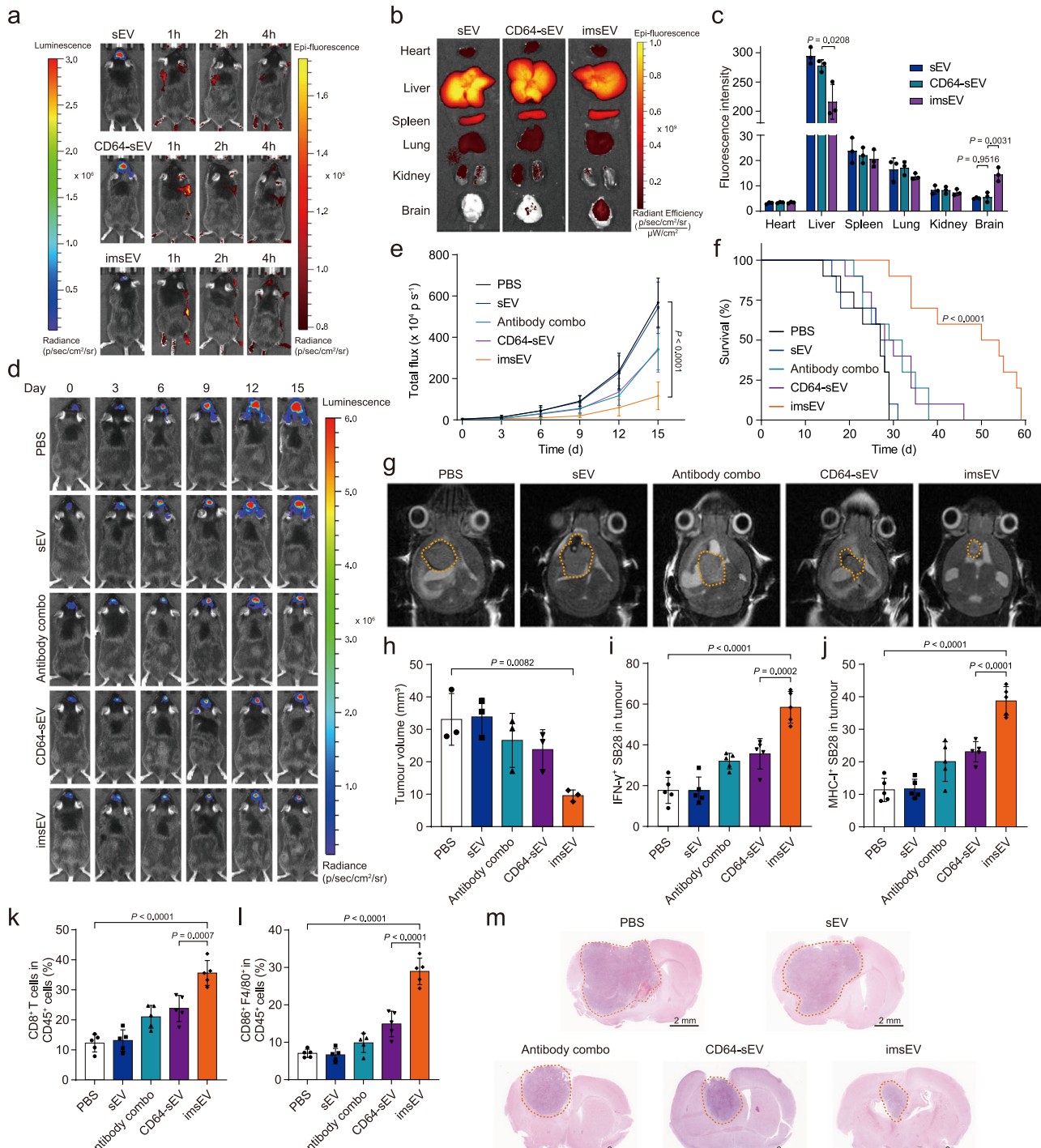

**Fig. 7 | In vivo therapeutic efficacy of imsEV in an orthotopic SB28 glioma model. a** In vivo imaging showed preferential accumulation of DiR-labelled imsEV within orthotopically implanted SB28 tumours in mice. **b, c** Tissue distribution analyses indicated that imsEV showed increased brain targeting and low hepatic accumulation ($n = 3$, biologically independent samples). **d, e** Tumour growth inhibition after tail-vein injection of PBS, empty sEV, antibody combo (anti-PD-L1 & anti-CD71), CD64-sEV, and IFN-γ mRNA-containing sEV with antibodies (imsEV) ($n = 5$, biologically independent samples). **f** imsEV extended the survival of mice with SB28 glioma. ($n = 10$, biologically independent samples). **g, h** Tumour size assessed by magnetic resonance imaging after the final treatment ($n = 3$, biologically independent samples). **i, j** IFN-γ and MHC-I staining in residual GBM tumour tissue from the

indicated treatment groups showed that imsEV increased the expression of IFN-γ and MHC-I ($n = 5$, biologically independent samples). **k, l** Quantitative analysis of T cells and M1 type macrophages in SB28 tumours from the indicated treatment groups analysed by flow cytometry showed that imsEV led to increased proportions of CD8[+] T cells and M1 type macrophages ($n = 5$, biologically independent samples). **m**, H&E stain of residual SB28 tumour tissue after the indicated treatments shown that imsEV inhibited cell proliferation in tumour tissue. Scale bar: 2 mm. Data represent means ± SD; analysed by one-way analysis of variance with Tukey's multiple comparisons tests (**c, e, h, i, j, k,** and **l**), or by log-rank (Mantel−Cox) tests (**f**). Source data are provided as a Source Date file.

streptomycin at 37 °C in a humidified condition equilibrated with 5% $CO_2$. SB28 cells were cultured in DMEM with 10% FBS supplemented with 1% penicillin−streptomycin at 37 °C and 10% $CO_2$. U87, U251, U138, and T98G cells were cultured in Eagle's minimum essential medium (EMEM) with 10% FBS supplemented with 1% penicillin−streptomycin at 37 °C and 5% $CO_2$.

## Plasmid preparation

Mouse CD64, mouse IFN-γ, mouse MLKL, mouse MLKL shRNA, human CD64, and human IFN-γ plasmids were purchased from Origene (MC208752, SC300109, MC206757, TR513478, RC207487, RC209993). Primers designed to encode N pep (MDAQTRRRERRAEKQAQWKAAN) were used to introduce the ligands into the C terminus of CD64. In the same way, box B (CGGGAAAAAGUCCCG) was introduced into the 3' end of IFN-γ.

## Microfluidic nanopulse channel device fabrication

A microfluidic device for nanopulse electroporation was fabricated by using a computer numeric control machine. One platinum wire of 50 μm in diameter was first embedded in a polymethyl methacrylate (PMMA) block by hot embossing. The platinum wire was then cut into two pieces when a microchannel (width × depth × length = 300 μm × 100 μm × 5 cm) was micro-milled perpendicular to the direction in which the wire is positioned on the PMMA block. These two platinum wires serve as the electrodes and are connected to the nanopulse circuit during cell stimulation. The flow rate of cell solution in the microfluidic device was regulated by a syringe pump (KDS 100 Legacy Syringe Pump).

## Nanosecond pulse electroporation

A self-built electroporation circuit was designed to generate electrical pulses with both high-voltage and tunable duration of nanosecond pulses. To avoid signal entanglement and pulse profile distortion, the nanosecond pulse generation circuit is separated from the high-voltage supply while connected with a radio frequency metal−oxide−semiconductor field-effect transistor (MOSFET). During the operation, the rectangular signal from the pulse waveform generator (Agilent 33220A) periodically triggers the closure of the electroporation circuit through the MOSFET switch when overcoming its threshold gate voltage. A power supply (KIKUSUI PMC250-0.25A) is used to provide the desired level of energy output by pre-charging a capacitor that stands by until the electroporation circuit is closed by the MOSFET switch to allocate high-voltage pulses on cells with nanosecond pulses while the pulse width, frequency, and number decided by the pulse generator. An oscilloscope was connected to monitor the actual profile of the nanosecond pulses.

## Cell transfection

For nanosecond electroporation (nsEP), MEF or HEK293T cells were digested, centrifuged at 1000×*g* for 10 min, and re-suspended in fresh serum-free OPTI-MEM medium at a density of $6 \times 10^7$ cells mL⁻¹. DNA plasmids were then mixed with the electroporated sample (100 μg mL⁻¹), which was passed through the microfluidic device and its integrated platinum electrode at a speed of 10 mL h⁻¹. The treated cell solutions were collected in a traditional electroporation cuvette downstream (with the parallel electrodes separated by 4 mm) and received immediately standard millisecond electroporation, according to the manufacturer's instructions (BTX Harvard Apparatus ECM630 Electro Cell Manipulator Generator). After electroporation, cells were transferred and further cultured in a fresh exosome-free medium prior to sEV harvesting or further analysis.

## Plasmid loading and mRNA transcription analysis

Copies of plasmids loaded in cells and subsequently transcribed mRNA in the transfected cells were estimated as follows (see flowchart in Supplementary Fig. 38). Briefly, $2 \times 10^5$ cells were first transfected with 2 μg plasmids by electroporation and divided into two separate groups for further culturing. After cells were re-attached on the culturing surface (~3 h later), half of the transfected cells were washed with fresh medium to ensure that all plasmids extracted later were those already inside cells. INF-γ plasmids were extracted from cells by using DNeasy Blood & Tissue Kits (QIAGEN) in accordance with the manufacturer's instructions. Copies of plasmids in the collected cells of this group were then determined by qPCR. The average plasmid copies per cell were then calculated by dividing the number of cells in each group (estimated by cell viability testing). The average copies of mRNA transcribed from the loaded plasmids were estimated similarly from the other half of the transfected cells (that had been collected 6 h after plasmid delivery). More experimental details are given in the legend for Supplementary Fig. 38.

## Collection and purification of sEVs

The sEVs were collected and purified by ultracentrifugation[2]. Briefly, before cells were transfected, the serum-containing cell culture medium was removed. After nsEP, the cells were cultured in an exosome-free culture medium for 48 h. Then, the cell-culture supernatants were centrifuged at 2000×*g* for 10 min to remove debris, and large vesicles and apoptotic bodies were removed by centrifugation at 10,000×*g* for 30 min. The final sEV fraction was then purified after ultracentrifugation at 100,000×*g* for 2 h. To prepare imsEVs, CD64-sEVs were incubated with anti-CD71 mAb (Bio X Cell) and anti-PD-L1 mAb (Bio X Cell) (1:1:3, w/w/w, CD64-sEV by protein mass) for 2 h at 37 °C. Subsequently, free antibodies were removed by ultracentrifugation at 100,000×*g* for 2 h.

## sEV number and size measurements

Absolute numbers and size distributions of sEVs were determined with a NanoSight NS300 device (Malvern, PA, USA).

## Cryogenic transmission electron microscopy (cryo-EM)

Cryo-EM was used to characterize purified sEVs from MEFs. A concentration of $10^{11}$ sEVs mL⁻¹ was necessary for this experiment. Sample preparation and data acquisition were performed by the Cryo-EM Core Facility at UTHealth Houston. A small aliquot (3 μL) of the sample was applied to the Quantifoil R2/1 Cu 200 specimen grid (Electron Microscopy Sciences). Glow discharge of the grid was operated with PELCO easiGlow (Ted Pella). Acquisitions were obtained with a Titan Krios microscope and data were acquired with EPU software (Thermo Fisher Scientific). Images were recorded on a K2 Summit direct electron detector (Gatan) operated in super-resolution counting mode.

## RT−qPCR of exosomal RNA expression levels

The expression of IFN-γ mRNA in sEVs was detected by RT−qPCR according to the manufacturer's instructions. Briefly, total RNA was isolated from sEVs by using TRIzol (Invitrogen) and was reverse-transcribed into cDNA with a Reverse Transcription Kit (Thermo Fisher Scientific). Gene expression was measured by using the SYBR Green qPCR kit (BioRad). Expression values were normalized to that of U6. Gene-specific primers included U6 forward (5′-CTCGCTTCGGCAG-CACA-3′), U6 reverse (5′-AACGCTTCACGAATTTGCGT-3′), *IFNG* (human) forward (5′-ACAGCAAGGCGAAAAAGGATG-3′), *IFNG* (human) reverse (5′-TGGTGGACCACTCGGATGA-3′), *Ifng* (mouse) forward (5′-CAGCAAC AGCAAGGCGAAAAAGG-3′), and *Ifng* (mouse) reverse (5′-TTTCCG CTTCCTGAGGCTGGAT-3′).

The absolute copy number of target mRNA in sEVs was also determined by qPCR results. The average number of target mRNAs per sEV was calculated by dividing by the sEV number measured using NS300. Briefly, the isolated RNA was first reverse-transcribed into complementary DNA (cDNA) by using the TaqMan™ reverse transcription kit (Life Technologies, Carlsbad, CA), following the

manufacturer's protocol. The subsequent quantitative polymerase chain reaction (qPCR) analysis was done in triplicate with 100 ng of DNA in a 20 μL reaction volume. Each 20 μL reaction contained 10 μL of TaqManTM Fast Advanced Master Mix, 1 μL of the Gene copy number assay (TaqMan™ *Ifng* Gene copy number assay Mm00734344_cn), and 9 μL of the DNA template. The qPCR conditions included an initial denaturation step at 50 °C for 2 min, followed by a 10 min step at 95 °C. Subsequently, a total of 40 cycles were performed, consisting of denaturation at 95 °C for 15 s, followed by annealing and extension at 60 °C for 1 min.

## Proteomics analysis

MEFs were treated with nsEP, harvested, digested overnight with trypsin at 37 °C, and incubated with DTT and iodoacetamide to reduce and alkylate proteins[64]. Samples were then subjected to solid-phase extraction cleanup with an Oasis HLB plate (Waters), and the resulting samples were loaded onto an EasySpray column (75 μm particles, 750 mm length) to analyse with an Orbitrap Fusion Lumos mass spectrometer coupled to an Ultimate 3000 RSLC-Nano liquid chromatography system. The gradient consisted of an increase from 1% to 28% solvent B (80% acetonitrile, 10% trifluoroethanol, and 0.1% formic acid in water) over 90 min; solvent A contained 2% acetonitrile and 0.1% formic acid in water. MS scans were acquired at 120,000× resolution in the Orbitrap, and up to 10 MS/MS spectra were obtained in the ion trap for each full spectrum acquired by higher-energy collisional dissociation for ions with charges. Dynamic exclusion was set for 25 s after an ion was selected for fragmentation. For enrichment analysis of proteins involved in sEV secretion and protein-protein interaction network analysis, the Gene Ontology, Kyoto Encyclopedia of Genes and Genomes, and STRING databases were applied, and the protein network data were visualized with Cytoscape software (v.3.7.2).

## Preparation of tethered lipoplex nanoparticles (TLN) biochips

The TLN biochips for the exosomal IFN-γ mRNA detection were fabricated as follows. An Au layer (-15 nm thick) was coated onto glass coverslips by using a Denton e-beam evaporator (DV-502A, Moorsetown, NJ), and the freshly coated coverslips were incubated overnight in an ethanol solution containing the lipidic anchor molecule WC14 (20-tetradecyloxy-3, 6, 7, 12, 15, 18, 22-heptaoxahexa-tricontane-1-thiol), the lateral spacer β-mercapto-ethanol, and biotin-SH at a molar ratio of 30:70:1. After incubation, the coverslips were washed carefully with 100% ethanol three times and air-dried. A polydimethylsiloxane chip (24 wells; 3 mm diameter, 4 mm thick) was anchored to the glass side with the Au coating, 20 μL neutravidin ethanol solution (100 μg μL⁻¹) was added to each well, and the chip was incubated for 15 min at room temperature. During incubation, the lipoplex nanoparticles containing molecular beacons against IFN-γ mRNA were freshly prepared as described below, for immediate use thereafter. First 9.75 μL of IFN-γ molecular beacon stock solution (at a concentration of 100 μM) was mixed with lipid in ethanol solution (29.5 μL, DOTMA:Cholesterol:Biotin-PEG6-SH = molar ratio of 49:49:2) and then quickly injected into 675 μL PBS and vortexed for 10 s. After the untethered neutravidin was removed from the wells with cold PBS, the freshly prepared lipoplex nanoparticles were added to each well and incubated for 15 min at room temperature. Untethered nanoparticles were washed away with cold PBS.

## sEV imaging with total internal refractory (TIRF) microscopy and flow cytometry

Purified engineered-sEVs from MEF cells were stained with PKH26, following a tangential flow filtration method to extensively remove the dye residual[65]. Later, the collected sEVs were added to the TLN biochip functionalized with IFN-γ molecular beacon and incubated at 37 °C for 2 h. Captured sEVs were subsequently incubated with anti-CD71 and anti-PD-L1 at room temperature for 1 h. Excessive staining agents were washed away before imaging[65–67]. For TLN biochips, sEVs were imaged with a TIRF microscopy. Images were recorded by an Andor iXon EMCCD camera with a ×100 oil lens, and the exposure time was set at 200 ms. The same staining protocol was used for flow cytometry. sEVs were imaged using Cytek@Aurora (CYTEK).

## Image analysis and colocalization

An automatic algorithm was used to quantify detected bright spots present on the TIRF microscopy images. The grey value is the sum of the intensities of all the pixels within the calculated spot area. The open-source plugin EzColocalization was applied with ImageJ to calculate the colocalization efficiency of sEVs stained with different biomarkers acquired from the TIRF microscopy images[66,68].

## sEV pull-down assay

Protein A magnetic beads (BioRad) were incubated with 5% (w/w) bovine serum albumin in PBS overnight at 4 °C, after which the beads were washed three times with cold PBS. FLAG antibody (Sigma-Aldrich) was then added to the magnetic beads and the mixture was incubated overnight at 4 °C and washed three times with PBS. The purified sEVs were incubated with the magnetic beads overnight at 4 °C and washed. The beads were then eluted in 0.1% sodium dodecyl sulfate (SDS), and 20 μL of the supernatant was loaded onto SDS gels for SDS−polyacrylamide gel electrophoresis (PAGE) analysis.

## Confocal microscopy

Transferrin-Alexa Fluor 647 (0.1 mg mL⁻¹), cholera toxin subunit B−Alexa Fluor 647 conjugates (0.005 mg mL⁻¹), and dextran-Alexa Fluor 680 (1 mg mL⁻¹) (Invitrogen) were each incubated with SB28 cells or GL261 cells for 1 h to label different endonucleases and then washed away. Cells were then incubated with sEVs stained with PKH26 (2 μM) (Sigma-Aldrich) for 4 h, washed twice with PBS, and fixed with 4% formaldehyde in PBS for 15 min. Nuclei were stained with 4′,6-diamidino-2-phenylindole in the gold coating solution, and fluorescence was observed and recorded on a laser scanning confocal microscope (LSM880, Carl Zeiss).

## MTS assay

The cytotoxic potential of sEVs was assessed with an MTS assay. SB28 and GL261 cells were plated in 96-well plates (5000 cells per well) and incubated overnight. Cells were then incubated with sEVs for 24 or 48 h, followed by the addition of MTS reagent (Promega) according to the manufacturer's instructions, and absorbance was measured at 490 nm wavelength after an additional 4 h of incubation.

## Western blots and antibodies

Protein samples were homogenized in RIPA lysis buffer (Thermo Fisher Scientific) with a 1% proteinase inhibitor cocktail (Thermo Fisher Scientific, no. 78429), and the protein lysates were normalized with a BCA protein assay kit (Thermo Fisher Scientific). Protein lysates were separated on 8%/10%/12% acrylamide gels and transferred onto polyvinylidene fluoride (PVDF) membranes (Millipore, USA) under denaturing conditions. The membranes were then blocked for 1 h with 5% non-fat milk in Tris-buffered saline solution at room temperature and incubated overnight at 4 °C with primary antibodies (Abcam, anti-MLKL, ab243142; anti-CD71, ab84036; anti-MHC-I, ab281902; anti-Thrombospondin 1, ab267388; Thermo Fisher Scientific, anti-CD63, MA5-35208; anti-CD9, MA5-31980; Cell Signaling Technology, anti-GAPDH, 2118; anti-β-actin, 4967), after which the membranes were washed three times and then incubated with a secondary antibody (Cell Signaling Technology) conjugated to horseradish peroxidase for 45 min at room temperature. The PVDF membranes were developed by using a chemiluminescence detection system. Full scan blots are in the Source Data file.

## Flow cytometry analysis

Cellular uptake of sEVs and assessment of cell surface and internal antigens were analysed by flow cytometry. To investigate sEV uptake by tumour cells, PKH26-labelled sEVs were incubated with tumour cells for 4 h, after which cells were washed three times with cold PBS and fixed in 4% paraformaldehyde. To assess cell surface and internal antigens, tumour tissues obtained at Day15 were isolated after transcardial perfusion from each treatment group were collected and digested at 37 °C for 60 min in 10 mmol L$^{-1}$ HEPES buffer with 300 U mL$^{-1}$ collagenase D, dispase, and 15 U mL$^{-1}$ DNase I to obtain cell suspensions. After dissociation, the cells were filtered through a 70 μm nylon cell strainer and collected. For flow cytometry, cells were fixed and permeated to allow the entry of fluorescence probes. To avoid nonspecific binding to the Fc receptor, cells were first blocked with an anti-CD16/CD32 antibody (Bio X Cell, catalogue no. BE0307, dilution of 1:200) for 15 min. Then, cells were incubated with various labelled antibodies (anti-IFN-γ-PE at a 1:200; anti-MHC-I-APC at 1:300; anti-CD8a-PE at 1:200; anti-CD86-PE at 1:300; anti-CD45-PerCP at 1:200; anti-CD3-APC at 1:300; or anti-F4/80-APC at 1:300 dilution) according to the manufacturer's instructions. Cell fluorescence intensity was analysed with a flow cytometer (Gallios 561, Beckman). At least 10,000 events were collected per cell sample. Representative gating strategies for all flow cytometry data are shown in Supplementary Fig. 39.

## Enzyme-linked immunosorbent assay

Cytokines in cell culture media were measured by ELISA as follows. Tumour cells were incubated with sEVs for 24 or 48 h, and INF-γ (BioLegend) levels were measured in the culture medium.

Levels of aspartate aminotransferase (Abcam), alanine aminotransferase (Abcam), blood urea nitrogen (Abcam), and creatinine (Thermo Fisher Scientific) in serum after 4 h for the systemic administration of sEVs were also tested using ELISA follow manufacturer's protocol for the biosafety measurements.

## Blood safety assessment

Prior to in vivo application, the sEV safety assessment was conducted. The haemolysis assay and complete blood counts (CBCs) were investigated. For the haemolysis assay, the whole blood from healthy female mice was collected in an anticoagulant solution tube and centrifuged at RT for 15 min (900×$g$) to get the RBCs. The harvested RBCs were then mildly rinsed with PBS. The RBCs (1 × 10$^9$ cells) were treated with PBS, imsEV at different concentrations, or 0.5% Triton X-100 (v/v in PBS) at 37 °C for 2 h. All samples were centrifuged for 15 min (900×$g$) and photographed. The CBC was obtained using ADVIA 2120i (Siemens, Erlangen, Germany).

## Animals

C57BL/6J mice (6- to 8-week-old, female, 20 g) were purchased from Jackson Laboratory or Weitong Lihua Experimental Animal Technology Co. and maintained at the animal facility of The University of Texas MD Anderson Cancer Center or Jilin University in isolator cages in a pathogen-free facility in a standard environmentally controlled room with 50% humidity and 22 °C temperature under a 12–12 h light–dark cycle. Standard water and diet were offered for the mice. According to animal ethics, the mice were humanely euthanized when animals displayed indicators of distress that aligned with the established institutional standards for initiating early euthanasia in all experiments of tumour inhibition. Signs for euthanasia include laboured breathing, abnormal movement, hypothermia, hunched posture, and more than 20% of body weight loss compared with baseline.

## Animal surgery and tumour implantation

GL261-Luc-mCherry or SB28 cells (1 × 10$^5$) were engrafted into the caudate nucleus of the mice with guide screws as follows[69]. Briefly, anaesthetised mice were restrained on the operating table and given preemptive analgesia. A 2- to 3-mm-long incision was then made just to the right of the midline and anterior to the interaural line, and the coronal and sagittal sutures were identified and the bregma marked. Then, a 2-mm diameter twist drill was used to drill a small hole at a point 2.5 mm lateral and 2.5 mm anterior to the bregma, corresponding to a point above the caudate nucleus; a sterilized guide screw was then placed in the hole and gently screwed in until it was flush with the skull. Seven days after placement of the guide screw, the mice were re-anaesthetised and the tumour cell suspension was infused slowly (0.2 μL min$^{-1}$) into the brain. The mice were kept warm until recovery from anaesthesia and were allowed to move around freely thereafter.

## In vivo biodistribution of sEVs

For the sEV biodistribution experiments, at 14 days after implantation of the GBM cells, a luciferase substrate was injected and the presence of tumour was confirmed with an IVIS 200 imaging system (Xenogen). Next, sEVs labelled with the lipophilic dye DiR at a dose of 5 × 10$^{11}$ sEVs were injected into the tail vein of each mouse. The IVIS 200 imaging system was used to assess fluorescence distribution in the intact mice at 1, 2, and 4 h after injection. Finally, at 4 h, the mice were euthanized, the heart, liver, spleen, lung, kidney, and brain were removed, and the fluorescence distribution in these organs was assessed with an IVIS 200.

## In vivo tumour-treatment assays

Seven days after implantation, the establishment of intracranial tumours was confirmed using bioluminescence imaging. The mice were randomly divided into five groups and treated with PBS, sEV, Antibody combo, CD64-sEV, or imsEV. The treatment was administered every 3 days through tail vein injections at a dose of 5 × 10$^{11}$ sEVs per injection. It was observed that the fluorescence signals of luciferase were captured and analysed at 0, 3, 6, 9, 12, and 15 days. Survival curves were constructed using Kaplan–Meier methods for each group.

## Magnetic resonance imaging

Mice were subjected to imaging at the MD Anderson Small Animal Imaging Facility with a 7 Tesla (T) 30-cm horizontal bore magnet (Bruker Biospin MRI, Billerica, MA). Acquisition and image analysis followed the Facility's protocol and other published procedures[70,71]. Briefly, during the imaging procedure, each mouse was placed under anaesthesia with 2% isoflurane. Tumour detection involved acquiring T2-weighted coronal and axial images using specific parameters: T2-weighted coronal slices with a thickness of 0.75 mm were captured in a field of view (FOV) of 30 × 40 and a matrix size of 256 × 192 pixels, resulting in an in-plane resolution of 0.156 μm. Similarly, T2-weighted axial slices with a thickness of 0.75 mm were obtained in a FOV of 30 × 22.5, using a matrix size of 256 × 192 pixels, yielding an in-plane resolution of 0.117 μm. These images were acquired with a RARE (rapid acquisition with relaxation enhancement) sequence, with a repetition time (TR) of 3000 ms and an echo time (TE) of 57 ms. The regions suspected of containing lesions were delineated on each slice in a blinded manner using ImageJ. The volume was calculated by summing the delineated regions of interest in mm$^2$ × 1 mm slice intervals.

## Histologic and immunofluorescence assays

Mice from each treatment group were euthanized on the indicated number of days after tumour inoculation. After transcardial perfusion with saline and paraformaldehyde, the brain was extracted and placed in a solution of 30% sucrose until the brain tissue sank to the bottom and was collected afterwards. Next, the brain tissues were embedded in Tissue-Tek optimum cutting temperature (OCT) compound, frozen in liquid nitrogen, and maintained at −80 °C. Frozen tissue blocks were cut into 10 μm slices using a freezer and attached to adhesive glass slides. The tissue slices were stored in a −80 °C refrigerator for later use. For analysis of IFN-γ, CD8, or MHC-I, tissue samples were thawed

and then incubated in 0.1% Triton X-100 for 30 min, and then blocked with 10% goat serum for 1 h, and incubated with the primary antibodies overnight, after which the tissue samples were washed three times and then incubated with a fluorescent secondary antibody for 1 h at room temperature. Images were acquired with an LSM880 microscope (Carl Zeiss) and processed with Zeiss Zen software.

For histologic evaluation of paraffin sections, after mice were euthanized (Day 15), hearts, lungs, livers, spleens, kidneys and brains were extracted and fixed with paraformaldehyde. The fixed tissue was dehydrated in ethanol solution and xylene successively until the tissue was transparent. Dehydrated tissue was embedded in paraffin, cut into 4 μm wax sections and placed on glass slides. Then, slides were deparaffinized three times in xylene, followed by graded ethanol rehydration. Antigen retrieval, immunofluorescence staining, and immunohistochemical staining were then performed per the manufacturers' instructions[2]. The primary antibody against Ki67 (Abcam, ab15580) was used at 1:500 dilution. Hematoxylin and eosin staining were used to analyse normal organs, including the liver, lung, heart, spleen, and kidney. Image acquisition was done with a BX3 microscope (OLYMPUS).

## Statistical analysis

Statistical analyses were done with GraphPad Prism 9 and presented as means ± SD. Statistical significance was evaluated by one-way analysis of variance with Tukey's multiple comparisons test and Student's $t$-tests. For survival studies, log-rank (Mantel–Cox) tests were used. $P$ values of <0.05 were considered to indicate statistically significant differences.

## Reporting summary

Further information on research design is available in the Nature Portfolio Reporting Summary linked to this article.

# Data availability

The authors declare that data supporting the findings of this study are available within the article and its supplementary information. The mass-spectrometry proteomics data generated and analysed during the current study have been deposited into the ProteomeXchange Consortium via the PRIDE partner repository with the dataset identifier PXD045239. Additional information regarding results and methods can be requested from the corresponding authors (W.J., Z.Y., B.Y.S.K., L.S.T. and S.W.). All equipment and reagents are commercially available and are described in the "Methods" section. The raw numbers for charts and graphs are available in the Source Data file whenever possible. Source data are provided in this paper. Source data are provided with this paper.

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

## Acknowledgements

This work was supported in part by a Cancer Center Support (Core) Grant from the National Cancer Institute to The University of Texas MD Anderson Cancer Center (P30-CA016672), National Science Foundation (OIA-1946231, S.W.), Louisiana Board of Regent (LEQSF (2021-22)-RD-D-06, S.W.). National Natural Science of China (21HAA01203, Z.Y.), and Foundation of Jilin Science-Technology Committee (20230402042GH, Z.Y.). The authors thank Christine F. Wogan, MS, ELS from MD Anderson's Division of Radiation Oncology for editorial assistance, and K. Ban from MD Anderson's Department of Neurosurgery for assisting with animal surgery and tumour implantation. We would like to thank Cryo-EM Core Facility at UTHealth in Houston, the MD Anderson Small Animal Imaging Facility (SAIF), and MDACC Advanced Cytometry & Sorting Facility for technical support.

## Author contributions

W.J., B.Y.S.K., S.D. and S.W. conceived the work; W.J., L.S.T., L.R.T., B.Y.S.K., A.S.L., L.J.L. and S.W. supervised the research; W.J., X.L., S.D. and S.W. developed the technology; S.D., W.J., L.S.T., Z.Y. and B.Y.S.K. designed the experiments; W.J., L.S.T., S.D., Y.W., X.F.L., Y.M. and Y.B. provided intellectual input; S.D., Y.B., Z.Y., S.W. and W.J., wrote the manuscript with input from all authors; S.D., Y.Z., R.S., X.S. and W.D. stained cells and tissues and performed other in vitro experiments; S.D., J.H., M.K. and Y.Z. injected mice orthotopically with tumour cells; S.D., Y.B., D.L., K.H., A.A., B.R.S., A.G., M.C., S.J., J.Y., J.C. and S.K.T. analysed and interpreted the data; All authors have given approval to the final version of the manuscript.

## Competing interests

A provisional patent application based on the technology described in the manuscript has been filed by The University of Texas MD Anderson Cancer Center, with S.D., S.W., B.Y.S.K., Z.Y. and W.J. as inventors. A.S.L. and L.J.L. are shareholders of Spot Biosystems, Ltd. All other authors declare no competing interests.
