## [Peer Review File · Nature Communications]

Adaptive design of mRNA-loaded extracellular vesicles for targeted immunotherapy of cancerREVIEWER COMMENTS

Reviewer #1 (Remarks to the Author):

This paper by Dong, Liu, Bi and co-workers described an important strategy to dock different targeting antibodies onto the surface of mRNA-loaded exosomes, which including a microfluidic electroporation approach to produce CD64 expressed IFN- γ mRNA loaded exosome and combination of CD71 and PD-L1 antibodies on their surface. The resulting immunogenic exosome (imEXO) can target glioblastoma cells and generate potent antitumor activities. This work is pretty interesting, even they have published another strategy to generate mRNA loaded exosome (mentioned in their cited literature#2, Yang, Z. et al. Nature BME 2020). However, there are many major and minor issues need to be further addressed before its publication.

1. The generation of exosome by nsEP is very interesting, but some details are missing. It is not so clear how the plasmid can be uptake by the MEFs or HEK2937. How many plasmid can be loaded into the cells and how many can then be transcribed into mRNA? Another interesting thing is that whether the high amount producing of exosome can influence the uptake of plasmid by cells.
2. What is the morphology of imEXO? Some TEM image should be included to demonstrate it.
3. How to calculate the generation amount of exosome? It is better adding this detail either in the method part or in the figure legend.
4. Fig. 4j, I wonder if there is another approach to demonstrate how many imEXO can be conjugated with both anti-CD71 and anti-PD-L1 antibodies. As calculating from confocal imaging is not so accurate.
5. Fig. 4k, how to get the 40 nm difference? It is confusing from Fig. 4k.
6. Fig. 4e is little bit blurred, can the author make it clearer?
7. How to use inhibitor to treat cells should be added in the legend of figure 5.
8. For the biosafety study, how many exosomes are injected for this study? Apology if I miss this information.
9. In Figure 6, immunostaining of IFN- γ , MHC-I, and CD8 are presented. The authors also mentioned increased proportion of M1-type macrophage, so the immunostaining of M1 macrophage should be further added.
10. There are many mistakes and confusing description for Figure 4 and Figure 6. For example, Fig. 4k and Fig. 4j are wrong labelled as their figure legend cannot match these figures. Fig. 4i is missing from Figure 4. Wrong citation in the M1-type macrophages at the tumour site (Fig. 6j), here should be Fig. 6k. Authors should double-check these figures, citation, and legend. And it is not clear from Fig. 6i that CD8+ cells increased, the color is so weak to see clearly.
11. Another important issue, the author demonstrated that they achieved a 7-fold increase in the amount of target mRNA within the exosome in the Discussion part. So how many mRNA can be encapsulated? As the most promising lipid nanoparticle (LNP) technology, each LNP can only encapsulated limited mRNA, such as 2-3. The authors need to add this information or a deeper discussion here if they don't get the accurate amount of mRNA encapsulated in the exosome.
12. It is overclaimed in the Abstract part that imEXO selectively targeted glioblastoma cells in vivo, as there is also distribution in other organs, the language here should be edited.

Reviewer #2 (Remarks to the Author):

The authors have developed a method using electroporation to generate immunogenic exosomes bound to FC receptor (CD64) which in turn binds to antibody of choice (CD72 and PDL1 here). This approach was then used to deliver IFN γ mRNA to animals harboring gliomas. Mice had extended survival when receiving these exosomes compared to control exosomes or combination of antibodies alone.

While the technology is novel and the authors do an excellent job comparing mechanisms of exosome uptake and exosome production, the therapeutic aspect of the paper requires further experiments and clarifications. The paper in its current form requires several key experiments to be conducted in replicates before being suitable for publication.

General comments:

Sample size (n) is too low for both the in vitro and in vivo experiments. Use of SEM is not appropriate. Please change error bars to SD or 95% CL throughout. Please show all data points and clearly indicate in figure legends what the samples size is. Also make clear the number of times an experiment was repeated. This information is not clear.

Specific comments:

Figure 2: The sample size is small for these experiments, as an N of 3 is not appropriate for these conclusions. Experiments use MEFs to show exosome generation. Additionally, Use of SEM is inappropriate. Please show SD or 95%CI. No theoretical mean exists in an experiment done for the first time and hence SEM is meaningless. Exosomes were not further analyzed. Does electroporating result in cell death and the exosomes generated are apoptotic bodies? Perhaps EM should be performed to compare these exosomes to controls.

Authors say "RT-qPCR of IFN- γ mRNA revealed that Exos produced by nsEP system treatment contained much larger quantities of transcribed mRNAs than exosomes produced by other methods." What methods were used to generate the control exosomes? Characterization of msEP and nsEP is required. What is the difference in quality of exosomes generated?

Figure 4 e-g is hard to interpret. No green is seen in the middle panel. Quantification in f suggest 10 % and g suggests even more. 10% have mRNA but 60% colocalize? It seems unclear how these quantifications were done. It is difficult to see green in the middle conditions in representative images provided as well.

"Data represent means \pm SEM (b, d, f, g, and i) from representatives of three independent experiments with n = 3". Please clarify what this means. Is each dot a random replicate from one of three experiments? Are each of the experiments repeated 3 times? Then is n=9? If n is 9, then show all data points. This would allow one to see the data range obtained from different experiments.

Figure 6C, authors must provide all data points in the graph. Effects seem modest and it is hard to see differences. Individual data points must be shown.

What cell type is making IFN γ ? It doesn't appear to colocalize with tumor cells. Can mcherry be distinguished from exosome fluorescence? Does tumor mcherry affect ivis results? Why is d15 picked for e? It appears that microscopy is not sensitive enough to see IFN γ and CD8

and MHC I. I would suggest flow cytometry so IFN γ can be accurately measured and the cell type expressing this cytokine can be identified.

j-k at what day is the analysis performed?

It is not clear why there is no MHC class I expression in most groups. Flow cytometry is needed, as microscopy does not appear to be sensitive enough to detect MHC I which is normally expressed on many cell types within the tumor microenvironment.

Low n is of concern. All studies must be repeated to ensure the effect on survival and tumor growth and MHC I upregulation is reproducible.

Is IVIS sensitive enough to pick up biodistribution? Please comment.

Naïve mouse groups receiving treatment should be evaluated for biodistribution and impact on peripheral and CNS immune cells.

Figure 7: The authors use antibody combination as a control. What about IFN γ ? If the results are dependent on IFN γ , then efficacy of imEx should be compared to systemic or intratumoral IFN γ +CD71 and PDL1 not to CD71 and anti PDL1 alone.

Quantification of MRIs are required across all slices. It appears the representatives are not the same section of the brain as evident by how the ventricles appear. Tumors should be volumetrically analyzed.

At what time point are microscopy and flow cytometry done?

T cells should be compared in blood and tumor in cohorts after injection of different exosomes. A broad immunological panel is required to assess changes in different groups.

Quantification of T cells needs to be done. Why do CD8 T cells have no MHC-I?

Low n. These experiments need to be repeated to ensure reproducibility.

Figure S12 does not convincingly show a lack of adverse effects on blood cells. Cell counts must be performed. Activation status of PBMCs and splenocytes and microglia at baseline and in tumor bearing mice must also be assessed. An appropriate immune phenotyping must be performed in naïve and tumor bearing mice. Assessment should include immune profiles of PBMCs and brain cells post treatment with different exosomes.

Figure S14 The flow cytometry does not look typical for the GL261 model. Detectable CD8 T cells can be identified in tumors. As shown there is no clear population. Gating strategy must be shown and quantification is needed.

Figure S15: what time point?

Figure S17: A time point missing. CD8 gate seems inaccurate. Separation by CD8 and SSCA seems clear even in PBS group. Why is the gate not placed where population break is seen? Please show gating strategy.

Discussion:

Systemic immunosuppression in GBM is also a major problem (PMIDs PMID: 33253355, 20179016, 30104766). The efficacy of imex should be discussed in the context of systemic immunosuppression. Would iv administration during lymphopenia affect outcomes?

Potential of combination therapy with anti PD1 or other cytokines or ICB should be discussed.

Benefits of IFN γ brought by exosomes as opposed to oncolytic viruses could be discussed.

The extent a soluble cytokine would be as beneficial as the technology put forward should be discussed.

Would imex treatment change T cell priming? If mRNA is picked up by DCs, can it change T cell priming?

Reviewer #3 (Remarks to the Author):

In the manuscript "Adaptive design of mRNA-loaded exosomes for targeted immunotherapy of cancer" by Dong and colleagues, the authors describe an approach to incorporate mRNA, in this case for IFN γ , into small extracellular vesicles (sEVs, the authors describe as exosomes), conjugate these exosomes to CD71/PDL1 antibodies to increase uptake by glioblastoma cells and demonstrate immune reactions and anti-cancer responses. Overall, this is a well-designed and executed study. It is an interesting approach of using sEVs as therapeutic carriers, with inherent advantages over man-made liposomes. While the study is very interesting, a few areas of the manuscript overstate the content of the result section and a number of additional controls/suggestions are listed below.

In general, a few aspects of the manuscript show a certain level of lack of rigor: The authors should decide whether to use American or British English. For example, tumor and tumour are both used interchangeably. Please correct.

As another general remark, the authors only very briefly introduce the mRNA therapeutic and sEV fields. While the general article style seem to prevent a thorough introduction, the authors should regardless introduce the generic concepts to allow a wider audience of readers to understand the article. Some relevant guidelines (PMID: 30637094) describe that a certain amount of characterization of extracellular vesicles need to be conducted in order to allow the naming of exosomes as a specific subset of sEVs. The authors do not fulfil these requirements and as it is not the main focus of the manuscript to define the exact sEV subset, it is likely best to call the vesicles sEVs instead of exosomes.

Specific points:

1) Conceptually, it is not clear why the authors, in this specific GBM approach, use IFN γ mRNA over protein. So why not load IFN γ protein instead? The delivery of mRNA is prone to questions (in a therapeutic setting) of potential mRNA integration into the genome of cells, prolonged/chronic autoimmune reactions (especially if IFN γ is produced) or immune suppression, and more. While I support the notion of mRNA as an excellent platform, in the context of this manuscript, may I suggest I) that a comparison with the efficacy of IFN γ protein (instead of mRNA) sEV delivery is done and II) the advantages/disadvantages of mRNA and protein delivery in cancer are discussed (see other point on discussion below).

- 2) The immunofluorescence images in general need to be quantified to show a colocalization score.
- 3) Figure 4a, this is insufficient evidence to claim (in the results text) that the two proteins are colocalizing on sEVs. The microscope approach is of insufficient resolution and the data is here overinterpreted.
- 4) Figure 4b: How do you confirm that the sEVs are not disrupted or the antibody for FLAG is not able to bind the 'inner vesicle' part? To claim that this is experiment/result is evidence of an extravesicular localization is not sufficient.
- 5) Figure 4e, j: 4e is suggested to show individual sEVs based on PKH26 labels in a confocal microscopy approach. This is insufficient as a) PKH dyes are well known and described to form aggregates, which at this resolution and approach, will be completely indistinguishable from sEV-bound PKH; and overall b) the resolution is far too slow to achieve single vesicle resolution. This data needs to be confirmed using alternative approaches (as done for example in PMID: 30949308)
- 6) Figure 4i shows Anti-CD71 and Anti-PD-L1 amounts on sEVs, but it does not show their binding to CD64. Using the CD64 FLAG construct, would it be possible to pull down FLAG to confirm the presence of the two antibodies?
- 7) Figure 5 nicely demonstrates that the immunogenic sEVs in vivo cause impaired Ki67 levels. Would this anti-proliferative or cell viability effect be also observed in vitro? If yes, how can this be interpreted? If no, which other cell type(s) are likely to be causative?
- 8) A serious concern and limitation of the approach of using sEVs as therapeutic platform is the possible immunogenicity, especially if sEVs are produced by a generic (and not patient/mouse line specific) donor line. Here, the use of C57/Bl6 MEFs in the same mouse strain bypassed this problem, which is a very laudable approach. Could the authors please provide, in addition to SFig16, acute (hours after injection) and delayed (days after single/repeat injections) evaluations of cytokines, white blood cell counts, T/NK cell phenotypes and/or other immune response markers for the injection of escalating doses of the imExo? Ideally, this should be done in naïve as well as GBM-bearing mice.
- 9) Discussion: The current discussion really lets the manuscript down. It is far too extensive on re-stating the results, and not exploring/contrasting the findings of other studies. Interesting sentences, for example "The current strategies of loading target mRNA into exosomes are passive and are strongly influenced by the size of the mRNA to be loaded. The larger the mRNA, the lower the efficiency of mRNA loading." Are not in-depth explored, neither are they referenced at all. Please re-write the discussion, remove redundant result descriptions from your work and describe the limitations, opportunities and challenges of your approach. Please see above the IFNg protein vs mRNA suggestion also for the discussion section.

Revised Manuscript (ID: NCOMMS-22-53386-T) “Adaptive design of mRNA-loaded sEVs for targeted immunotherapy of cancer”

Response to Reviewer Comments

REVIEWER 1 (COMMENTS FOR THE AUTHOR)

This paper by Dong, Liu, Bi and co-workers described an important strategy to dock different targeting antibodies onto the surface of mRNA-loaded exosomes, which including a microfluidic electroporation approach to produce CD64 expressed IFN- γ mRNA loaded exosome and combination of CD71 and PD-L1 antibodies on their surface. The resulting immunogenic exosome (imEXO) can target glioblastoma cells and generate potent antitumor activities. This work is pretty interesting, even they have published another strategy to generate mRNA loaded exosome (mentioned in their cited literature#2, Yang, Z. et al. Nature BME 2020). However, there are many major and minor issues need to be further addressed before its publication.

Response: We appreciate all the reviewers' comments, which we have addressed one by one below. We believe that we have improved the quality of our manuscript.

Before going through the detailed responses, we would like to address one issue: upon our review of the position statement of the International Society for Extracellular Vesicles and updated MISEV2018 guidelines¹, we chose to change the term “Exo” (exosome) throughout the manuscript to “sEV” (small extracellular vesicle).

1. The generation of exosome by nsEP is very interesting, but some details are missing. It is not so clear how the plasmid can be uptake by the MEFs or HEK2937. How many plasmid can be loaded into the cells and how many can then be transcribed into mRNA? Another interesting thing is that whether the high amount producing of exosome can influence the uptake of plasmid by cells.

Response: (1) As noted on page 2 of the revised manuscript, we used a two-step electroporation process for plasmid transfection: nanosecond electrical pulses were first applied to polarize the membrane structure of organelles (e.g., nucleus), followed by millisecond pulses that permeabilized the cellular plasma membrane of the source cells (Fig. 2a and Supplementary Fig. 1). This approach increased the efficiency and effectiveness with which plasmids were delivered to cells. In other words, nsEP produces more sEVs and supplies sufficient mRNA in host cells for their encapsulation in the produced sEVs. Additional information is provided in the first two paragraphs of the Discussion section on page 13 as “In this study, we report a nsEP system with microfluidic configuration that is capable of generating large quantities of sEVs with mRNA probes effectively encapsulated. By applying separately millisecond and nanosecond pulses, the main impact of electroporation could shift from the cell membrane to the membrane structure of organelles.²⁻⁵ And longer nanosecond pulses at 600-800 ns was found to effectively deliver exogenous cargos in a more transient and reversible manner with little compromise to cell viability when compared to the irreversible electroporation at 10-300 ns that triggered the pulse-induced cell apoptosis.⁴ In this study, we investigate the potential of nsEP stimulation strategy to leverage sEV secretion for its remotely tailoring ability of organelle membrane in cells. An impressive enhancement on the yield of sEVs that is over 40-folds higher than that from natural secretion was achieved under optimized stimulation conditions. Additionally, coupled with microfluidic further improved sEV yield following electroporation by suppressing air bubbles formation that interfered with electric pulses and impaired the treated cell viability and enabling parallel throughput processing.”

(2) To estimate the number of plasmids taken up by each cell, we used qPCR to quantify the copies of plasmids and later transcribed mRNA in the transfected cells according to the following assay flowchart (Supplementary Fig. 38):

Supplementary Fig. 38

Briefly, cells were first transfected with plasmids by electroporation as described above and divided into two separate groups for further culturing as follows: After cells re-attached to the surfaces of the culturing wells (~3 hours later), half of the transfected cells were washed with fresh medium to ensure that all subsequently extracted plasmids would be those that had been inside the cells. Copy numbers of plasmids were then determined by qPCR. The copy number of plasmids per cell was then calculated by dividing the measured copy number of target gene by the number of cells (estimated by measuring cell viability). Numbers of mRNAs transcribed from the loaded plasmids were estimated similarly, but from the other half of the transfected cells, which had been collected 6 hours after plasmid delivery (because proteins had started appearing inside cells by that time).

In detail, a mean of ~5600 copies of plasmid DNA were present in each cell (Supplementary Fig. 2a), and a mean of ~16000 copies of the corresponding mRNA were transcribed and available within each cell at 6 hours after plasmid delivery (Supplementary Fig. 2b).

Supplementary Fig. 2

(3) We believe that the sEVs did not interfere with DNA plasmid transfection for the following reasons. Before transfection, cells were washed, and the transfection was done in fresh serum-free OPTI-MEM medium. After plasmid transfection, cells were cultured in fresh exosome-free medium. In this way, we avoided the possibility that the high sEV numbers would affect the effectiveness of plasmid transfection into cell samples.

2. What is the morphology of imEXO? Some TEM image should be included to demonstrate it.

Response: Thank you for this question and the opportunity to address it (see new Fig. 4m). Cryo-electron microscopy demonstrated that CD64-sEV and imsEV derived from MEFs treated with the nsEP system exhibited electron-dense cargo in the lumen, whereas sEVs from untreated MEFs seemed to be devoid of such content. The surface characteristics of imsEV, relative to CD64-sEV, exhibited increased depth, thereby confirming that IgG was attached on the surface of the imsEVs. The corresponding methods have been added to the revised manuscript.

Fig. 4m

3. How to calculate the generation amount of exosome? It is better adding this detail either in the method part or in the figure legend.

Response: Concentrations of sEVs in the cell culture medium were measured with NanoSight; the total amount of sEVs = concentration of sEVs × volume. The number of viable cells was calculated by trypan blue staining and cell counting. The number of sEVs produced per cell = total number of sEVs/number of living cells. We have added these details to the legend of Fig. 2.

4. Fig. 4j, I wonder if there is another approach to demonstrate how many imEXO can be conjugated with both anti-CD71 and anti-PD-L1 antibodies. As calculating from confocal imaging is not so accurate.

Response: To address this excellent question on characterizing antibody conjugation on the imsEVs, we would like to clarify the image analysis method we used. Our group previously designed the TLN assay for molecular characterization of single extracellular vesicles; that assay was shown to successfully detect mRNAs and proteins at the single-extracellular vesicle level⁶⁻⁹. Our TLN biochip can successfully characterize multiplexed single extracellular vesicles upon high-resolution total internal reflection fluorescence (TIRF) microscopy. In the current study, we followed the same protocol, first conjugating the fluorescence-labeled antibodies (i.e.,

anti-CD71 and anti-PD-L1) with surface-expressed CD64 proteins and then detecting both simultaneously at the single-extracellular vesicle level with the biochip. To further clarify the imsEV conjugation with anti-CD71 and anti-PD-L1 in this study, we added high-resolution flow cytometry as a supplementary characterization method; **Supplementary Fig. 13** and its associated methods have been added to the manuscript. By focusing on double-positive vesicles (anti-CD71 and anti-PD-L1), we showed that $\sim 53.12 \pm 3.26\%$ of the double-positive events were detected above the fluorescence background in the PKH26-positive imsEV samples. This observation demonstrates that imsEVs can effectively bind with functional anti-CD71 and anti-PD-L1. Although the efficiency of colocalization varied, we hypothesized that those variations stemmed primarily from the static and dynamic movement of imsEVs while they were being detected by the different platforms.

Supplementary Fig. 13

5. Fig. 4k, how to get the 40 nm difference? It is confusing from Fig. 4k.

Response: We sincerely apologize for our error in which we used the peak particle size to represent the average particle size. We re-reviewed our raw data files and determined that the average size of CD64-sEV particles was 172.7 ± 3.1 nm, and the average size of imsEV particles was 184.3 ± 4.8 nm. In other words, binding IgG to CD64-sEVs increased the particle size by 11.6 nm.

Q6. Fig. 4e is little bit blurred, can the author make it clearer?

Response: Thank you for the opportunity to improve the quality of our images. We have added a higher-resolution version of **Fig. 4f**:

Fig. 4f

Q7. How to use inhibitor to treat cells should be added in the legend of figure 5.

Response: Thank you for this comment. We have expanded the legend of Fig. 5 as follows: “Before being incubated with PKH26-labeled sEVs, tumour cells were incubated for 1 hour with various endocytosis inhibitors (sucrose [0.4 μM], a clathrin-dependent endocytosis inhibitor; nystatin [50 μM], a caveolae-dependent endocytosis inhibitor; and cytochalasin D [5 μM], a micropinocytosis inhibitor) or an antibody to CD71 (10 μg mL⁻¹), after which the cells were thoroughly washed three times with PBS before being exposed to the PKH26-labeled sEVs.”

Q8. For the biosafety study, how many exosomes are injected for this study? Apology if I miss this information.

Response: For the biosafety evaluation, we injected 0, 2.5×10¹¹, 5×10¹¹, 7.5×10¹¹, or 10×10¹¹ sEVs into healthy mice and measured BUN, AST, ALT, and creatinine in blood samples obtained 4 hours later.

Q9. In Figure 6, immunostaining of IFN-γ, MHC-I, and CD8 are presented. The authors also mentioned increased proportion of M1-type macrophage, so the immunostaining of M1 macrophage should be further added.

Response: We appreciate this valuable comment. Frozen brain sections were obtained from SB28 tumour-bearing mice after the various treatments (Supplementary Fig. 31b) and subjected to immunofluorescence analysis. The macrophage marker F4/80 appeared as green, and the M1 polarization-specific cell marker CD80 appeared as red. Notably, an augmented yellow fluorescence signal was observed in brain sections from mice treated with imsEV as compared with mice receiving other treatments. These findings led us to propose that the increased presence of IFN-γ in the TME induced the M1 polarization of tumour-associated macrophages.

Supplementary Fig. 31b

Q10. There are many mistakes and confusing description for Figure 4 and Figure 6. For example, Fig. 4k and Fig. 4j are wrong labelled as their figure legend cannot match these figures. Fig. 4i is missing from Figure 4. Wrong citation in the M1-type macrophages at the tumour site (Fig. 6j), here should be Fig. 6k. Authors should double-check these figures, citation, and legend. And it is not clear from Fig. 6i that CD8⁺ cells increased, the color is so weak to see clearly.

Response: We appreciate the valuable feedback on Fig. 4 and 6. We apologize for the mistakes and the ensuing confusion. We have carefully reviewed the figures and their citations and legends and corrected the descriptions throughout the manuscript.

We also acknowledge the concern raised regarding the clarity of the increase in CD8⁺ cells in Fig. 6i; we adjusted these images accordingly to improve visualization of CD8⁺ cells in tumours of imsEV-treated mice.

Fig. 6i

Q11. Another important issue, the author demonstrated that they achieved a 7-fold increase in the amount of target mRNA within the exosome in the Discussion part. So how many mRNA can be encapsulated? As the most promising lipid nanoparticle (LNP) technology, each LNP can only encapsulated limited mRNA, such as 2-3. The authors need to add this information or a deeper discussion here if they don't get the accurate amount of mRNA encapsulated in the exosome.

Response: Our sincere apologies for mistakenly referencing the figure depicting the increase in total sEV while discussing the 7-fold increase in mRNA. Closer examination of the actual

increase in mRNA showed that it increased by 3-fold (Fig. 4e). We greatly appreciate this astute observation and apologize for the confusion caused by this error.

To address this point, we quantified mRNA in sEVs at 6 hours after stimulation and measured the number of sEVs. Around 54 copies and 176 copies of target mRNA per 100 sEVs were obtained for nsEP and nsEP/Npep, respectively (Supplementary Fig. 10b). Together with fluorescence single vesicle detection method (Fig. 4f, g), we confirmed that using the nsEP/N peptide strategy further increased the encapsulation efficiency. The affinity of the N peptide and box B interactions influenced the average number of mRNAs in individual sEVs, leading to two mRNA probes being actively loaded in each sEV with the help of the N peptide. The qPCR results (Supplementary Fig. 10b) suggested an average of 1.7 mRNAs were encapsulated into each single sEV. All details have been added to the Discussion on pages 13 as “In addition to the large-scale production of sEVs, our method also facilitates enrichment of the doses of target RNA probes in the sEVs, in two ways. First, combining electric pulses of different duration (i.e., nsEP and msEP) enhances the loading efficiency of plasmids as well as their expression kinetics²⁹. In detail, the nanosecond pulses help to increase the permeability of the nuclear membrane of the treated cells and accelerate transportation of plasmids to the nucleus and the overall transcription process. Around 54 copies of target mRNA per 100 sEVs were obtained as in Supplementary Fig. 10b. The second means of enriching target-mRNA doses in sEVs is by promoting the recruitment of the target mRNA (IFN- γ mRNA) by engineering a small box B sequence in the 3' end of the target mRNA and the N peptide on CD64, which are overexpressed on the membrane of host cells. We confirmed that the specific binding affinity between the box B and the N peptide on its amino-terminal arginine-rich domain could selectively enrich the target RNA probes in sEVs during their formation and leveraged the average mRNA number in individual sEVs (Fig. 4e, g, Supplementary, 10b). Considering that the sEV population is similar in both cases (nsEP with and without N-peptide introduction), the increase in mRNA probes in sEVs produced by the nsEP-plus-N peptide approach is mainly attributable to having more than one mRNA per individual sEVs. Indeed, our fluorescence intensity analysis of the TLN images revealed that an estimated two or three mRNA probes were actively loaded in each sEV with the help of the N peptide (Fig. 4f). The qPCR results (Supplementary Fig. 10b) also suggest an average of 1.7 mRNA/sEV among sEVs with target mRNA.

Supplementary Fig. 10b

Q12. It is overclaimed in the Abstract part that imEXO selectively targeted glioblastoma cells *in vivo*, as there is also distribution in other organs, the language here should be edited.

Response: We appreciate the feedback regarding imsEVs selectively targeting glioblastoma cells *in vivo*. We agree that imsEVs were distributed in other organs, and we edited the abstract accordingly: “The resulting immunogenic exosomes (imsEV) preferentially targeted glioblastoma cells and generated potent antitumour activity *in vivo*, including against tumours intrinsically resistant to immunotherapy.”

REVIEWER 2: COMMENTS TO THE AUTHOR

The authors have developed a method using electroporation to generate immunogenic exosomes bound to FC receptor (CD64) which in turn binds to antibody of choice (CD72 and PDL1 here). This approach was then used to deliver IFN γ mRNA to animals harboring gliomas. Mice had extended survival when receiving these exosomes compared to control exosomes or combination of antibodies alone.

While the technology is novel and the authors do an excellent job comparing mechanisms of exosome uptake and exosome production, the therapeutic aspect of the paper requires further experiments and clarifications. The paper in its current form requires several key experiments to be conducted in replicates before being suitable for publication.

Response: Thank you for the positive feedback and these comments, which have improved the quality of our presentation.

As noted in the response to Reviewer 1, we wish to clarify that we modified the nomenclature in this revised manuscript to agree with the International Society for Extracellular Vesicles position statement and updated MISEV2018 guidelines,¹ and chose to change the term “Exo” (exosome) to “sEV” (small extracellular vesicle)¹ throughout the manuscript.

General Comments: Sample size (n) is too low for both the in vitro and in vivo experiments. Use of SEM is not appropriate. Please change error bars to SD or 95% CL throughout. Please show all data points and clearly indicate in figure legends what the samples size is. Also make clear the number of times an experiment was repeated. This information is not clear.

Response: Thank you for these comments on sample size and replicates; we agree that these details are necessary for clinical application of the platform. We have presented all data points and added sample sizes and replicates in the figure legends in the revised manuscript. We have also addressed the sample size (n) question in our responses to Q1, Q9, and Q11 below.

Specific Comments:

Q1. Figure 2: The sample size is small for these experiments, as an N of 3 is not appropriate for these conclusions. Experiments use MEFs to show exosome generation. Additionally, Use of SEM is inappropriate. Please show SD or 95%CI. No theoretical mean exists in an experiment done for the first time and hence SEM is meaningless. Exosomes were not further analyzed. Does electroporating result in cell death and the exosomes generated are apoptotic bodies? Perhaps EM should be performed to compare these exosomes to controls.

Response: We thank the reviewer for these comments, which we address in 4 parts below.

(1) We fully acknowledge the importance of conducting additional independent replication experiments to enhance the reliability of our findings, and to provide a justification for the chosen sample size. We initially determined the sample size based on a combination of factors such as feasibility, availability of resources, and previous literature. We referred to previous publications in *Nature Communications*¹⁰⁻¹² that share similarities with our experiments presented in **Fig. 2** and included a set number of three independent experiments. Consequently, considering the available data, we maintain that the conclusions derived from **Fig. 2** are plausible. We have carefully reviewed the number of independent experiments for all procedures documented in the complete manuscript, explicitly indicating whether the "n" represents each independent trial, or the number of independent experiments conducted within each specific procedure.

(2) Thank you for raising the issue of the inappropriate use of SEM and for requesting SD or 95% CI instead. We fully understand that SEM does not provide the necessary information for the dispersion of data points around the mean. After re-evaluating our data files in GraphPad, we discovered that we had actually used SD rather than SEM. We sincerely apologize for the confusion caused by the incorrect use of terminology, and we greatly appreciate your attention to detail, which has helped improve the accuracy of our study. We have thoroughly reviewed our data and ensured that all the images were analyzed and reported with SD to provide more robust statistical analysis and strengthen the rigor of our findings.

(3) We also appreciate the valuable feedback regarding the use of electron microscopy to characterize EV size and morphology, which we did and included the results in **Fig. 4m**. Cryo-electron microscopy demonstrated that sEV, CD64-sEV, and imsEV particles were of similar size and morphology; CD64-sEV and imsEV derived from MEFs treated with the nsEP system exhibited electron-dense cargo in the lumen, whereas sEV from untreated MEFs seemed devoid of such content. The surface characteristics of imsEV, relative to CD64-sEV, exhibited increased depth, thereby confirming that IgG was attached on the surface of the imsEVs.

Fig. 4m

(4) Notably, apoptotic bodies are typically 50 nm to 5000 nm in size,¹³ but our sEV samples were 40 nm to 200 nm. The overlap in size ranges complicates qualitative distinction and characterization of apoptotic bodies by electron microscopy. As an alternative approach, we investigated the presence of apoptotic bodies in our sEV products by western blotting. We collected natural sEVs and sEVs generated by nsEP. As a positive control for apoptotic bodies, we collected the supernatant of MEFs cultured in PBS for 48 hours followed by ultracentrifugation (100,000×g for 2 hours). Marker proteins associated with apoptotic bodies were subsequently measured in each sample, and the corresponding results are shown in **Supplementary Fig. 6**. Western blotting revealed that the levels of the apoptosis marker thrombospondin-1¹⁴ in nsEP-based sEVs were no different than those in the natural sEVs.

Supplementary Fig. 6

Q2. Authors say “RT-qPCR of IFN- γ mRNA revealed that Exos produced by nsEP system treatment contained much larger quantities of transcribed mRNAs than exosomes produced by other methods.” What methods were used to generate the control exosomes? Characterization of msEP and nsEP is required. What is the difference in quality of exosomes generated?

Response: We used sEVs produced by msEP as a control, and we also investigated the morphology of native sEVs and sEVs produced after msEP or nsEP. Our results

(Supplementary Fig. 5) indicated no significant differences in particle size or surface morphology.

Supplementary Fig. 5

Q3. Figure 4 e-g is hard to interpret. No green is seen in the middle panel. Quantification in f suggest 10 % and g suggests even more. 10% have mRNA but 60% colocalize? It seems unclear how these quantifications were done. It is difficult to see green in the middle conditions in representative images provided as well.

Response: We greatly appreciate this feedback and the opportunity for clarification. We have added the following higher-resolution version of Fig. 4f:

Fig. 4f

To address the questions on quantification and colocalization, we would like to clarify the method we used to characterize sEVs in this study. We previously designed the TLN assay to characterize sEVs, and that assay successfully detected mRNAs and proteins at the sEV level⁷⁻⁹. Our TLN biochip successfully characterized multiplexed sEVs upon high-resolution total

internal reflection fluorescence (TIRF) microscopy. In the current study, we followed the same protocol, first conjugating fluorescence-labeled anti-CD71 and anti-PD-L1 to surface-expressed CD64 proteins, then detected both simultaneously by using the biochip; the molecular beacon INF- γ was used to characterize INF- γ mRNA. The grey value is the sum of the intensities of all pixels within the calculated spot area³⁻⁵. High grey values represent more abundant encapsulated INF- γ mRNAs within sEVs as detected by the molecular beacons. In the current study, the nsEP/N-pep EVs encapsulated 3.5 times more mRNAs within each single sEV than the nsEPs. Colocalization is the co-existence of INF- γ mRNAs within sEVs, was calculated based on the pixel overlap between different channels. In the current study, both the nsEP/N-pep and nsEP methods led to encapsulated INF- γ mRNAs, as reflected by the colocalization rates of 80% and 60%. However, existence was not associated with mRNA encapsulation abundance (or intensity detected by the molecular beacon), the nsEP encapsulated much lesser INF- γ mRNAs compared with nsEP/N pep. **We have added the relevant methods to the revised manuscript.**

Q4. “Data represent means \pm SEM (b, d, f, g, and i) from representatives of three independent experiments with $n = 3$ ”. Please clarify what this means. Is each dot a random replicate from one of three experiments? Are each of the experiments repeated 3 times? Then is $n=9$? If n is 9, then show all data points. This would allow one to see the data range obtained from different experiments.

Response: We sincerely apologize for the lack of clarity in defining “ n ” in each image. In the revised manuscript, we have explicitly specified that “ n ” represents the number of independent trials or independent experiments, as applicable.

Q5. Figure 6C, authors must provide all data points in the graph. Effects seem modest and it is hard to see differences. Individual data points must be shown.

Response: Individual data points are shown in our revised Fig. 6c below.

Fig. 6c

Q6. What cell type is making IFN γ ? It doesn't appear to colocalize with tumor cells. Can mchery be distinguished from exosome fluorescence? Does tumor mchery affect ivis results? Why is d15 picked for e? It appears that microscopy is not sensitive enough to see IFN γ and CD8 and

MHCI. I would suggest flow cytometry so IFN γ can be accurately measured and the cell type expressing this cytokine can be identified.

Response: (1) Thank you for the suggestion to use flow cytometry to measure the proportion of IFN- γ in SB28 cells in tumour-cell suspensions from mice in the various treatment groups. We have added the following flow cytometry results:

Fig. 7i

Supplementary Fig. 28a

Briefly, flow cytometric analyses of tumour-cell suspensions revealed an increase in IFN- γ ⁺ tumour cells in the imsEV-treated mice. To enhance clarity, we modified the revised manuscript by rearranging the sequence of the presentation of our microscopic characterization and flow cytometry results.

(2) In this study, we used non-fluorescent sEVs in the evaluation of *in vivo* therapeutic effects, thereby eliminating the possibility of interactions with mCherry.

(3) The emission and excitation wavelengths of mCherry span roughly 550–650 nm and 540–590 nm, respectively; DiR is a fluorescent compound with an excitation peak at 754 nm and an emission peak at 778 nm. To facilitate our ability to monitor tumour growth, we used GL261 and SB28 cells stably expressing luciferase for the *in vivo* experiments. The fluorescence generated by luciferase and the excited fluorescence of the dye fall under different detection modes in IVIS, eliminating the possibility of interference.

(4) In the SB28 model, mice treated with PBS died at day 15 day after treatment, and thus the tumour growth curve ended at d15. To maintain consistency in the experiment, the tumour growth curves of the GL261 model also ended at d15.

(5) We agree that flow cytometry would also accurately measure the proportion of MHC-I in SB28 tumour-cell suspensions from mice in the different treatment groups. We have included the following flow cytometry results as Fig. 7j, and Supplementary Fig. 29a:

Fig. 7j

Supplementary Fig. 29a

Briefly, flow cytometric analysis revealed notable upregulation of MHC-I expression in SB28 tumour cells from the mice treated with imsEVs. This observed increase is probably attributable to the upregulation of IFN- γ within the TME of the mice in the imsEV treatment group.

Q7. J-k at what day is the analysis performed?

Response: We euthanized mice at the end of the treatment period (on day 15) and collected tissue samples for the immunoassays.

Q8. It is not clear why there is no MHC class I expression in most groups. Flow cytometry is needed, as microscopy does not appear to be sensitive enough to detect MHC-I which is normally expressed on many cell types within the tumor microenvironment.

Response: We appreciate the suggestion to use flow cytometry as a more sensitive method to visualize MHC class I expression. As noted in the response to Q6, we have included flow cytometry results indicating notable upregulation of MHC-I expression in SB28 tumour cells in mice treated with imsEVs, which could be attributed to upregulation of IFN- γ in the TME of mice in the imsEV group.

Fig. 7j

Supplementary Fig. 29a

Q9. Low n is of concern. All studies must be repeated to ensure the effect on survival and tumor growth and MHC I upregulation is reproducible.

Response: We deeply apologize for the inadvertent error in our original manuscript, in which we mistakenly labeled the number of independent replicates (n) as 3 instead of the correct values of 5 for tumour growth effects and 10 for survival analysis. After we updated the appropriate independent replicate values, we consider our data reproducible.

We have thoroughly reviewed all experiments in the manuscript to ensure that the correct number of independent replicates has been stated in the corresponding figure legends.

Q10. Is IVIS sensitive enough to pick up biodistribution? Please comment.

Response: IVIS is known for its sensitivity in detecting and quantifying fluorescent signals emitted by fluorochrome-labeled nanoparticles, and has been used for such measurements in mice in many published studies^{10,15,16}. Thus we used IVIS for this purpose in the current study.

Q11. Naïve mouse groups receiving treatment should be evaluated for biodistribution and impact on peripheral and CNS immune cells.

Response: We appreciate this valuable comment, which we have addressed as follows. We conducted additional experiments to evaluate the biodistribution of DiR-labeled sEVs, CD64-sEVs, and imsEVs in healthy mice after tail-vein injection. The mice were killed 4 h after the

injection and the brain, heart, lung, liver, spleen, and kidney were isolated and subjected to IVIS imaging. Findings are shown in **Supplementary Fig. 21**. Most sEVs were found to be distributed in the liver. Although sEVs can cross the BBB, the brains of mice in the sEV and CD64-sEV groups showed only weak fluorescence. Slightly stronger fluorescence was detected in the brains of imsEV-injected mice, possibly because of the vascular endothelial cells in the brain capillaries that compose the BBB express TfR1¹⁷. Notably, TfR1 expression is generally low in most normal cells¹⁸. By actively targeting cells with elevated TfR1 expression *in vivo*, the imsEVs, modified to express anti-CD71 on their surfaces, facilitated their accumulation in the brain.

Supplementary Fig. 21

We also assessed blood cell counts in naïve mice during the acute phase (at 6 hours after a single injection, **Supplementary Fig. 22 a-f**) and in the delayed phase (after 5 injections, **Supplementary Fig. 22 g-i**). The results indicated that neither the single injection nor the repeated injection of the various sEV preparations led to notable changes in the red blood cell, white blood cell, or lymphocyte counts in naïve mice, suggesting that the sEVs have good biological safety and biological compatibility.

Supplementary Fig. 22

The immunophenotype of T cells in naive mice (Supplementary Fig. 23 and 24) was also included. The proportions of CD8⁺ and CD4⁺ T cells in the blood and spleen of naive mice remained relatively unchanged after 5 repeated injections of the indicated treatments, confirming the safety, biocompatibility, and low immunogenicity of sEVs as an mRNA delivery carrier.

Supplementary Fig. 23

Supplementary Fig. 24

Q12. Figure 7: The authors use antibody combination as a control. What about IFNG? If the results are dependent on IFNG, then efficacy of imEx should be compared to systemic or intratumoral IFNG+CD71 and PDL1 not to CD71 and anti PDL1 alone.

Response: We appreciate this comment on control conditions. Although using IFN- γ would be a more comprehensive comparison, IFN- γ , given systemically, has significant side effects¹⁹⁻²¹. The FDA has approved an IFN- γ 1b formulation for subcutaneous injection to reduce the severity of these side effects²². Moreover, the short half-life of IFN- γ has led to IFN- γ 1b having disappointing results in the clinic^{23,24}. Also, the ability of IFN- γ to target tumours is limited because of the widespread expression of IFNGR²⁴. As an alternative, including the IFN- γ -encoding gene in various carriers (adenovirus²⁵, oncolytic viruses²⁶, and liposomes¹⁹) has been attempted for targeted delivery to tumour or immune cells to enhance the production of IFN- γ . For the GBM-specific approach used in the current study, we chose to use sEVs to encapsulate the IFN- γ mRNA because they can cross the BBB.

Q13. Quantification of MRIs are required across all slices. It appears the representatives are not

the same section of the brain as evident by how the ventricles appear. Tumors should be volumetrically analyzed.

Response: We visualized the therapeutic effects of the different sEVs by using T2-weighted coronal images of the heads of the mice in the different treatment groups (Fig. 7h) and marked the edge of the tumour for accuracy. Quantitative MRI showed that the imsEVs significantly inhibited tumour growth. The relevant methods are included in the revised manuscript.

Fig. 7h

Q14. At what time point are microscopy and flow cytometry done?

Response: We euthanized mice at the end of the treatment period (on day 15) and collected tissue samples for immunoassay at that time.

Q15. T cells should be compared in blood and tumor in cohorts after injection of different exosomes. A broad immunological panel is required to assess changes in different groups.

Response: We appreciate this recommendation and have included it as follows. We evaluated the proportions of CD8⁺ and CD4⁺ T cells in the blood and tumours of mice in the different treatment groups by flow cytometry (Supplementary Fig. 23). The proportions of these cells in the blood and spleen of naive mice remained relatively unchanged after 5 repeated injections, confirming that sEVs, as an mRNA delivery carrier, have excellent biological safety, biocompatibility, and low immunogenicity.

Supplementary Fig. 23

Distribution of these cell types within the tumours is presented in Fig. 6j and 7k in the revised manuscript.

Fig. 6j

Fig. 7k

Q16. Quantification of T cells needs to be done. Why do CD8 T cells have no MHC-I?

Response: We thank the reviewer's suggestion. We have quantified T cell typing in tumour, blood, and spleen, both in naive and tumour-bearing mice (Fig. 6j, Fig. 7k and Supplementary Fig. 23, 24, 34, and 35). Regarding MHC-I genes, they show constitutive expression in all tissues (include T cells), although there are marked differences in class I expression levels²⁷. In this study, we focused on the expression of MHC-I in GBM cells. We analyzed the treated SB28 tumours by flow cytometry, and the experimental results are shown in Fig. 7j and Supplementary Fig. 29a. Increased MHC-I expression in SB28 following imsEV treatment was observed.

Fig. 6j

Fig. 7k

Supplementary Fig. 23

Supplementary Fig. 24

Supplementary Fig. 34

Supplementary Fig. 35

Fig. 7j

Supplementary Fig. 29a

Q17. Low n. These experiments need to be repeated to ensure reproducibility.

Response: We appreciate your attention to the sample size (n) in our experiments. We acknowledge that larger samples would have provided more robust and reliable results, and we recognize the need to justify our choice of sample size. We initially determined the sample size based on a combination of factors such as feasibility, availability of resources, and previous literature. Specifically, we referred to previous publications in *Nature Communications* that share similarities with our experimental settings. In that context, a set number of three independent experiments was considered appropriate for providing meaningful insights. Of course, the choice of sample size involves trade-offs between various constraints and practical considerations, but we do acknowledge that using a larger sample size may strengthen our findings.

Q18. Figure S12 does not convincingly show a lack of adverse effects on blood cells. Cell counts must be performed. Activation status of PBMCs and splenocytes and microglia at baseline and in tumor bearing mice must also be assessed. An appropriate immune phenotyping must be performed in naïve and tumor bearing mice. Assessment should include immune profiles of PBMCs and brain cells post treatment with different exosomes.

Response: We first assessed blood cell counts in naïve (non-tumour-bearing) mice during the acute phase (at 6 hours after a single injection, Supplementary Fig. 22 a-f) and in the delayed phase (after 5 injections, Supplementary Fig. 22 g-i). The results indicated that neither the single injection nor the repeated injection of the various sEV preparations led to notable changes in the red blood cell, white blood cell, or lymphocyte counts in naïve mice, suggesting that the sEVs have good biological safety and biological compatibility.

Supplementary Fig. 22

We also analyzed the immunophenotype of T cells in naive mice (Supplementary Fig. 23 and 24). The proportions of CD8⁺ and CD4⁺ T cells in the blood and spleen of naive mice remained relatively unchanged after 5 repeated injections of the indicated treatments, confirming the safety, biocompatibility, and low immunogenicity of sEVs as an mRNA delivery carrier.

Supplementary Fig. 23

Supplementary Fig. 24

We conducted the same analyses of cell counts and T cells in the blood and spleen of mice bearing SB28 tumours after repeated injections of the indicated preparations (Supplementary Fig. 33, 34, and 35).

Numbers of white blood cells and lymphocytes were seemingly reduced by control (PBS), sEV, Antibody combo, and CD64-sEV treatment, as were proportions of CD4⁺ and CD8⁺ T cell populations in the blood and spleen, which could reflect the systemic immunosuppression associated with GBM. However, this apparent systemic immunosuppression was significantly reduced in mice treated with imsEVs.

These results, in combination with the findings from tumour growth experiments, suggest that tumour growth of mice treated with imsEVs was slower than tumour growth in the other treatment groups, and this effect may be associated with attenuated systemic immune suppression.

Supplementary Fig. 33

Supplementary Fig. 34

Supplementary Fig. 35

We also evaluated the activation status of microglia in the brains of SB28 tumour-bearing mice with Iba1 staining (Supplementary Fig. 32). The results of frozen-section staining indicated that

mice given imsEVs had elevated expression of Iba1 in tumours and microglia at the tumour margins. Increased Iba1 expression is a hallmark of microglial activation and an indication of adaptive immunity mediated by cytotoxic CD8⁺ T cells²⁸.

Supplementary Fig. 32

Q19. Figure S14 The flow cytometry does not look typical for the GL261 model. Detectable CD8 T cells can be identified in tumors. As shown there is no clear population. Gating strategy must be shown and quantification is needed.

Response: Thank you for the opportunity to respond to this comment. We understand the importance of quantification for validating our results; we have added details on our gating strategy (Supplementary Fig. 39) and included quantification data (Fig. 6j) in the revised manuscript. The gating strategy we used was based on the flow cytometry characteristics of unstained cells (shown below). Unstained cells serve as an essential control to establish baseline fluorescence levels and to accurately define the gating parameters for subsequent analyses. Characterizing the fluorescence characteristics of unstained cells allowed us to establish appropriate gates to distinguish specific cell populations and exclude background noise.

Supplementary Fig. 39

Q20. Figure S15: what time point?

Response: We euthanized mice at the end of the treatment period (on day 15) and collected tissue samples for immunoassay at that time.

Q21. Figure S17: A time point missing. CD8 gate seems inaccurate. Separation by CD8 and

SSCA seems clear even in PBS group. Why is the gate not placed where population break is seen? Please show gating strategy.

Response: We euthanized mice at the end of treatment (day 15) and collected tissue samples for immunoassay. We have added a schematic illustrating details of the gating strategy (Supplementary Fig. 39). As addressed in our response to Q19 above, our gating strategy was based on the flow cytometry findings from unstained cells (see below and Supplementary Fig. 39). The revised manuscript now includes a comprehensive description of all the flow cytometry gating strategies used in our experiments.

Supplementary Fig. 39

Discussion

Q22. Systemic immunosuppression in GBM is also a major problem (PMIDs PMID: 33253355, 20179016, 30104766). The efficacy of imex should be discussed in the context of systemic immunosuppression. Would iv administration during lymphopenia affect outcomes?

Response: We appreciate the reviewer's question. It is indeed recognized that glioblastoma

(GBM) is frequently associated with significant systemic immunosuppression. However, it is crucial to emphasize that this association between the tumour and the suppressive phenotype does not necessarily indicate a comprehensive impairment in cellular immunity.²⁹

T cells can be present within GBM specimens in humans, although in limited numbers and primarily located in perivascular areas. Interestingly, these intratumoural populations seem to differ from the expected CD8⁺ ('effector') phenotype, skewing towards more dominant CD4⁺ cells. Patients with GBM generally exhibit a loss of CD4⁺ cells and overall mild lymphopenia. In this study, a comparable pattern was observed in terms of blood cell counts and T-cell immune analysis in mouse models of GBM, but treatment with imsEVs attenuated this trend. We hypothesize that the imsEVs slows GBM growth by mitigating systemic immunosuppression.

Q23. Potential of combination therapy with anti PD1 or other cytokines or ICB should be discussed.

Response: We appreciate this comment. Although the occasional success of checkpoint blockade has generated considerable enthusiasm, combination immuno-oncology therapy has been hindered by significant clinical setbacks. Immune checkpoint blockade eliminates inhibitory signals that hinder the activation of T cells, allowing tumour-reactive T cells to surmount regulatory mechanisms and orchestrate an effective antitumour response.³⁰⁻³² Theoretically, cytokines such as interleukin (IL)-2, IL-15, IL-21, IL-12, and IL-10, among other pro-inflammatory cytokines, hold promise for enhancing the effects of immune checkpoint blockade. However, combining cytokines with immune checkpoint blockade is challenging, in part because soluble cytokines have a wide range of side effects depending on dose, route of administration, and frequency. Moreover, the small size (~10-40 kDa) and short half-life of circulating cytokines necessitate frequent dosing or continuous infusion to sustain therapeutic efficacy, which also have adverse toxic effects³³. Another challenge arises from the nonspecificity of cytokines for tumour targeting; non-specific cytokine distribution can have off-target effects and potentially limit therapeutic efficacy. Overcoming these challenges is a significant obstacle to the development of cytokine-conjugated immune checkpoint blockade therapies. In the current study, we attempted to circumvent these challenges by designing imsEVs to deliver IFN- γ mRNA and anti-PDL1 specifically to GBM, thereby mitigating the adverse reactions associated with cytokine therapy and simultaneously promoting the therapeutic effects of immune checkpoint blockade.

Q24. Benefits of IFN γ brought by exosomes as opposed to oncolytic viruses could be discussed.

Response: Thank you for this suggestion, which we have addressed in the revised **Discussion (pages 13-14)**. In this specific GBM therapy strategy, carriers for IFN- γ must cross the BBB, target tumour cells, and allow abundant and constant secretion of IFN- γ into the TME for effective immunotherapy³⁴⁻³⁶. Although oncolytic viruses encoding IFN- γ have shown cytokine release in the TME and beneficial antitumour effects *in vitro*^{26,37}, those studies were not designed specifically for GBM therapy. To date, adequate and constant IFN- γ expression in the TME within the brain has not been confirmed in trials of oncolytic virotherapy³⁸. One potential challenge for such studies is that, unlike sEVs, only specific groups of viruses can cross the BBB³⁸. Second, although one study found that an inserted peptide could increase the infectivity of glioma cells, most virus carriers result in untargeted viral replication, whereas sEVs can be used flexibly to target specific cell-surface functions^{39,40}. Third, unlike DNA-based drugs, mRNA does not carry a risk of accidental infection or opportunistic insertional mutagenesis, as it does not need to enter the nucleus to be functional^{41,42}. An intrinsic advantage of mRNA-based

immunotherapy lies in the fact that small amounts of loading are adequate to provide vigorous efficacy signals⁴³. Nevertheless, encapsulating mRNA into viruses that can cross the BBB is difficult because of their limited capacity (e.g., 4.5 kb for AAV rh10, parvovirus)^{38,44}. We conclude that sEVs have satisfying IFN- γ mRNA encapsulation rates, their surfaces can be easily modified for targeting tumours, and they are biocompatible for GBM therapy *in vivo*.

Q25. The extent a soluble cytokine would be as beneficial as the technology put forward should be discussed.

Response: Thank you for raising this point. As we describe in our response to Q23 and in the Discussion (page 13), IFN- γ -modulating cancer immunotherapies are being tested in clinical trials²¹. However, soluble IFN- γ , like other soluble cytokines, have a wide range of adverse effects when administered systemically¹⁹⁻²¹. The FDA approved has approved an IFN- γ 1b formulation for subcutaneous injection with the goal of reducing these severe side effects²². Moreover, as previously discussed, clinical studies of IFN- γ 1b have been disappointing because of the short half-life of the IFN- γ protein^{23,24}, and the capacity for tumour targeting is limited owing to the widespread expression of IFNGR²⁴. For these reasons, we chose to encapsulate IFN- γ mRNA within sEVs that can cross the BBB for GBM immunotherapy.

Q26. Would imex treatment change T cell priming? If mRNA is picked up by DCs, can it change T cell priming?

Response: Thank you for these interesting questions. We consider it unlikely that the imsEVs would affect T-cell priming by T cells, macrophages, or dendritic cells (DCs), for the following reasons. Gene knock-in or overexpression in T cells or macrophages via transfection reagents is difficult because these cells have low transfection efficiency, especially when they are quiescent. The transfection capability of sEVs is lower than that of commercial transfection reagents, leading us to speculate that imsEVs are unlikely to influence T-cell priming by macrophages.

In DCs, plasmid transfection and mRNA delivery can be done by viral vectors or electroporation. Liposomal transfection of DCs is possible but inefficient; even at high concentrations of liposomes or mRNA, the transfection rate is low⁴⁵⁻⁴⁷. Thus, the inadvertent transfection of DCs via sEVs is unlikely given the challenges both in delivering the mRNA to DCs and the uptake and expression of the mRNA cargo by DCs. We believe that imsEV treatment is unlikely to enhance cross-activation and priming of CD8⁺ T cells by DCs.

REVIEWER #3 (REMARKS TO THE AUTHOR)

In the manuscript “Adaptive design of mRNA-loaded exosomes for targeted immunotherapy of cancer” by Dong and colleagues, the authors describe an approach to incorporate mRNA, in this case for IFN γ , into small extracellular vesicles (sEVs, the authors describe as exosomes), conjugate these exosomes to CD71/PDL1 antibodies to increase uptake by glioblastoma cells and demonstrate immune reactions and anti-cancer responses.

Overall, this is a well-designed and executed study. It is an interesting approach of using sEVs as therapeutic carriers, with inherent advantages over man-made liposomes. While the study is very interesting, a few areas of the manuscript overstate the content of the result section and a number of additional controls/suggestions are listed below.

In general, a few aspects of the manuscript show a certain level of lack of rigor: The authors should decide whether to use American or British English. For example, tumor and tumour are both used interchangeably. Please correct.

As another general remark, the authors only very briefly introduce the mRNA therapeutic and sEV fields. While the general article style seem to prevent a thorough introduction, the authors should regardless introduce the generic concepts to allow a wider audience of readers to understand the article. Some relevant guidelines (PMID: 30637094) describe that a certain amount of characterization of extracellular vesicles need to be conducted in order to allow the naming of exosomes as a specific subset of sEVs. The authors do not fulfil these requirements and as it is not the main focus of the manuscript to define the exact sEV subset, it is likely best to call the vesicles sEVs instead of exosomes.

Response: We appreciate these valuable comments, and we believe that our addressing them has improved the quality of our manuscript.

As noted in the responses to Reviewers 1 and 2, we modified the nomenclature in the revised manuscript to agree with the International Society for Extracellular Vesicles position statement and updated MISEV2018 guidelines,¹ and chose to change the term “Exo” (exosome) to “sEV” (small extracellular vesicle)¹ throughout the manuscript. We have also taken care to use the word “tumour” throughout.

Specific points:

Q1. Conceptually, it is not clear why the authors, in this specific GBM approach, use IFN γ mRNA over protein. So why not load IFN γ protein instead? The delivery of mRNA is prone to questions (in a therapeutic setting) of potential mRNA integration into the genome of cells, prolonged/chronic autoimmune reactions (especially if IFN γ is produced) or immune suppression, and more. While I support the notion of mRNA as an excellent platform, in the context of this manuscript, may I suggest I) that a comparison with the efficacy of IFN γ protein (instead of mRNA) sEV delivery is done and II) the advantages/disadvantages of mRNA and protein delivery in cancer are discussed (see other point on discussion below).

Response: Thank you for raising this issue. We contend that demonstrating the efficacy of IFN- γ mRNA delivery over other strategies is essential in this work. The rationale for our choosing to use IFN- γ mRNA rather than other IFN- γ therapeutic agents in this GBM approach is as follows.

IFN- γ exerts antitumour effects by modulating the functions of tumour cells, immune cells, and other cells in the TME.⁴⁸ Abundant and constant secretion of IFN- γ into the TME is necessary for effective immunotherapy.³⁴⁻³⁶ Unlike DNA-based drugs, mRNA carries no risk of accidental infection or opportunistic insertional mutagenesis, because it does not have to enter the nucleus to be functional.^{41,42} Compared with protein/peptide drugs, mRNA can be translated continuously into encoded proteins/peptides to ensure long-lasting expression.⁴⁹⁻⁵³ Another intrinsic advantage of mRNA-based immunotherapy is that small amounts of loading are adequate to provide vigorous efficacy signals.⁴³ Secreted IFN- γ is known to regulate immune response by binding to its receptor (IFNGR) on various cells; several investigations have concluded that the presence of IFN- γ in the TME is required for triggering optimal antitumour responses in patients with cancer receiving immunotherapy⁵⁴⁻⁵⁷. Encapsulating IFN- γ protein into sEVs does not allow the continuous production of IFN- γ in the TME. Moreover, our surface-functionalized, non-toxic, and low-immunogenicity sEVs allowed specific interactions with targeted cells,⁵⁸ protected IFN- γ from endonucleases, and prevented it from immune detection, leading to its targeted delivery to the cells of interest, efficient entry into the cell, and sufficient potency with only mild side effects.⁵² Finally, the abundant safety and efficacy data obtained

from the SARS-CoV-2 mRNA vaccines, and their approval by the US FDA, further increased our confidence in using mRNA therapy for cancer immunotherapy.^{49,53,59} Therefore, we chose to use sEVs to encapsulate mRNA rather than protein/peptides, to enhance the effectiveness of immunotherapy. That said, considerable work is still needed to improve mRNA encapsulation, targeting, cellular uptake, and large-scale production to fulfill manufacturing needs and clinical approval.

Q2. The immunofluorescence images in general need to be quantified to show a colocalization score.

Response: Thank you for the opportunity to address the importance of the colocalization score; information on our methods and quantification has been added to the revised manuscript.

Q3. Figure 4a, this is insufficient evidence to claim (in the results text) that the two proteins are colocalizing on sEVs. The microscope approach is of insufficient resolution and the data is here overinterpreted.

Response: thank the reviewer's comment. We also used western blotting to demonstrate the coexpression of CD64 in CD64-sEV (Fig. 2b). We also modified the overinterpretation / overstatement of the results in the revised manuscript.

Fig. 2b

Q4. Figure 4b: How do you confirm that the sEVs are not disrupted or the antibody for FLAG is not able to bind the 'inner vesicle' part? To claim that this is experiment/result is evidence of an extravascular localization is not sufficient.

Response: Thank you for this insightful comment. To further confirm the topological structure of CD64 on the sEV membrane, we cloned Myc into the C-terminus of 3XFlag-CD64, resulting in 3XFlag-CD64-Myc-sEV. To assess the relationship of CD64 with the sEV membrane, we used pull-down experiments involving beads conjugated with Anti-Flag or Anti-Myc antibodies. The presence of sEV marker proteins was analyzed by western blotting. Results are shown in **Supplementary Fig. 9**. The 3XFlag-CD64-Myc-sEV construct could be pulled down with Anti-Flag beads, but not with Anti-Myc beads. For pre-lysate 3XFlag-CD64-Myc-sEV, both Anti-Flag and Anti-Myc beads pulled down CD64. This confirmed that the 3XFlag, bound to the N-terminal of CD64, was on the outside of the sEVs, whereas Myc at the C-terminal was located in the inner vesicle of the sEVs.

Supplementary Fig. 9

Q5. Figure 4e, j: 4e is suggested to show individual sEVs based on PKH26 labels in a confocal microscopy approach. This is insufficient as a) PKH dyes are well known and described to form aggregates, which at this resolution and approach, will be completely indistinguishable from sEV-bound PKH; and overall b) the resolution is far too slow to achieve single vesicle resolution. This data needs to be confirmed using alternative approaches (as done for example in PMID: 30949308).

Response: To address this excellent question on our characterization of the imsEVs (a question also posed as Q3 by Reviewer 1), we would like to clarify the method we used to characterize the sEVs in this manuscript. Our group previously designed the TLN assay for the molecular characterization of sEVs, and that assay could successfully detect mRNAs and proteins within sEVs.⁷⁻⁹ Those studies also showed that a TLN biochip could successfully characterize multiplexed sEVs by using high-resolution total internal reflection fluorescence (TIRF) microscopy. In the current work, we followed the same protocol, first conjugating the fluorescence-labeled antibodies (i.e., anti-CD71 and anti-PD-L1) with surface-expressed CD64 proteins and then detecting CD64; the INF- γ molecular “beacon” was used to characterize INF- γ mRNA.

Specific aspects of each point are addressed below.

To demonstrate that PKH26 labeled EVs at the single-vesicle level, we purified sEVs as follows. (TIRF images of PKH26 diluted in PBS in the same purification process were used as a negative control [Supplementary Fig. 10a]). We found no obvious signals in the PKH26 dye-only condition. We previously showed the superiority of our surface chemistry for preventing background noise caused by dye residuals³. The size of the pixels in the TIRF images also indicated minimal PKH26 dye aggregation; we previously found that EV clusters (aggregations) had large pixels but sEVs had small pixels.⁷ The pixels of the PKH26-sEVs on the TIRF images were within the range for those of single extracellular vesicles (i.e., <10 pixels).

We further confirmed that our biochip could detect single EVs as follows. First, to ensure that all analyses were conducted at the single-vesicle level, we used the upper limit of the linear range of single EVs (at $\sim 10^7 - 10^9$ sEVs), as we did previously.³ Second, as noted above, we found that clusters of EVs (aggregations) had large pixels whereas single EVs had small ones.⁷ In the current study, cryo-electron microscopy of MEFs showed that MEF-sEV did not aggregate or form clusters (Fig. 4m), and the pixel size of the MEF-EVs measured on the TIRF images was also <10 pixels.⁷

Finally, we agree that an alternative strategy would be helpful for verifying that we were characterizing single sEVs. We previously validated our chip technology by using high-resolution flow cytometry (ImageStream^x mark II). In this work, we added flow cytometry as a supplementary method for analyzing colocalization. Focusing on PKH26-positive vesicles, the nsEP/N-peptide condition demonstrated higher colocalization of IFN- γ (72.8%) compared with 0.098% for blank sEVs and 36.4% for the nsEP (Supplementary Fig. 10c). Quantitative analysis of IFN- γ mRNA positivity in the PKH26-positive vesicles showed that the nsEP/N-pep condition led to significantly greater colocalization than the nsEP group (Supplementary Fig. 10). Although the efficiency of colocalization varied, we hypothesized that the observed differences stemmed primarily from static and dynamic movements of the imsEVs during their detection with the different platforms.

Supplementary Fig. 10

Q6. Figure 4i shows Anti-CD71 and Anti-PD-L1 amounts on sEVs, but it does not show their binding to CD64. Using the CD64 FLAG construct, would it be possible to pull down FLAG to confirm the presence of the two antibodies?

Response: To confirm that the IgG bound to CD64, we incubated sEVs or CD64-sEVs with

anti-CD71 and anti-PD-L1 and then removed unbound IgG by ultracentrifugation. Western blotting was then used to detect the heavy and light chains of the anti-PD-L1 and anti-CD71 antibodies in the pellet and sEV marker proteins. Both heavy and light chains of anti-PD-L1 and anti-CD71 were detected in the CD64-sEV samples but not in the sEV samples (Supplementary Fig. 12). These findings indirectly support the notion that anti-PD-L1 and anti-CD71 are connected to sEVs through their interaction with CD64.

Supplementary Fig. 12

Q7. Figure 5 nicely demonstrates that the immunogenic sEVs *in vivo* cause impaired Ki67 levels. Would this anti-proliferative or cell viability effect be also observed *in vitro*? If yes, how can this be interpreted? If no, which other cell type(s) are likely to be causative?

Response: We did not observe any significant cytotoxicity from the imsEVs in the MTS assays (Supplementary Fig. 15), suggesting that the imsEVs did not affect cell viability *in vitro*.

Supplementary Fig. 15

Q8. A serious concern and limitation of the approach of using sEVs as therapeutic platform is the possible immunogenicity, especially if sEVs are produced by a generic (and not patient/mouse line specific) donor line. Here, the use of C57/BI6 MEFs in the same mouse strain bypassed this problem, which is a very laudable approach. Could the authors please provide, in addition to SFig16, acute (hours after injection) and delayed (days after single/repeat injections) evaluations of cytokines, white blood cell counts, T/NK cell phenotypes and/or other immune response markers for the injection of escalating doses of the imExo? Ideally, this should be done in naïve as well as GBM-bearing mice.

Response: Thank you for raising this important point, and for the opportunity to address it. We first assessed acute effects (at 6 hours after a single injection, Supplementary Fig. 22a-f) and delayed effects (after 5 repeated injections, Supplementary Fig. 22g-i) on blood cell counts in naïve mice. The experimental results (Supplementary Fig. 22) indicated that neither the single injection nor repeated injections of different sEV preparations led to notable changes in the red blood cell, white blood cell, or lymphocyte counts in naïve mice. These findings confirm the biological safety and compatibility of the sEVs.

Supplementary Fig. 22

We also analyzed the immunophenotype of T cells in the blood and spleen of naïve mice and SB28 tumour-bearing mice; results are presented in Supplementary Fig. 23, 24, 34, and 35. Notably, the proportions of CD8 T cells and CD4 T cells in the blood and spleen of naïve mice remained relatively unchanged after 5 repeated injections, confirming that sEVs, used as mRNA delivery carriers, had excellent biological safety and biocompatibility as well as low immunogenicity.

Supplementary Fig. 23

Supplementary Fig. 24

We also assessed blood cell counts and T-cell immunoassays in blood and spleen from SB28 tumour-bearing mice subjected to repeated injections (Supplementary Fig. 33). As shown below, treatment with PBS, sEV, the antibody combo, and CD64-sEV led to various levels of leukopenia and lymphopenia in blood samples. We also observed that the proportions of CD4⁺ T cells and CD8⁺ T cells were slightly reduced in both blood and spleen, which could be attributable to the systemic immunosuppression associated with GBM. However, in mice treated with imsEVs, the extent of this systemic immunosuppression was attenuated.

These findings, in combination with those from the tumour growth experiments, lead us to conclude that tumours grew more slowly in the imsEV mice than in the other treatment groups, and that imsEVs led to some attenuation of systemic immune suppression.

Supplementary Fig. 33

Supplementary Fig. 34

Supplementary Fig. 35

We subsequently evaluated the activation status of microglia by Iba1 staining in frozen sections of SB28 tumour-bearing mouse brains. Our results showed that mice given imsEVs showed

elevated expression of Iba1 in tumours and microglia at the tumour margins. Increased Iba1 expression is a hallmark of microglial activation and an indicator of adaptive immunity mediated by cytotoxic CD8⁺ T cells.²⁸

Supplementary Fig. 32

Q9. Discussion: The current discussion really lets the manuscript down. It is far too extensive on re-stating the results, and not exploring/contrasting the findings of other studies. Interesting sentences, for example “The current strategies of loading target mRNA into exosomes are passive and are strongly influenced by the size of the mRNA to be loaded. The larger the mRNA, the lower the efficiency of mRNA loading.” Are not in-depth explored, neither are they referenced at all. Please re-write the discussion, remove redundant result descriptions from your work and describe the limitations, opportunities and challenges of your approach. Please see above the IFN γ protein vs mRNA suggestion also for the discussion section.

Response: Thank you for this valuable feedback, and for your insights and suggestions for improvement. We apologize for the excessive restatement of results and lack of exploration and contrast with findings from other studies. To address these concerns, we have revised the entire Discussion section as follows:

In this study, we report an nsEP system with microfluidic configuration that is capable of generating large quantities of sEVs that encapsulate mRNA probes. Applying millisecond and nanosecond pulses separately shifted the main impact of electroporation from the cell membrane to the membrane structure of cellular organelles. These effects have been confirmed in work involving irreversible electroporation for cancer treatment *in vivo*³ and in our previous studies of the effective delivery of exogenous probes into cells²⁹. In irreversible electroporation, pulses of 10-300 ns were used to damage only the membrane structure of intracellular organelles, ultimately triggering pulse-induced cell apoptosis⁴. To use electroporation to effectively deliver exogenous cargos without compromising cell viability, we found that longer pulses (600-800 ns) led to more transient and reversible disruption of the membrane structures of both cell and cell nuclei⁵. In the current study, we investigated the potential of this new stimulation strategy to leverage sEV secretion for its ability to modify organelle membranes in cells. We achieved impressive enhancement of sEV yield—more than 40 times the yield from natural secretion—under optimized stimulation conditions. Integration of a microfluidic platform into this nsEP technology not only allows parallel processing for high yield of imsEVs with greater throughput but also suppresses gas bubble formation during electrical stimulation, which can interfere with the electric pulses and damage the treated cells. With the microfluidic setup, the flow quickly sweeps any gas bubbles away from the

electrode surface before they grow to undesirable size, to ensure that passing cells receive effective stimulation. This strategy improves the viability of source cells after electroporation, which is crucial for maintaining high sEV yield.

In addition to the large-scale production of sEVs, our method also facilitates enrichment of the doses of target RNA probes in the sEVs, in two ways. First, combining electric pulses of different duration (i.e., nsEP and msEP) enhances the loading efficiency of plasmids as well as their expression kinetics²⁹. In detail, the nanosecond pulses help to increase the permeability of the nuclear membrane of the treated cells and accelerate transportation of plasmids to the nucleus and the overall transcription process. The second means of enriching target-mRNA doses in sEVs is by promoting the recruitment of the target mRNA (IFN- γ mRNA) by engineering a small box B sequence in the 3' end of the target mRNA and the N peptide on CD64, which are overexpressed on the membrane of host cells. We confirmed that the specific binding affinity between the box B and the N peptide on its amino-terminal arginine-rich domain could selectively enrich the target RNA probes in sEVs during their formation and leveraged the average mRNA number in individual sEVs. Considering that the sEV population is similar in both cases (nsEP with and without N-peptide introduction), the increase in mRNA probes in sEVs produced by the nsEP-plus-N peptide approach is mainly attributable to having more than one mRNA per individual sEVs.

For a deeper understanding of the biological mechanisms underlying this nsEP-triggered sEV release, we used proteomics analysis, which implicated three proteins in the sEV secretion process: MLKL, ISG15, and Usp18. MLKL is known to be required for the effective generation of intraluminal and extracellular vesicles⁶⁰. We also verified that MLKL was pivotal in controlling sEV production after nsEP treatment, as MLKL deficiency led to reduced levels of sEV secretion, below the basal level of untreated cells. Others have found that an ISGylation modification of the multivesicular body protein TSG101 by ISG15 can facilitate its co-localization with lysosomes and promote their aggregation, thereby impairing sEV secretion, and that this effect could be reversed by the Ub-specific protease USP18⁶¹. The ISGylation targets of functional proteins in the secretion of sEVs are TSG101 and heat-shock proteins (HSPs)^{62,63}. Interestingly, although our proteomics profiling revealed ISG15 and USP18 as top candidates in the sEV secretion process (nsEP led to a 189-fold increase in ISG15 and an 81-fold increase in USP18), most downstream functional proteins of ISG15/USP18 signaling, including TSG101 and HSP90, were not significantly changed. Therefore, we excluded ISG15/USP18 as being the main factors for promoting sEV trafficking during nsEP. This differs from our previous findings on sEV secretion after cellular nanoporation (CNP), in which HSP90 and HSP70 were found to be critical for electroporation-stimulated sEV production: inhibiting both greatly reduced the numbers of sEVs produced after CNP⁶⁴. One possible explanation for this difference is the formation of a transient, localized heat shock to the cell membrane close to the nanopore during CNP, but not during nsEP. Hence even though an electroporation step is involved in both techniques, the major mechanisms underlying the enhancement of sEV secretion are different, although they may share some similarities. More detailed investigations may shed light on the molecular mechanisms underlying the biogenesis of sEVs and cargo sorting resulting from electroporation stimulation.

Because a natural receptor for the Fc domain on IgG is anchored on the external surface (on the N-terminal of CD64), the sEVs produced by our approach could be used to selectively target other cell types simply by changing the antibodies. In this work, we investigated the potential of these imsEVs for immunotherapy in GBM. Although the success of checkpoint blockade has generated considerable enthusiasm for immunotherapy in general, immunotherapy for GBM has not been successful clinically⁶⁵. GBM effectively evades immune surveillance, in part through downregulating MHC-I is usually downregulated in GBM cells⁶⁶. Exposing GBM cells to IFN- γ is thought to restore MHC-I expression on their surfaces⁶⁷. IFN- γ has antitumour effects by modulating the functions of tumour cells, immune cells, and other cells in the TME,⁴⁸ and effective immunotherapy seems to require abundant and constant secretion of IFN- γ into the TME³⁴⁻³⁶. However, delivery of soluble IFN- γ has a wide range of side effects that depend on dose, route of administration, and frequency¹⁹⁻²¹. The US FDA has approved the use of the recombinant protein IFN- γ 1b, given as subcutaneous injections, to reduce the risk of sEV side effects²². Moreover, IFN- γ is known to have a short half-life, which necessitates frequent dosing or continuous infusion to sustain therapeutic efficacy. Thus far IFN- γ 1b has shown disappointing results in

the clinic because of the short half-life of the IFN- γ protein and the toxicity associated with frequent dosing^{23,24}. Limited tumour targeting is another significant clinical challenge for the clinical use of cytokine immune checkpoint blockade, which in the case of IFN- γ is limited because of the widespread expression of IFN γ R²⁴. The nonspecific distribution of IFN- γ can also result in off-target effects and potentially limit its therapeutic efficacy. For these reasons, we explored ways of introducing the IFN- γ gene into the targeted tumour or immune cells by encapsulating the mRNA for IFN- γ in carriers to result in localized and constant production of IFN- γ .

Various carriers such as adenovirus²⁵, oncolytic viruses²⁶, and liposomes¹⁹ have been used to load the gene that encodes IFN- γ and allow cytokine release in the TME; some of these carriers have had beneficial antitumour effects *in vitro*^{26,37}. However, those studies were not designed specifically for GBM therapy. To date, adequate and constant IFN- γ expression in the TME within the brain has not been confirmed in trials of oncolytic virotherapy³⁸. One potential challenge for such studies is that, unlike sEVs, only specific groups of viruses can cross the BBB³⁸. Second, encapsulating large molecules (e.g., mRNA) into viruses that can cross the BBB is difficult because of their limited capacity (e.g., 4.5 kb for AAV rh10, parvovirus)^{38,44}. Moreover, although one study found that an inserted peptide could increase the infectivity of glioma cells, most virus carriers result in untargeted viral replication, whereas sEVs demonstrate flexible surface functionalization capability to target specific cells.^{39,40} Hence, sEVs present satisfying gene encapsulation capacity, with easy surface modification for targeting, and excellent biocompatibility as an IFN- γ carrier for GBM immunotherapy. Unlike DNA-based drugs, mRNA does not carry a risk of accidental infection or opportunistic insertional mutagenesis, as it does not need to enter the nucleus to be functional.^{41,42} An intrinsic advantage of mRNA-based immunotherapy lies in the fact that small amounts of loading are adequate to provide vigorous efficacy signals⁴³. Also, the abundance of positive safety and efficacy data obtained from the SARS-CoV-2 mRNA vaccines, together with approval and regulation of such vaccines by the US FDA, underscores the broad therapeutic potential of mRNA therapy, including cancer immunotherapy^{49,53,59}. For all of these reasons, we chose to encapsulate mRNA rather than other IFN- γ encoding drugs for effective immunotherapy.

In our current study, we verified that our imsEVs successfully bound both anti-CD71 and anti-PD-L1. We also found that our GBM cell-targeted imsEV, delivering IFN- γ mRNA and PD-L1 antibody, could reprogram the immune microenvironment of the tumour from an immunosuppressive to an immune-stimulating phenotype. Evidence of this reprogramming included the increased infiltration of effector immune cells, upregulation of MHC-I on cancer cells, and polarization of suppressive myeloid cells to an activating phenotype. These changes inhibited tumour growth and extended survival in preclinical GBM models, including models that are intrinsically immune-resistant. Correspondingly, our surface-functionalized, non-toxic, low-immunogenic sEVs allowed specific interactions with targeted cells⁵⁸, protected IFN- γ from endonucleases, and prevented its detection by the immune system, leading to targeted delivery to cells of interest, efficient entry into those cells, and potency with few severe side effects⁵². Collectively, our findings demonstrate that an adaptive design strategy that efficiently produces mRNA-loaded sEVs with targeting functionalities could pave the way for their adoption in cancer immunotherapy applications, offering a new avenue for improving the responsiveness of immune-resistant tumours. However, to meet manufacturing practice requirements and obtain regulatory approval for clinical dosages, further improvement regarding percentage of mRNA encapsulation, downstream processing, stability, and biosafety are still needed.

References

- 1 Théry, C. *et al.* Minimal information for studies of extracellular vesicles 2018 (MISEV2018): a position statement of the International Society for Extracellular Vesicles and update of the MISEV2014 guidelines. *Journal of extracellular vesicles* **7**, 1535750 (2018).
- 2 Breton, M. & Mir, L. M. Microsecond and nanosecond electric pulses in cancer treatments. *Bioelectromagnetics* **33**, 106-123 (2012).
- 3 Beebe, S. J., Fox, P. M., Rec, L. J., Willis, L. K. & Schoenbach, K. H. Nanosecond, high-intensity pulsed electric fields induce apoptosis in human cells. *The FASEB journal* **17**, 1-23 (2003).

- 4 Ford, W. E., Ren, W., Blackmore, P. F., Schoenbach, K. H. & Beebe, S. J. Nanosecond pulsed electric fields stimulate apoptosis without release of pro-apoptotic factors from mitochondria in B16f10 melanoma. *Archives of biochemistry and biophysics* **497**, 82-89 (2010).
- 5 Wang, S. & Lee, L. J. Micro-/nanofluidics based cell electroporation. *Biomicrofluidics* **7**, 011301 (2013).
- 6 Zhang, J. *et al.* Engineering a Single Extracellular Vesicle Protein and RNA Assay (siEVPRA) via In Situ Fluorescence Microscopy in a UV Micropatterned Array. *bioRxiv*, 2022.2008. 2005.502995 (2022).
- 7 Nguyen, L. T. *et al.* An immunogold single extracellular vesicular RNA and protein (AuSERP) biochip to predict responses to immunotherapy in non-small cell lung cancer patients. *Journal of Extracellular Vesicles* **11**, e12258 (2022).
- 8 Zhou, J. *et al.* High-throughput single-EV liquid biopsy: Rapid, simultaneous, and multiplexed detection of nucleic acids, proteins, and their combinations. *Science Advances* **6**, eabc1204 (2020).
- 9 Hu, J. *et al.* A signal-amplifiable biochip quantifies extracellular vesicle-associated RNAs for early cancer detection. *Nature communications* **8**, 1683 (2017).
- 10 Li, G. *et al.* An injectable liposome-anchored teriparatide incorporated gallic acid-grafted gelatin hydrogel for osteoarthritis treatment. *Nature Communications* **14**, 3159 (2023).
- 11 Li, J. *et al.* Boron encapsulated in a liposome can be used for combinational neutron capture therapy. *Nature Communications* **13**, 2143 (2022).
- 12 Chang, Y. *et al.* CAR-neutrophil mediated delivery of tumor-microenvironment responsive nanodrugs for glioblastoma chemo-immunotherapy. *Nature Communications* **14**, 2266 (2023).
- 13 Kakarla, R., Hur, J., Kim, Y. J., Kim, J. & Chwae, Y.-J. Apoptotic cell-derived exosomes: messages from dying cells. *Experimental & Molecular Medicine* **52**, 1-6 (2020).
- 14 Akers, J. C., Gonda, D., Kim, R., Carter, B. S. & Chen, C. C. Biogenesis of extracellular vesicles (EV): exosomes, microvesicles, retrovirus-like vesicles, and apoptotic bodies. *Journal of neuro-oncology* **113**, 1-11 (2013).
- 15 Luo, Z. *et al.* Neutrophil hitchhiking for drug delivery to the bone marrow. *Nature Nanotechnology*, 1-10 (2023).
- 16 Baek, M.-J. *et al.* Tailoring renal-clearable zwitterionic cyclodextrin for colorectal cancer-selective drug delivery. *Nature Nanotechnology*, 1-12 (2023).
- 17 Daniels, T. R., Delgado, T., Rodriguez, J. A., Helguera, G. & Penichet, M. L. The transferrin receptor part I: Biology and targeting with cytotoxic antibodies for the treatment of cancer. *Clinical immunology* **121**, 144-158 (2006).
- 18 Uhlén, M. *et al.* Tissue-based map of the human proteome. *Science* **347**, 1260419 (2015).
- 19 Yuba, E. *et al.* pH-sensitive polymer-liposome-based antigen delivery systems potentiated with interferon- γ gene lipoplex for efficient cancer immunotherapy. *Biomaterials* **67**, 214-224 (2015).
- 20 Wu, J. *et al.* Dynamic distribution and expression in vivo of the human interferon gamma gene delivered by adenoviral vector. *BMC cancer* **9**, 1-7 (2009).
- 21 Gocher, A. M., Workman, C. J. & Vignali, D. A. Interferon- γ : teammate or opponent in the tumour microenvironment? *Nature Reviews Immunology* **22**, 158-172 (2022).
- 22 Todd, P. A. & Goa, K. L. Interferon gamma-1b: a review of its pharmacology and therapeutic potential in chronic granulomatous disease. *Drugs* **43**, 111-122 (1992).
- 23 Razaghi, A., Owens, L. & Heimann, K. Review of the recombinant human interferon gamma as an immunotherapeutic: Impacts of production platforms and glycosylation. *J Biotechnol* **240**, 48-60, doi:10.1016/j.jbiotec.2016.10.022 (2016).
- 24 Gleave, M. E. *et al.* Interferon gamma-1b compared with placebo in metastatic renal-cell carcinoma. *New England journal of medicine* **338**, 1265-1271 (1998).

- 25 Liu, R. Y. *et al.* Adenovirus-mediated delivery of interferon- γ gene inhibits the growth of nasopharyngeal carcinoma. *J Transl Med* **10**, 256, doi:10.1186/1479-5876-10-256 (2012).
- 26 Bourgeois-Daigneault, M. C. *et al.* Oncolytic vesicular stomatitis virus expressing interferon- γ has enhanced therapeutic activity. *Mol Ther Oncolytics* **3**, 16001, doi:10.1038/mto.2016.1 (2016).
- 27 Howcroft, T. K., Weissman, J. D., Geronne, A. & Singer, D. S. A T lymphocyte-specific transcription complex containing RUNX1 activates MHC class I expression. *J Immunol* **174**, 2106-2115, doi:10.4049/jimmunol.174.4.2106 (2005).
- 28 von Roemeling, C. A. *et al.* Therapeutic modulation of phagocytosis in glioblastoma can activate both innate and adaptive antitumour immunity. *Nature communications* **11**, 1508 (2020).
- 29 Waziri, A. Glioblastoma-derived mechanisms of systemic immunosuppression. *Neurosurgery Clinics* **21**, 31-42 (2010).
- 30 Patel, S. A. & Minn, A. J. Combination cancer therapy with immune checkpoint blockade: mechanisms and strategies. *Immunity* **48**, 417-433 (2018).
- 31 Kubli, S. P., Berger, T., Araujo, D. V., Siu, L. L. & Mak, T. W. Beyond immune checkpoint blockade: emerging immunological strategies. *Nature reviews Drug discovery* **20**, 899-919 (2021).
- 32 Sharma, P. & Allison, J. P. Immune checkpoint targeting in cancer therapy: toward combination strategies with curative potential. *Cell* **161**, 205-214 (2015).
- 33 Holder, P. G. *et al.* Engineering interferons and interleukins for cancer immunotherapy. *Advanced drug delivery reviews* **182**, 114112 (2022).
- 34 Ivashkiv, L. B. IFN γ : signalling, epigenetics and roles in immunity, metabolism, disease and cancer immunotherapy. *Nature Reviews Immunology* **18**, 545-558 (2018).
- 35 Tau, G. Z., Cowan, S. N., Weisburg, J., Braunstein, N. S. & Rothman, P. B. Regulation of IFN- γ signaling is essential for the cytotoxic activity of CD8 $^+$ T cells. *The Journal of Immunology* **167**, 5574-5582 (2001).
- 36 Jorgovanovic, D., Song, M., Wang, L. & Zhang, Y. Roles of IFN- γ in tumor progression and regression: a review. *Biomarker research* **8**, 1-16 (2020).
- 37 Oh, E., Choi, I.-K., Hong, J. & Yun, C.-O. Oncolytic adenovirus coexpressing interleukin-12 and decorin overcomes Treg-mediated immunosuppression inducing potent antitumor effects in a weakly immunogenic tumor model. *Oncotarget* **8**, 4730 (2017).
- 38 Foreman, P. M., Friedman, G. K., Cassady, K. A. & Markert, J. M. Oncolytic virotherapy for the treatment of malignant glioma. *Neurotherapeutics* **14**, 333-344 (2017).
- 39 Salunkhe, S., Basak, M., Chitkara, D. & Mittal, A. Surface functionalization of exosomes for target-specific delivery and in vivo imaging & tracking: Strategies and significance. *Journal of Controlled Release* **326**, 599-614 (2020).
- 40 Das, C. K. *et al.* Exosome as a novel shuttle for delivery of therapeutics across biological barriers. *Molecular pharmaceutics* **16**, 24-40 (2018).
- 41 Conry, R. M. *et al.* Characterization of a messenger RNA polynucleotide vaccine vector. *Cancer Res* **55**, 1397-1400 (1995).
- 42 Pardi, N., Hogan, M. J., Porter, F. W. & Weissman, D. mRNA vaccines - a new era in vaccinology. *Nat Rev Drug Discov* **17**, 261-279, doi:10.1038/nrd.2017.243 (2018).
- 43 Pastor, F. *et al.* An RNA toolbox for cancer immunotherapy. *Nature Reviews Drug Discovery* **17**, 751-767 (2018).
- 44 Hoshino, Y. *et al.* The adeno-associated virus rh10 vector is an effective gene transfer system for chronic spinal cord injury. *Scientific reports* **9**, 9844 (2019).
- 45 Martino, S. *et al.* Efficient siRNA delivery by the cationic liposome DOTAP in human hematopoietic stem cells differentiating into dendritic cells. *Journal of biomedicine and biotechnology* **2009** (2009).

- 46 Okano, K. *et al.* Evaluation of an mRNA lipofection procedure for human dendritic cells and induction of cytotoxic T lymphocytes against enhanced green fluorescence protein. *Tumor biology* **24**, 317-324 (2004).
- 47 Zhang, L. *et al.* Gene knock-in by CRISPR/Cas9 and cell sorting in macrophage and T cell lines. *JoVE (Journal of Visualized Experiments)*, e62328 (2021).
- 48 Mendoza, J. L. *et al.* Structure of the IFN γ receptor complex guides design of biased agonists. *Nature* **567**, 56-60 (2019).
- 49 Beck, J. D. *et al.* mRNA therapeutics in cancer immunotherapy. *Molecular cancer* **20**, 1-24 (2021).
- 50 Ilyichev, A., Orlova, L., Sharabrin, S. & Karpenko, L. mRNA technology as one of the promising platforms for the SARS-CoV-2 vaccine development. *Vavilov Journal of Genetics and Breeding* **24**, 802 (2020).
- 51 Guan, S. & Rosenecker, J. Nanotechnologies in delivery of mRNA therapeutics using nonviral vector-based delivery systems. *Gene therapy* **24**, 133-143 (2017).
- 52 Chaudhary, N., Weissman, D. & Whitehead, K. A. mRNA vaccines for infectious diseases: principles, delivery and clinical translation. *Nature reviews Drug discovery* **20**, 817-838 (2021).
- 53 Qin, S. *et al.* mRNA-based therapeutics: powerful and versatile tools to combat diseases. *Signal Transduction and Targeted Therapy* **7**, 166 (2022).
- 54 Mo, X. *et al.* Interferon- γ Signaling in Melanocytes and Melanoma Cells Regulates Expression of CTLA-4/IFN γ Regulates CTLA-4 in Melanocytes and Melanoma Cells. *Cancer research* **78**, 436-450 (2018).
- 55 Pegram, H. J. *et al.* Tumor-targeted T cells modified to secrete IL-12 eradicate systemic tumors without need for prior conditioning. *Blood, The Journal of the American Society of Hematology* **119**, 4133-4141 (2012).
- 56 Shi, L. Z. *et al.* Interdependent IL-7 and IFN- γ signalling in T-cell controls tumour eradication by combined α -CTLA-4+ α -PD-1 therapy. *Nature communications* **7**, 12335 (2016).
- 57 Schroder, K., Hertzog, P. J., Ravasi, T. & Hume, D. A. Interferon- γ : an overview of signals, mechanisms and functions. *Journal of leukocyte biology* **75**, 163-189 (2004).
- 58 Pullan, J. E. *et al.* Exosomes as drug carriers for cancer therapy. *Molecular pharmaceuticals* **16**, 1789-1798 (2019).
- 59 Shi, J., Kantoff, P. W., Wooster, R. & Farokhzad, O. C. Cancer nanomedicine: progress, challenges and opportunities. *Nature reviews cancer* **17**, 20-37 (2017).
- 60 Yoon, S., Kovalenko, A., Bogdanov, K. & Wallach, D. MLKL, the protein that mediates necroptosis, also regulates endosomal trafficking and extracellular vesicle generation. *Immunity* **47**, 51-65. e57 (2017).
- 61 Villarroya-Beltri, C. *et al.* ISGylation controls exosome secretion by promoting lysosomal degradation of MVB proteins. *Nature communications* **7**, 1-11 (2016).
- 62 Giannakopoulos, N. V. *et al.* Proteomic identification of proteins conjugated to ISG15 in mouse and human cells. *Biochemical and biophysical research communications* **336**, 496-506 (2005).
- 63 Sanyal, S. *et al.* Type I interferon imposes a TSG101/ISG15 checkpoint at the Golgi for glycoprotein trafficking during influenza virus infection. *Cell host & microbe* **14**, 510-521 (2013).
- 64 Yang, Z. *et al.* Large-scale generation of functional mRNA-encapsulating exosomes via cellular nanoporation. *Nature biomedical engineering* **4**, 69-83 (2020).
- 65 Jackson, C. M., Choi, J. & Lim, M. Mechanisms of immunotherapy resistance: lessons from glioblastoma. *Nature immunology* **20**, 1100-1109 (2019).
- 66 Wu, A. *et al.* Expression of MHC I and NK ligands on human CD133+ glioma cells: possible targets of immunotherapy. *Journal of neuro-oncology* **83**, 121-131 (2007).

- 67 Tanaka, K., Hayashi, H., Hamada, C., Khoury, G. & Jay, G. Expression of major histocompatibility complex class I antigens as a strategy for the potentiation of immune recognition of tumor cells. *Proceedings of the National Academy of Sciences* **83**, 8723-8727 (1986).

REVIEWERS' COMMENTS

Reviewer #1 (Remarks to the Author):

The author addressed most of my comments, one more issue is that CD80 may not be a good marker for immunostaining of M1 macrophage (Figure 6). I recommend using another M1 marker (like iNOS or CD86) for immunostaining. The paper can be published after they modified this issue in Figure 6.

Reviewer #3 (Remarks to the Author):

The authors have made a tremendous effort improving the quality of the manuscript and the additional data strengthens the work significantly.

A few aspects the authors suggest/discuss are not in full agreement with my opinion, I fully accept these as appropriate and overall recommend the authors for improving all aspects of the work, including the discussion.

One point is your response to reviewer#3/Q8: While the data you include is concordant with your suggestion that the treatment is not immunogenic (or better obviously impacting immune cell abundances), this nevertheless does not address the potential immunogenic response it might have if the modified sEV (imsEV) have if originating from a generic source and the (human) recipient detects those as nonself. While beyond the scope of the current study, it might be good to add this to the discussion, especially as this updated version discusses in the second part related clinical effects. As part of this, a few of the multiple references to the results of the study could be removed from the discussion, especially in the first half.

Revised Manuscript (ID: NCOMMS-22-53386A) “Adaptive design of mRNA-loaded extracellular vesicles for targeted immunotherapy of cancer”

Response to Reviewer Comments

Reviewer #1 (Remarks to the Author)

The author addressed most of my comments, one more issue is that CD80 may not be a good marker for immunostaining of M1 macrophage (Figure 6). I recommend using another M1 marker (like iNOS or CD86) for immunostaining. The paper can be published after they modified this issue in Figure 6.

Response: We thank the reviewer's comments. We have added CD86 as the M1 marker for M1 macrophage immunostaining in **Supplementary Fig. 31**. Immunofluorescence assessment of proportions of CD86⁺ macrophages in tumour tissues of mice in the indicated treatment groups showed that imsEV led to increased proportions of CD86⁺ macrophages.

Reviewer #3 (Remarks to the Author):

The authors have made a tremendous effort improving the quality of the manuscript and the additional data strengthens the work significantly.

A few aspects the authors suggest/discuss are not in full agreement with my opinion, I fully accept these as appropriate and overall recommend the authors for improving all aspects of the work, including the discussion.

One point is your response to reviewer#3/Q8: While the data you include is concordant with your suggestion that the treatment is not immunogenic (or better obviously impacting immune cell abundances), this nevertheless does not address the potential immunogenic response it might have if the modified sEV (imsEV) have if originating from a generic source and the (human) recipient detects those as nonself. While beyond the scope of the current study, it might be good to add this to the discussion, especially as this updated version discusses in the second part related clinical effects. As part of this, a few of the multiple references to the results of the study could be removed from the discussion, especially in the first half.

Response: We thank the reviewer's comments. We have added the discussion regarding the potential immunogenicity of modified sEVs in medical practice. We have also removed the redundant results in the discussion. The modified **Discussion section** is below:

“In this study, we report a nsEP system with the microfluidic configuration that is capable of generating large quantities of sEVs that encapsulate mRNA probes. Applying millisecond and nanosecond pulses separately shifted the main impact of electroporation from the cell membrane to the membrane structure of cellular organelles. These effects have been confirmed in work involving irreversible electroporation for cancer treatment *in vivo*¹ and in our previous studies of the effective delivery of exogenous probes into cells²⁻⁴. In the current study, we integrated this adequate stimulation strategy with microfluidic that is capable of parallel processing to leverage high-throughput sEV secretion and achieved an impressive enhancement of sEV yield—more than 40 times the yield from natural secretion. Additionally, the flow in microfluidic quickly sweeps any gas bubbles away from the electrode surface before they grow to undesirable size, to ensure that passing cells receive effective stimulation. It also improves the viability of source cells after electroporation, ensuring high sEV yield.

In addition to the large-scale production of sEVs, our method also facilitates enrichment of the doses of target RNA probes in the sEVs, in two ways. First, combining electric pulses of different duration (i.e., nsEP and msEP) enhances the loading efficiency of plasmids as well as their expression kinetics²⁹. In detail, the nanosecond pulses help to increase the permeability of the nuclear membrane of the treated cells and accelerate the transportation of plasmids to the nucleus and the overall transcription process. The second means of enriching target-mRNA doses in sEVs is by promoting the recruitment of the target mRNA (IFN- γ mRNA) by engineering a small box B sequence in the 3' end of the target mRNA and the N peptide on CD64, which is overexpressed on the membrane of host cells. We confirmed that the specific binding affinity between the box B and the N peptide on its amino-terminal arginine-rich domain could selectively enrich the target RNA probes in sEVs during their formation and leverage the average mRNA number in individual sEVs. Considering that the sEV population is similar in both cases (nsEP with and without N-peptide introduction), the increase in mRNA probes in sEVs produced by the nsEP-plus-N peptide approach is mainly attributable to having more than one mRNA per individual sEVs.

For a deeper understanding of the biological mechanisms underlying this nsEP-triggered sEV release, we used proteomics analysis, which implicated three proteins in the sEV secretion process: MLKL, ISG15, and Usp18. MLKL is known to be required for the effective generation of intraluminal and extracellular vesicles⁴. We also verified that MLKL was pivotal in controlling sEV production after nsEP treatment, as MLKL deficiency led to reduced levels of sEV secretion, below the basal level of untreated cells. Others have found that an ISGylation modification of the multivesicular body protein TSG101 by ISG15 can facilitate its co-localization with lysosomes and promote their aggregation, thereby impairing sEV secretion, and that this effect could be reversed by the Ub-specific protease USP18⁵. The ISGylation targets of functional proteins in the secretion of sEVs are TSG101 and heat-shock proteins (HSPs)^{6,7}. Interestingly, although our proteomics profiling revealed ISG15 and USP18 as top candidates in the sEV secretion process (nsEP led to a 189-fold increase in ISG15 and an 81-fold increase in USP18), most downstream functional proteins of ISG15/USP18 signaling, including TSG101 and HSP90, were not significantly changed. Therefore, we excluded ISG15/USP18 as being the main factors for promoting sEV trafficking during nsEP. This differs from our previous findings on sEV secretion after cellular nanoporation (CNP), in which HSP90 and HSP70 were found to be critical for electroporation-stimulated sEV production: inhibiting both greatly reduced the numbers of sEVs produced after CNP⁸. One possible explanation for this difference is the formation of a transient, localized heat

shock to the cell membrane close to the nanopore during CNP, but not during nsEP. Hence even though an electroporation step is involved in both techniques, the major mechanisms underlying the enhancement of sEV secretion are different, although they may share some similarities. More detailed investigations may shed light on the molecular mechanisms underlying the biogenesis of sEVs and cargo sorting resulting from electroporation stimulation.

Because a natural receptor for the Fc domain on IgG is anchored on the external surface (on the N-terminal of CD64), the sEVs produced by our approach could be used to selectively target other cell types simply by changing the antibodies. In this work, we investigated the potential of these imsEVs for immunotherapy in GBM. Although the success of checkpoint blockade has generated considerable enthusiasm for immunotherapy in general, immunotherapy for GBM has not been successful clinically⁹. GBM effectively evades immune surveillance, in part through downregulating MHC-I is usually downregulated in GBM cells¹⁰. Exposing GBM cells to IFN- γ is thought to restore MHC-I expression on their surfaces¹¹. IFN- γ has antitumour effects by modulating the functions of tumour cells, immune cells, and other cells in the TME,¹² and effective immunotherapy seems to require abundant and constant secretion of IFN- γ into the TME¹³⁻¹⁵. However, delivery of soluble IFN- γ has a wide range of side effects that depend on dose, route of administration, and frequency¹⁶⁻¹⁸. The US FDA has approved the use of the recombinant protein IFN- γ 1b, given as subcutaneous injections, to reduce the risk of sEV side effects¹⁹. Moreover, IFN- γ is known to have a short half-life, which necessitates frequent dosing or continuous infusion to sustain therapeutic efficacy. Thus far IFN- γ 1b has shown disappointing results in the clinic because of the short half-life of the IFN- γ protein and the toxicity associated with frequent dosing^{20,21}. Limited tumour targeting is another significant clinical challenge for the clinical use of cytokine immune checkpoint blockade, which in the case of IFN- γ is limited because of the widespread expression of IFNGR²¹. The nonspecific distribution of IFN- γ can also result in off-target effects and potentially limit its therapeutic efficacy. For these reasons, we explored ways of introducing the IFN- γ gene into the targeted tumour or immune cells by encapsulating the mRNA for IFN- γ in carriers to result in localized and constant production of IFN- γ .

Various carriers such as adenovirus²², oncolytic viruses²³, and liposomes¹⁶ have been used to load the gene that encodes IFN- γ and allow cytokine release in the TME; some of these carriers have had beneficial antitumour effects *in vitro*^{23,24}. However, those studies were not designed specifically for GBM therapy. To date, adequate and constant IFN- γ expression in the TME within the brain has not been confirmed in trials of oncolytic virotherapy²⁵. One potential challenge for such studies is that, unlike sEVs, only specific groups of viruses can cross the BBB²⁵. Second, encapsulating large molecules (e.g., mRNA) into viruses that can cross the BBB is difficult because of their limited capacity (e.g., 4.5 kb for AAV rh10, parvovirus).^{25,26} Moreover, although one study found that an inserted peptide could increase the infectivity of glioma cells, most virus carriers result in untargeted viral replication, whereas sEVs demonstrate flexible surface functionalization capability to target specific cells.^{27,28} Hence, sEVs present satisfying gene encapsulation capacity, with easy surface modification for targeting, and excellent biocompatibility as an IFN- γ carrier for GBM immunotherapy. Unlike DNA-based drugs, mRNA does not carry a risk of accidental infection or opportunistic insertional mutagenesis, as it does not need to enter the nucleus to be functional.^{29,30} An intrinsic advantage of mRNA-based immunotherapy lies in the fact that small amounts of loading are adequate to provide vigorous efficacy signals³¹. Also, the abundance of positive safety and efficacy data obtained from the SARS-CoV-2 mRNA vaccines, together with approval and regulation of such vaccines by the US FDA, underscores the broad therapeutic

potential of mRNA therapy, including cancer immunotherapy³²⁻³⁴. For all of these reasons, we chose to encapsulate mRNA rather than other IFN- γ encoding drugs for effective immunotherapy.

In our current study, we verified that our imsEVs successfully bound both anti-CD71 and anti-PD-L1. We also found that our GBM cell-targeted imsEV, delivering IFN- γ mRNA and PD-L1 antibody, could reprogram the immune microenvironment of the tumour from an immunosuppressive to an immune-stimulating phenotype. Evidence of this reprogramming included the increased infiltration of effector immune cells, upregulation of MHC-I on cancer cells, and polarization of suppressive myeloid cells to an activating phenotype. These changes inhibited tumour growth and extended survival in preclinical GBM models, including models that are intrinsically immune-resistant. Correspondingly, our surface-functionalized, non-toxic, low-immunogenic sEVs allowed specific interactions with targeted cells³⁵, protected IFN- γ from endonucleases, and prevented its detection by the immune system, leading to targeted delivery to cells of interest, efficient entry into those cells, and potency with few severe side effects³⁶. Collectively, our findings demonstrate that an adaptive design strategy that efficiently produces mRNA-loaded sEVs with targeting functionalities could pave the way for their adoption in cancer immunotherapy applications, opening up avenues for improving the responsiveness of immune-resistant tumours. Nevertheless, to meet manufacturing practice requirements and secure regulatory approval for clinical dosages, further improvements, including production, stability, quality control and safety assessments are still needed. For modified sEVs to be deemed suitable for human use and to mitigate potential risks such as potency loss, stringent control over immunogenicity is paramount, particularly for interventions involving repeated administrations. Encouragingly, modified sEVs derived from the HEK293T cell line have been shown to possess minimal immunogenicity in mice after repeated doses³⁷, and modified sEVs sourced from stem cells, such as mesenchymal stem cells (MSCs)³⁸, are expected to lack immunogenicity, given that MSCs are known for their low immunogenic potential. However, there is little evidence to prove modified cargos have low immunogenic activities or not in human recipients at the moment. Rigorous preclinical studies and adherence to regulatory guidelines are imperative before applying modified sEVs in human clinical settings.”

References

- 1 Beebe, S. J., Fox, P. M., Rec, L. J., Willis, L. K. & Schoenbach, K. H. Nanosecond, high-intensity pulsed electric fields induce apoptosis in human cells. *The FASEB journal* **17**, 1-23 (2003).
- 2 Ford, W. E., Ren, W., Blackmore, P. F., Schoenbach, K. H. & Beebe, S. J. Nanosecond pulsed electric fields stimulate apoptosis without release of pro-apoptotic factors from mitochondria in B16f10 melanoma. *Archives of biochemistry and biophysics* **497**, 82-89 (2010).
- 3 Wang, S. & Lee, L. J. Micro-/nanofluidics based cell electroporation. *Biomicrofluidics* **7**, 011301 (2013).
- 4 Yoon, S., Kovalenko, A., Bogdanov, K. & Wallach, D. MLKL, the protein that mediates necroptosis, also regulates endosomal trafficking and extracellular vesicle generation. *Immunity* **47**, 51-65. e57 (2017).
- 5 Villarroya-Beltri, C. *et al.* ISGylation controls exosome secretion by promoting lysosomal degradation of MVB proteins. *Nature communications* **7**, 1-11 (2016).

- 6 Giannakopoulos, N. V. *et al.* Proteomic identification of proteins conjugated to ISG15 in mouse and human cells. *Biochemical and biophysical research communications* **336**, 496-506 (2005).
- 7 Sanyal, S. *et al.* Type I interferon imposes a TSG101/ISG15 checkpoint at the Golgi for glycoprotein trafficking during influenza virus infection. *Cell host & microbe* **14**, 510-521 (2013).
- 8 Yang, Z. *et al.* Large-scale generation of functional mRNA-encapsulating exosomes via cellular nanoporation. *Nature biomedical engineering* **4**, 69-83 (2020).
- 9 Jackson, C. M., Choi, J. & Lim, M. Mechanisms of immunotherapy resistance: lessons from glioblastoma. *Nature immunology* **20**, 1100-1109 (2019).
- 10 Wu, A. *et al.* Expression of MHC I and NK ligands on human CD133+ glioma cells: possible targets of immunotherapy. *Journal of neuro-oncology* **83**, 121-131 (2007).
- 11 Tanaka, K., Hayashi, H., Hamada, C., Khoury, G. & Jay, G. Expression of major histocompatibility complex class I antigens as a strategy for the potentiation of immune recognition of tumor cells. *Proceedings of the National Academy of Sciences* **83**, 8723-8727 (1986).
- 12 Mendoza, J. L. *et al.* Structure of the IFN γ receptor complex guides design of biased agonists. *Nature* **567**, 56-60 (2019).
- 13 Ivashkiv, L. B. IFN γ : signalling, epigenetics and roles in immunity, metabolism, disease and cancer immunotherapy. *Nature Reviews Immunology* **18**, 545-558 (2018).
- 14 Tau, G. Z., Cowan, S. N., Weisburg, J., Braunstein, N. S. & Rothman, P. B. Regulation of IFN- γ signaling is essential for the cytotoxic activity of CD8+ T cells. *The Journal of Immunology* **167**, 5574-5582 (2001).
- 15 Jorgovanovic, D., Song, M., Wang, L. & Zhang, Y. Roles of IFN- γ in tumor progression and regression: a review. *Biomarker research* **8**, 1-16 (2020).
- 16 Yuba, E. *et al.* pH-sensitive polymer-liposome-based antigen delivery systems potentiated with interferon- γ gene lipoplex for efficient cancer immunotherapy. *Biomaterials* **67**, 214-224 (2015).
- 17 Wu, J. *et al.* Dynamic distribution and expression in vivo of the human interferon gamma gene delivered by adenoviral vector. *BMC cancer* **9**, 1-7 (2009).
- 18 Gocher, A. M., Workman, C. J. & Vignali, D. A. Interferon- γ : teammate or opponent in the tumour microenvironment? *Nature Reviews Immunology* **22**, 158-172 (2022).
- 19 Todd, P. A. & Goa, K. L. Interferon gamma-1b: a review of its pharmacology and therapeutic potential in chronic granulomatous disease. *Drugs* **43**, 111-122 (1992).
- 20 Razaghi, A., Owens, L. & Heimann, K. Review of the recombinant human interferon gamma as an immunotherapeutic: Impacts of production platforms and glycosylation. *J Biotechnol* **240**, 48-60, doi:10.1016/j.jbiotec.2016.10.022 (2016).
- 21 Gleave, M. E. *et al.* Interferon gamma-1b compared with placebo in metastatic renal-cell carcinoma. *New England journal of medicine* **338**, 1265-1271 (1998).
- 22 Liu, R. Y. *et al.* Adenovirus-mediated delivery of interferon- γ gene inhibits the growth of nasopharyngeal carcinoma. *J Transl Med* **10**, 256, doi:10.1186/1479-5876-10-256 (2012).
- 23 Bourgeois-Daigneault, M. C. *et al.* Oncolytic vesicular stomatitis virus expressing interferon- γ has enhanced therapeutic activity. *Mol Ther Oncolytics* **3**, 16001, doi:10.1038/mto.2016.1 (2016).
- 24 Oh, E., Choi, I.-K., Hong, J. & Yun, C.-O. Oncolytic adenovirus coexpressing interleukin-12 and decorin overcomes Treg-mediated immunosuppression inducing potent antitumor effects in a weakly immunogenic tumor model. *Oncotarget* **8**, 4730 (2017).

- 25 Foreman, P. M., Friedman, G. K., Cassady, K. A. & Markert, J. M. Oncolytic virotherapy for the treatment of malignant glioma. *Neurotherapeutics* **14**, 333-344 (2017).
- 26 Hoshino, Y. *et al.* The adeno-associated virus rh10 vector is an effective gene transfer system for chronic spinal cord injury. *Scientific reports* **9**, 9844 (2019).
- 27 Salunkhe, S., Basak, M., Chitkara, D. & Mittal, A. Surface functionalization of exosomes for target-specific delivery and in vivo imaging & tracking: Strategies and significance. *Journal of Controlled Release* **326**, 599-614 (2020).
- 28 Das, C. K. *et al.* Exosome as a novel shuttle for delivery of therapeutics across biological barriers. *Molecular pharmaceuticals* **16**, 24-40 (2018).
- 29 Conry, R. M. *et al.* Characterization of a messenger RNA polynucleotide vaccine vector. *Cancer Res* **55**, 1397-1400 (1995).
- 30 Pardi, N., Hogan, M. J., Porter, F. W. & Weissman, D. mRNA vaccines - a new era in vaccinology. *Nat Rev Drug Discov* **17**, 261-279, doi:10.1038/nrd.2017.243 (2018).
- 31 Pastor, F. *et al.* An RNA toolbox for cancer immunotherapy. *Nature Reviews Drug Discovery* **17**, 751-767 (2018).
- 32 Qin, S. *et al.* mRNA-based therapeutics: powerful and versatile tools to combat diseases. *Signal Transduction and Targeted Therapy* **7**, 166 (2022).
- 33 Beck, J. D. *et al.* mRNA therapeutics in cancer immunotherapy. *Molecular cancer* **20**, 1-24 (2021).
- 34 Shi, J., Kantoff, P. W., Wooster, R. & Farokhzad, O. C. Cancer nanomedicine: progress, challenges and opportunities. *Nature reviews cancer* **17**, 20-37 (2017).
- 35 Pullan, J. E. *et al.* Exosomes as drug carriers for cancer therapy. *Molecular pharmaceuticals* **16**, 1789-1798 (2019).
- 36 Chaudhary, N., Weissman, D. & Whitehead, K. A. mRNA vaccines for infectious diseases: principles, delivery and clinical translation. *Nature reviews Drug discovery* **20**, 817-838 (2021).
- 37 Zhu, X. *et al.* Comprehensive toxicity and immunogenicity studies reveal minimal effects in mice following sustained dosing of extracellular vesicles derived from HEK293T cells. *Journal of extracellular vesicles* **6**, 1324730 (2017).
- 38 Zheng, G. *et al.* Mesenchymal stromal cell-derived extracellular vesicles: regenerative and immunomodulatory effects and potential applications in sepsis. *Cell and tissue research* **374**, 1-15 (2018).